# Multi-signal regulation of the GSK-3β homolog Rim11 controls meiosis entry in budding yeast

Johanna Kociemba[1], Andreas Christ Sølvsten Jørgensen[2,3], Nika Tadić[4], Anthony Harris [5], Theodora Sideri[1], Wei Yee Chan[1], Fairouz Ibrahim[1], Elçin Ünal[5], Mark Skehel[1], Vahid Shahrezaei [2✉], Orlando Argüello-Miranda [4✉] & Folkert Jacobus van Werven [1✉]

## Abstract

**Starvation in diploid budding yeast cells triggers a cell-fate program culminating in meiosis and spore formation. Transcriptional activation of early meiotic genes (EMGs) hinges on the master regulator Ime1, its DNA-binding partner Ume6, and GSK-3β kinase Rim11. Phosphorylation of Ume6 by Rim11 is required for EMG activation. We report here that Rim11 functions as the central signal integrator for controlling Ume6 phosphorylation and EMG transcription. In nutrient-rich conditions, PKA suppresses Rim11 levels, while TORC1 retains Rim11 in the cytoplasm. Inhibition of PKA and TORC1 induces Rim11 expression and nuclear localization. Remarkably, nuclear Rim11 is required, but not sufficient, for Rim11-dependent Ume6 phosphorylation. In addition, Ime1 is an anchor protein enabling Ume6 phosphorylation by Rim11. Subsequently, Ume6-Ime1 coactivator complexes form and induce EMG transcription. Our results demonstrate how various signaling inputs (PKA/TORC1/Ime1) converge through Rim11 to regulate EMG expression and meiosis initiation. We posit that the signaling-regulatory network elucidated here generates robustness in cell-fate control.**

**Keywords** Rim11; GSK-3β; Meiosis; Yeast; Ume6
**Subject Categories** Cell Cycle; Chromatin, Transcription & Genomics; Signal Transduction

## Introduction

How cells decide to take a new fate is critical for development. The underlying signaling and regulatory programs ensure that fate decisions are taken at the correct time and space. Mis-regulated cell differentiation can be detrimental to development and cause disease pathologies. Understanding the molecular mechanisms, including regulatory controls of underlying cell-fate decisions, remains a longstanding aim in molecular, cell, and developmental biology.

In budding yeast, cells can undergo a cell-fate decision called sporulation or gametogenesis during which a diploid cell produces four haploid spores. During sporulation, diploid cells undergo premeiotic DNA replication, two consecutive chromosome divisions called meiosis, and subsequently packaging into haploid spores (Marston and Amon, 2004). The sporulation program is essential for the reshuffling of genetic information and for the survival of yeasts during harsh environmental conditions.

The regulatory program that controls entry into meiosis is set in motion by a master regulator called Ime1 (Nachman et al, 2007; Kassir et al, 1988; van Werven and Amon, 2011). Cells lacking *IME1* will not enter meiosis nor form spores. *IME1* transcription is highly regulated through its complex promoter. The *IME1* promoter is repressed in cells harboring a single mating type, which ensures that haploid cells cannot enter meiosis (Covitz and Mitchell, 1993; van Werven et al, 2012). Moreover, the target of rapamycin complex 1 (TORC1) and protein kinase A (PKA) signaling keep the Tup1-Cyc8 co-repressor at the *IME1* promoter in a repressed state in nutrient-rich conditions (Tam and van Werven, 2020; Weidberg et al, 2016). When diploid cells are starved, TORC1 and PKA signaling to *IME1* promoter is not active, Tup1-Cyc8 is relieved from the *IME1* promoter allowing for *IME1* transcription and consequently cells enter meiosis.

Ime1 is not sufficient to drive entry into meiosis. Two additional proteins, Ume6 and Rim11, are part of the regulatory circuit that activates transcription of the early meiotic genes (EMGs) (Bowdish and Mitchell, 1993; Bowdish et al, 1995; Rubin-Bejerano et al, 1996; Malathi et al, 1999, 1997). Ume6 is the DNA-binding component, while Ime1 harbors the transcriptional activation domain function. Under nutrient-rich conditions Ume6 is a repressor of EMGs, by recruiting the co-repressor and histone deacetylase complex Sin3-Rpd3L (Harris and Ünal, 2023; Raithatha et al, 2021; Bowdish and Mitchell, 1993; Carrozza et al, 2005). When diploid cells are starved, Ume6 and Ime1 interact, which drives the transcription of the EMGs (Harris and Ünal, 2023; Raithatha et al, 2021). The kinase Rim11 and GSK-3β homolog also plays a central role in the activation of EMGs. Rim11

[1]The Francis Crick Institute, 1 Midland Road, London NW1 1AT, UK. [2]Department of Mathematics, Imperial College London, London SW7 2BX, UK. [3]I-X Centre for AI In Science, Imperial College London, White City Campus, 84 Wood Lane, London W12 0BZ, UK. [4]Department of Plant and Microbial Biology, North Carolina State University, Raleigh, NC 27695-7612, USA. [5]Department of Molecular and Cell Biology, University of California, Berkeley, Berkeley, CA 94720, USA. ✉E-mail: v.shahrezaei@imperial.ac.uk; oargell@ncsu.edu; folkert.vanwerven@crick.ac.uk

phosphorylates both Ime1 and Ume6, which is required for the two proteins to interact and to activate the transcription of EMGs (Xiao and Mitchell, 2000; Bowdish et al, 1994; Washburn and Esposito, 2001; Malathi et al, 1997). Moreover, Rim11-directed phosphorylation of Ime1 is also essential for the Ime1 transcriptional activation domain function when part of the Ume6-Ime1 complex (Xiao and Mitchell, 2000; Bowdish and Mitchell, 1993). Rim11 can also directly interact with Ime1 in vitro (Bowdish et al, 1994). Thus, the Ime1-Ume6-Rim11 regulon drives entry into meiosis. Transcriptional control of *IME1* by external and cell-intrinsic cues is well understood. However, the regulatory mechanisms governing Rim11 remain largely unexplored.

Here, we show that Rim11 acts as an integrator of multiple signals to robustly regulate EMG transcription and thus meiosis. We observe that Rim11 expression and localization undergo meticulous regulation contingent upon nutrient availability. In nutrient-rich conditions Rim11 expression is kept low by PKA and is kept mostly cytoplasmic by TORC1. During starvation Rim11 expression is increased and the protein localized to the nucleus. Rim11's presence in the nucleus is indispensable for activating EMGs and initiating meiosis, but it alone is insufficient. Rim11 relies on the presence of Ime1, which serves as a scaffold protein enabling Rim11 to phosphorylate Ume6. The Rim11-directed Ume6-Ime1 complexes drive the recruitment of coactivators, chromatin remodellers, and basal transcription factors. Single-cell analysis and mathematical modeling provide evidence that the accumulation of Rim11 in the nucleus can act as a bottleneck for the induction of EMGs. Collectively, our findings elucidate Rim11's role as an integrator of nutrient and cell type-specific signals (TORC1/PKA/Ime1), orchestrating precise temporal control over Ume6 phosphorylation, EMG transcription, and the regulation of meiotic entry.

# Results

## Rim11 phosphorylates Ume6 in diploid cells induced to enter meiosis

Ime1, Ume6, and Rim11 regulate the transcription induction of EMGs (Bowdish and Mitchell, 1993; van Werven and Amon, 2011). While mating type and nutrient control of the *IME1* promoter and thus Ime1 expression are well understood, much less is known about how Rim11 is regulated (van Werven and Amon, 2011). To obtain insight into Rim11 regulation, we measured Rim11-dependent phosphorylation by western blotting of Ume6, a substrate previously described for Rim11 (Bowdish et al, 1995; Malathi et al, 1999; Xiao and Mitchell, 2000; Zhan et al, 2000). WT and *rim11Δ* diploid *MATa/α* cells grown in the exponential growth phase (exp) showed no difference in Ume6 migration (Fig. 1A). When cells were prompted to enter meiosis in sporulation medium (SPO) from pre-SPO medium, we identified a slower Rim11-dependent migrating Ume6 form at 2, 4, 6, and 8 h in SPO, reaching its peak at 4 h. However, no significant difference was observed at 0, 1, and 24 h in SPO (Fig. 1A). In addition, we observed a subtle, yet noteworthy Rim11-independent Ume6 migration shift between exponentially grown cells and SPO, indicating the possibility of another kinase phosphorylating Ume6. Indeed Ume6 is a substrate for the kinases Rim15 and Mck1 (Vidan and Mitchell, 1997; Xiao and Mitchell, 2000). Our results are consistent with previous work that showed that Ume6 is phosphorylated in a nutrient-dependent manner (Xiao and Mitchell, 2000). We conclude that Rim11-directed

phosphorylation of Ume6 occurs at specific time points during meiosis, indicating that Rim11 is regulated by distinct meiotic entry signals.

## Rim11 expression and localization differ greatly between mitotic and meiotic cells

Considering that the Rim11 substrate, Ume6, functions as a DNA-binding protein, associating with specific DNA sequence motifs and likely exclusively residing in the nucleus, we focused on understanding the regulation of Rim11 nuclear levels and the potential mechanisms involved (Williams et al, 2002; Strich et al, 1994). We fused Rim11 with mNeonGreen (Rim11-mNG), which was co-expressed with a nuclear marker, histone H2B fused to mCherry (Htb1-mCh) (Fig. 1B). The Rim11-mNG strain exhibited a small delay (about 1 h) compared to the wild type (WT) in the kinetics of meiotic entry, indicating that the mNeonGreen tag did not have a severe impact on Rim11 function (Fig. EV1A). Rim11 levels markedly increased in cells entering meiosis compared to cells grown in nutrient-rich conditions (exp.) (Figs. 1B,C and EV1B). This increase in meiotic cells was also reflected at the *RIM11* mRNA levels suggesting that *RIM11* transcription is regulated in a nutrient-dependent manner (Fig. EV1C). In addition, we observed notable changes in Rim11 localization. In cells grown in nutrient-rich conditions (exp.) the majority of Rim11 resided in the cytoplasm, while cells entering meiosis showed strong nuclear localization of Rim11 (Figs. 1D and EV1B). Both Rim11 nuclear intensity (Fig. EV1B) and nucleocytoplasmic ratio (Fig. 1D) were the strongest at 2 h in SPO. These findings highlight substantial variations in Rim11 mRNA and protein levels, as well as localization, between mitotic and meiotic cells.

To correlate meiosis with Rim11 nuclear levels in the same cell, we tracked single cells using fluorescence time-lapse microscopy of Rim11-mNG (Movie EV1). Consistent with the time course analysis of Rim11 localization (Fig. 1D), we observed that Rim11 expression (Figs. 1E and EV1D) and nuclear localization (Fig. 1F,G) increased over time in cells that entered meiosis. Strikingly, we found that cells that entered meiosis showed significantly more Rim11 in the nucleus compared to cells that did not enter meiosis (Fig. 1F,G). This finding suggests that the increased expression and nuclear accumulation of Rim11 are specific to meiotic cells, rather than being a general signal induced by starvation.

Computational alignment of meiotic cells to the onset time of the first meiotic division (MI) revealed that nuclear Rim11 peaked between 3 and 6 h prior to MI (Fig. 1H). Subsequently, Rim11 expression and nuclear localization decreased as cells progressed further into meiosis. By aligning the peaks of the Rim11 nuclear levels from meiotic cells, we observed that Rim11 peaks prior to the increase in histone H2B signal, indicative of premeiotic DNA replication (Fig. 1I). In line with these results, we established a robust correlation between the nuclear Rim11 peak signal and the timing of MI, which occurred ~6 h later (Fig. 1J). This finding implies that nuclear Rim11 levels are a predictive indicator for the onset of meiosis. Thus, the elevated nuclear concentrations of Rim11 correlated with the timing of Ume6 phosphorylation, EMG transcription, and the overall propensity to enter meiosis.

## Rim11 nuclear accumulation is required for entry into meiosis

Next, we examined whether nuclear accumulation of Rim11 is essential for induction of EMG transcription and progression into

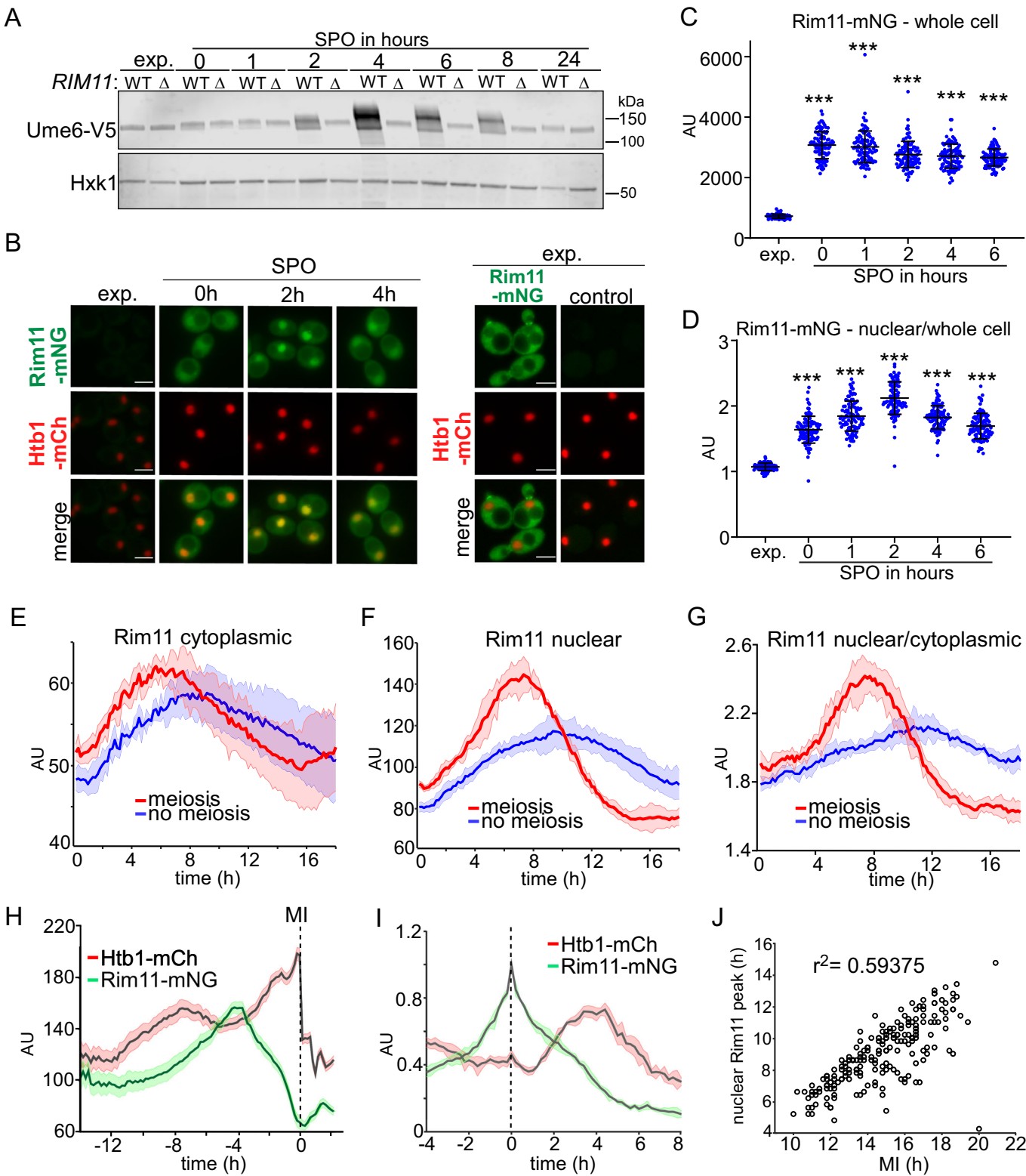

meiosis. To disrupt nuclear Rim11 accumulation, we adopted the anchor-away method as previously described (Haruki et al, 2008). We fused Rim11 to the FRB-GFP (*RIM11-FRB-GFP*) in cells harboring *fpr1Δ*/*RPL13A-2xFKBP12* (Fig. 2A). The rapamycin-induced interaction between Rim11-FRB-GFP and Rpl3a-2xFKBP12 led to an efficient Rim11 nuclear depletion from the nucleus (*RIM11-FRB-GFP* + rapamycin) (Fig. 2B,C; Appendix Fig. S1A). Nuclear depletion of Rim11 abolished meiosis completely

Figure 1.  Rim11 expression and localization dynamics prior to and during meiosis.

(A) Western blot of Ume6 in WT and *rim11Δ*. Cells harboring Ume6 epitope-tagged with V5 in WT or *rim11Δ* (FW1208 and FW10033) were grown until the exponential phase (exp.) or were grown to saturation in rich medium (YPD), then grown for an additional 16 h in presporulation medium (BYTA), and subsequently cells were transferred to sporulation medium (SPO). Samples were taken at the indicated time points. Membranes were probed with anti-V5 antibodies. Hxk1 was used as a loading control. (B) Localization of Rim11 in exponential growth and several time points in SPO. We used a strain with epitope-tagged Rim11 with mNeongreen (Rim11-mNG) and histone H2B epitope-tagged with mCherry (Htb1-mCh) (FW10297). As negative control cells expressing Htb1-mCh was used (FW7794). Scale bar: 5 μm. (C) Quantification of whole-cell Rim11-mNG intensity. Image analysis was carried out using Fiji, where whole-cell selection was determined manually, and nuclei selection was determined via segmentation (Schindelin et al, 2012). The mean values, arbitrary units (AU), were calculated by taking the sum of the gray values of all pixels within the region of interest over the number of pixels. At least $n = 50$ cells were quantified per time point. Black bars with error bars represent the mean+SD. One-way ANOVA. ***$P < 0.001$. (D) Similar to (C), except that the mean nuclear over whole-cell signal is displayed. Black bars with error bars represent the mean+SD. The ratio between the nuclear mean intensity over the whole mean intensity was taken. One-way ANOVA. ***$P < 0.001$. (E–G) Quantification of intensity of Rim11 using live-cell imaging setup using the strain described in (B). Indicated are the mean traces of cells that completed meiosis I (meiosis) and cells that did not enter meiosis (non-meiotic). Shown are the Rim11 cytoplasmic concentrations (E), nuclear (F), and nuclear/cytoplasmic (G). The arbitrary units (AU) displayed represent the total intensity registered in a fluorescent channel after correction for background, autofluorescence, and bleed-through, as described in the fluorescence extraction code as described previously (Kamat, 2019; Arguello-Miranda et al, 2018; Acuña-Rodriguez et al, 2022). At least $n = 200$ cells were quantified per condition (meiosis and no meiosis). The bold colored line represents the mean, and the 95% intervals are highlighted by the thin colored lines. (H) The same data of Rim11 and Htb1 intensities were aligned according to the first meiotic division (MI). (I) Quantification of Rim11 and Htb1 intensity. Cells were aligned according to the Rim11 peak concentration. (J) Scatter plot displaying the timing of Rim11 nuclear peak versus the timing of MI divisions. Source data are available online for this figure.

(*RIM11-FRB-GFP* + rapamycin) (Fig. 2D). Rapamycin treatment had little adverse effect on Rim11-mNG nuclear localization (Appendix Fig. S1B) and meiosis (Fig. 2D) in cells carrying the *fpr1Δ*/*RPL13A-2xFKBP12* alleles.

To assess the effect of Rim11 nuclear depletion on EMG expression, we analyzed its impact on the expression of the protein kinase Ime2. Consistent with the delay in meiosis observed in the *RIM11-FRB-GFP* strain (Fig. EV2A,C), *IME2* was induced with a delay in SPO in mock-treated cells (Fig. EV2C; Appendix Fig. 1C), while *IME2* expression was strongly reduced in cells depleted for nuclear Rim11 (*RIM11-FRB-GFP* + rapamycin) (Figs. 2E and EV2C). In addition, we observed a higher migrating form of Ume6 in the control (*RIM11-FRB-GFP* + mock) but not in cells with nuclear depletion of Rim11 (*RIM11-FRB-GFP* + rapamycin) at 48 h in SPO (Fig. 2F). We conclude that nuclear accumulation of Rim11 is required for Ume6 phosphorylation, leading to the induction of *IME2* transcription and facilitating meiotic entry.

Having established that Rim11 nuclear localization is required for meiotic entry, we examined whether Rim11 nuclear localization alone is sufficient to induce Ume6 phosphorylation and trigger meiosis. We forced nuclear Rim11 by adding a nuclear localization sequence (NLS) to the amino terminus of Rim11 expressed from the *CUP1* promoter (*pCUP1-NLS-RIM11*) (Appendix Fig. S1D). Driving overexpressed Rim11 into nuclei with *pCUP1-NLS-RIM11* did not noticeably affect the migration of Ume6 (Fig. 2G) and modestly delayed the kinetics of meiotic entry (Appendix Fig. 1E). Thus, high Rim11 nuclear levels are not sufficient for meiotic entry.

## Rim11 expression and localization are regulated by TOR and PKA signaling

Our data emphasize the importance of Rim11 localization for meiotic entry. To investigate the signaling pathways governing Rim11 localization, we determined how Rim11 localization is regulated in different nutrient environments. We shifted cells grown in a rich medium (YPD) to a rich medium lacking a carbon source (YP) (Fig. 3A) or from pre-SPO medium to either SPO or SPO plus glucose (Fig. 3B). Rim11 nuclear localization was significantly increased when cells were shifted from YPD to YP. Conversely, nuclear Rim11 intensity was significantly reduced

when cells were shifted to SPO plus glucose instead of SPO (Fig. 3B). We conclude that a fermentable carbon source (glucose) in the growth medium retained Rim11 outside the nucleus.

The target of rapamycin complex 1 (TORC1) and PKA are signaling kinase complexes for nutrient sensing in eukaryotes including yeasts (Thevelein and de Winde, 1999) (González and Hall, 2017). Both PKA and TORC1 signaling also play a central role in regulating the decision to enter meiosis (Weidberg et al, 2016; Matsuura et al, 1990; Colomina et al, 2003). Inhibition of both PKA and TORC1 prompts diploid cells to induce transcription from the *IME1* promoter, enabling EMG transcription and entry into meiosis (Weidberg et al, 2016). We investigated whether TORC1 and PKA control Rim11 expression and localization.

Inhibiting TORC1 with rapamycin in both YPD and SPO conditions resulted in enhanced Rim11-mNG localization compared to untreated cells (Fig. 3A). Likewise, cells shifted to SPO plus rapamycin showed increased nuclear localization of Rim11 (Fig. 3B), and a more rapid increase in Ume6 phosphorylation (Appendix Fig. S2A). Inhibiting PKA activity with 1NM-PP1 in a strain with the analog-sensitive *tpk1-as* allele and gene deletions in *TPK2* and *TPK3* led to an overall increase in Rim11-mNG signal in YPD, while Rim11 nuclear localization remained unchanged (Fig. 3C). When both PKA and TORC1 were inhibited, we observed a significant increase in Rim11 expression and nuclear localization (Fig. 3C). We conclude that PKA predominantly regulates Rim11 expression, while TORC1 controls Rim11 cellular localization.

To gain a further understanding of the dynamics of Rim11 localization in response to nutrient availability, we conducted live-cell imaging. We starved cells for 24 h in SPO, subsequently changed the medium to synthetic complete plus glucose (SCD). Cells that had not entered meiosis and showed Rim11-mNG in the nucleus after 24 h in SPO were used for the analysis. During the transition from starvation to nutrient-rich conditions, Rim11 was downregulated gradually (50% reduction in 70 +/− 40 min) (Fig. 3D). Interestingly, Rim11 depletion was much faster in the nucleus (Fig. 3E) compared to the cytoplasm (Fig. 3F) (18 ± 6 min for a 50% reduction in nuclear levels vs 72 ± 40 min minutes for 50% reduction in cytoplasmic levels), suggesting a switch-like change in the nuclear to cytoplasmic ratios. The difference in

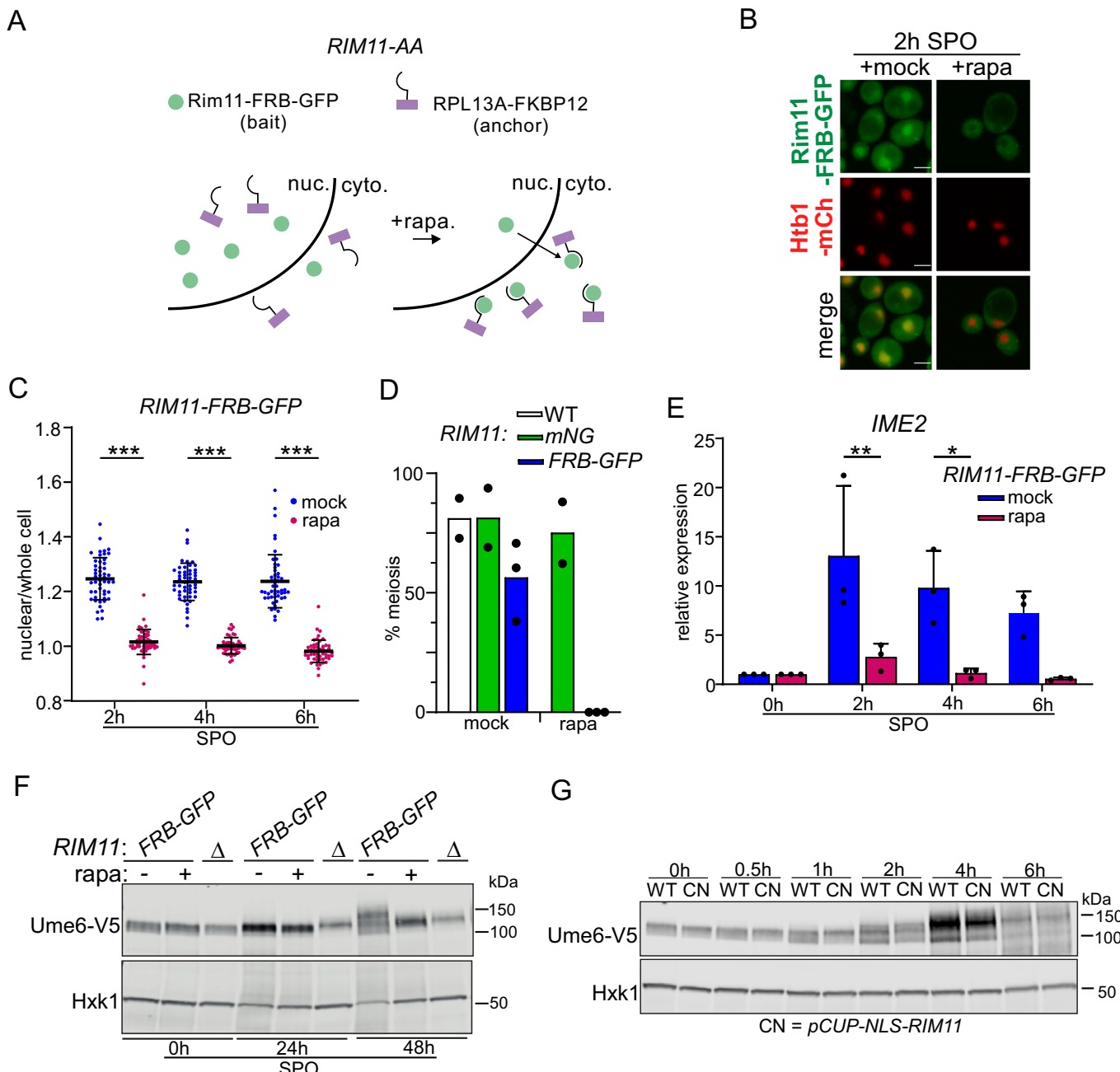

**Figure 2. Rim11 nuclear localization is required for meiosis, but not sufficient.**

(**A**) Schematic of anchoring away of Rim11 from the nucleus. A strain was used harboring *RIM11* tagged with FRB-GFP, *RPL13A* tagged with 2xFKBP12, and gene deletion of *FPR1* (FW11124). (**B**) Rim11-FRB-GFP localization in cells that were induced to enter meiosis and that were mock-treated or treated with rapamycin for 2 h. Scale bar: 5 µm. (**C**) Quantification of nuclear over whole-cell signal of cells and conditions described in (**A, B**). At least 50 cells were per time pointed were quantified. Black bars with error bars represent the mean+SD. One-way ANOVA; ***$P < 0.001$. (**D**) Quantification of cells that entered meiosis of strain and conditions described in (**A, B**) (FW1511, FW11126, FW11124). Meiosis was quantified using DAPI staining. Cells that contained two or more DAPI masses were considered meiosis. The mean of $n = 2$ experiments is shown. At least 200 cells were used for the analysis. (**E**) *IME2* expression determined by RT-qPCR in strain and condition described in (**A, B**). Cells were mock-treated or rapamycin-treated, and samples were taken at the indicated time points. RT-PCR was performed. Samples were normalized to the time point 0 h. $n = 3$ biological repeats were performed. The *IME2* signals were normalized over *ACT1*. Two-way ANOVA. **$P < 0.01$; *$P < 0.05$. (**F**) Western blot of Ume6-V5 in cell harboring *RIM11-FRB-GFP*, *RPL13A-FKBP12* and *fpr1Δ* (FW11371). Cells were induced to enter meiosis and mock-treated or rapamycin-treated. Samples were taken at the indicated time points for western blotting. Membranes were probed with anti-V5 antibodies. As a loading control, Hxk1 was used. (**G**) The effect of overexpression of Rim11 on Ume6 phosphorylation as determined by western blotting. The NLS sequences were fused to Rim11 and expressed from the *CUP1* promoter (FW11394) in a single-copy integration plasmid in a stained background where the endogenous *RIM11* gene was deleted. Cells were treated with 50 µM CuSO4 at 0 h in SPO. As a control, a matching WT *RIM11* strain was used (FW11395). Source data are available online for this figure.

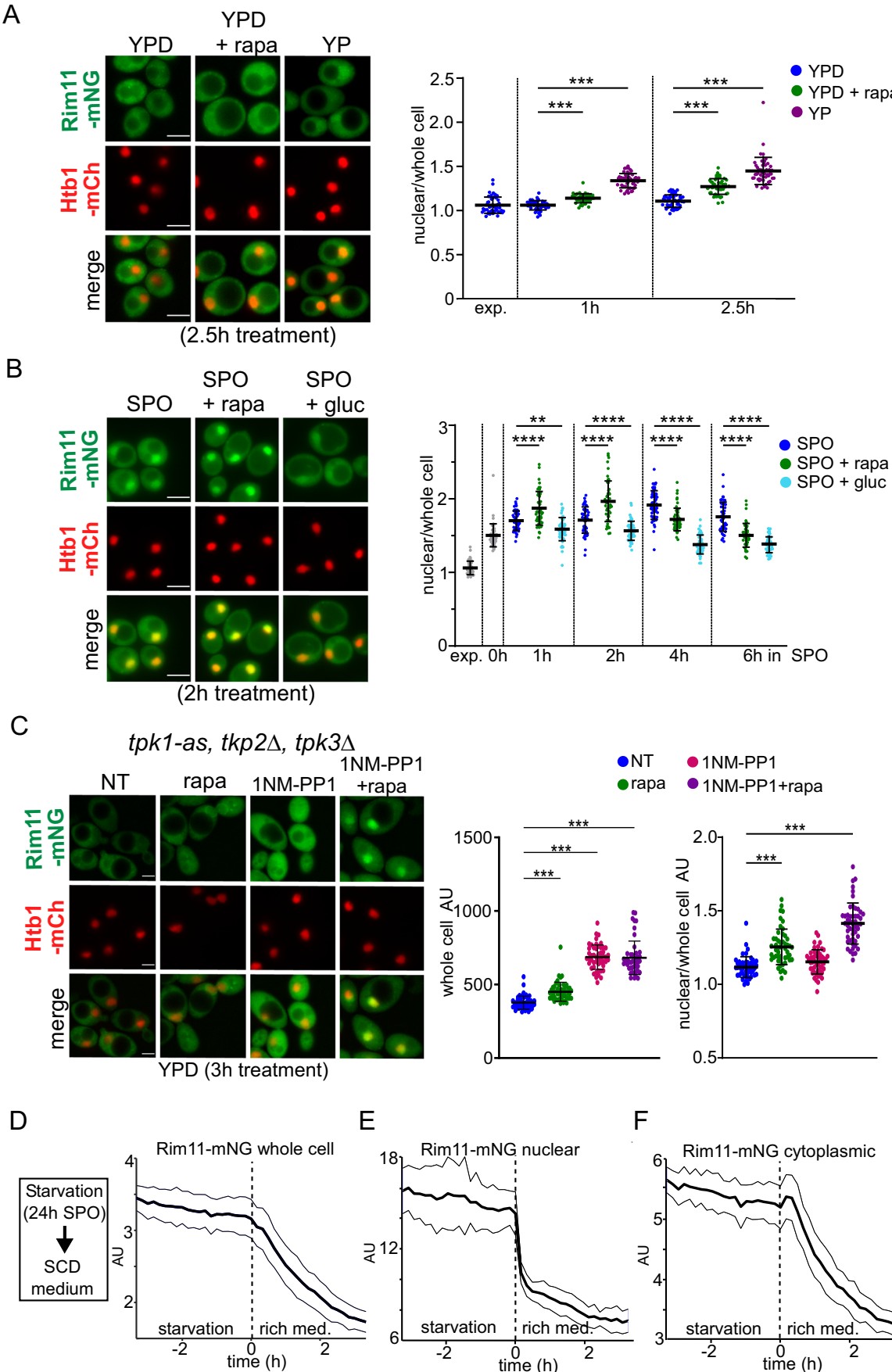

◀ **Figure 3.** **TORC1 and PKA control Rim11 localization and expression.**

(A) Rim11-mNG (FW10297) localization upon rapamycin treatment. Cells were grown in rich medium to exponential growth (exp) and treated with rapamycin for the indicated times. Representative images are shown (left), and nuclear over whole-cell quantification is shown (right). At least $n = 50$ cells per condition were quantified. Black bars with error bars represent the mean+SD. One-way ANOVA; ***$P < 0.001$. Scale bar: 5 µm. (B) Similar analysis as in A, except that cells were induced to enter meiosis in SPO, SPO + rapamycin, and SPO + glucose. At least $n = 50$ cells per condition were quantified. Black bars with error bars represent the mean+SD. One-way ANOVA; **$P < 0.01$; ****$P < 0.0001$. Scale bar: 5 µm. (C) Similar analysis as in (A), except that Rim11-mNG also contained *tpk1-as* allele described previously (FW10483) was used for the analysis both whole cell and nuclear over whole-cell signals were quantified. Cells were grown in rich medium to exponential growth (exp.) treated with 1NM-PP1, rapamycin or both compounds for 3 h. At least $n = 50$ cells per condition were quantified. Black bars with error bars represent the mean+SD. One-way ANOVA; ***$P < 0.001$. Scale bar: 5 µm. (D–F) Rim11-mNG localization dynamics during the shift from starvation to rich medium in whole cell, nucleus, and cytoplasm (FW10297). Cells were starved under microfluidics conditions for 24 h before being exposed to synthetic complete medium (SCD) for 4 h. Shown is the Rim11-mNG signal. The data were centered on the starvation to rich medium transition. At least $n = 200$ cells were quantified. The bold line represents the mean, and the 95% intervals are highlighted by the thin lines. Source data are available online for this figure.

regulation of nuclear and cytoplasmic pools of Rim11 further suggests that distinct pathways, namely TORC1 and PKA, govern nuclear and cytoplasmic Rim11 levels, respectively.

## PKA-mediated phosphorylation controls Rim11 accumulation

Previous work suggested that PKA directly phosphorylates amino-terminal residues in Rim11 (Rubin-Bejerano et al, 2004). When three serines at amino terminus were substituted with alanines (*rim11-3SA*) to prevent PKA phosphorylation, Rim11 kinase activity increased. Consequently, cells were able to enter meiosis in the presence of glucose (Rubin-Bejerano et al, 2004). We determined whether PKA directed phosphorylation regulates Rim11 expression and localization by examining the *rim11-3SA* mutant. We found that Rim11 protein expression was increased in the *rim11-3SA* mutant (Fig. 4A), but Rim11 localization was not altered (Fig. 4B, left) during the first few hours after cells were transferred to SPO or to SPO plus glucose (Appendix Fig. S3B; Fig. 4B, right). In addition, the onset of meiosis was not affected in the *rim11-3SA* mutant compared to the control (Appendix Fig. S3A).

We also performed live-cell imaging with the *rim11-3SA* mutant. Consistent with the still images, we found that the whole-cell intensity, but not nuclear intensity, was higher for *rim11-3SA* compared to the control (Fig. EV3A). Interestingly, *rim11-3SA* mutant displayed more rapid onset of meiosis in the live-cell imaging setup (Fig. EV3B). These data suggest that PKA phosphorylation destabilizes the Rim11 protein, resulting in reduced Rim11 levels and, consequently, a diminished ability to enter meiosis.

## Mds3 promotes Rim11 localization via TORC1

Next, we determined how TORC1 regulates Rim11 nuclear localization. TORC1 is a multi-subunit complex with several downstream effectors that could be involved in mediating TORC1 signaling to Rim11 (Wullschleger et al, 2006). We identified the paralogs Mds3 and Pmd1 as two potential candidates to regulate Rim11 localization via TORC1. Mds3 was identified as a TORC1 effector that controls morphogenesis in *Candida albicans* (Zacchi et al, 2010). In budding yeast, *mds3Δ* and *pmd1Δ* suppress the meiotic defect of *mck1Δ* cells, another GSK-3 homolog that can phosphorylate Ume6 (Benni and Neigeborn, 1997). In addition, *mds3Δ* and *pmd1Δ* cells display an aberrant expression of meiotic

genes in vegetative growth. Moreover, there is a potential interaction between Rim11 and Pmd1, as reported in previous studies (Ho et al, 2002; Breitkreutz et al, 2010).

We measured *IME2* expression (Fig. 4C) and meiosis in *mds3Δ* and *pmd1Δ* cells (Fig. 4D). We found that *IME2* expression was more rapidly induced in *mds3Δ* cells, but not in *pmd1Δ* cells. Likewise, the onset of meiosis was more rapid in *mds3Δ* cells (Fig. 4D). These data suggest that Mds3, but not Pmd1, acts as a repressor of *IME2* expression, consequently influencing the onset of meiosis.

Next, we examined whether Mds3 controls Rim11 localization and Ume6 phosphorylation. Consistent with the *IME2* expression data (Fig. 4C), Rim11 localized to the nucleus more in *mds3Δ* compared to the control for the early time points in SPO (Fig. 4E), while whole-cell Rim11 levels were reduced (Appendix Fig. S3C). In addition, slower Ume6 migrating forms were consistently more abundant in *mds3Δ* cells compared to WT in the first hour after shifting cells to SPO (Fig. 4F; Appendix Fig. S3D). Ume6 levels were also lower in *mds3Δ* cells compared to the WT (Fig. 4F; Appendix Fig. S3D), which could affect the onset of meiosis in the *mds3Δ* mutant. However, when we expressed Ume6 from the *CUP1* inducible promoter, the leaky expression from the *CUP1* promoter, which was lower than in the WT, was sufficient for wild-type like entry into meiosis (Fig. EV3D). In contrast, induced expression of Ume6 was significantly higher than in the WT and had a negative effect on meiotic entry. Together our data suggests that Mds3 regulates Rim11 localization, and consequently Ume6 phosphorylation and meiotic entry.

Finally, to determine whether Mds3 acts via TORC1, we treated cells entering meiosis with rapamycin and determined *IME2* expression (Fig. EV3C). Consistent with the observation that the rapamycin treatment led to faster Rim11 localization to the nucleus (Fig. 3B), we found that *IME2* expression was more rapidly induced (Fig. EV3C). In *mds3Δ* cells, however, we observed no difference in the induction of *IME2* expression between rapamycin and mock-treated cells. We conclude that while Mds3 is dispensable for meiosis entry, it plays a specific role in facilitating signaling from TORC1 to Rim11, thereby influencing the timing of nuclear Rim11 localization and EMG expression.

## Rim11 kinase activity and Ime1 contribute to Rim11 nuclear accumulation

In addition to the impacts of TORC1, the localization of Rim11 may also be influenced by its kinase activity and interacting

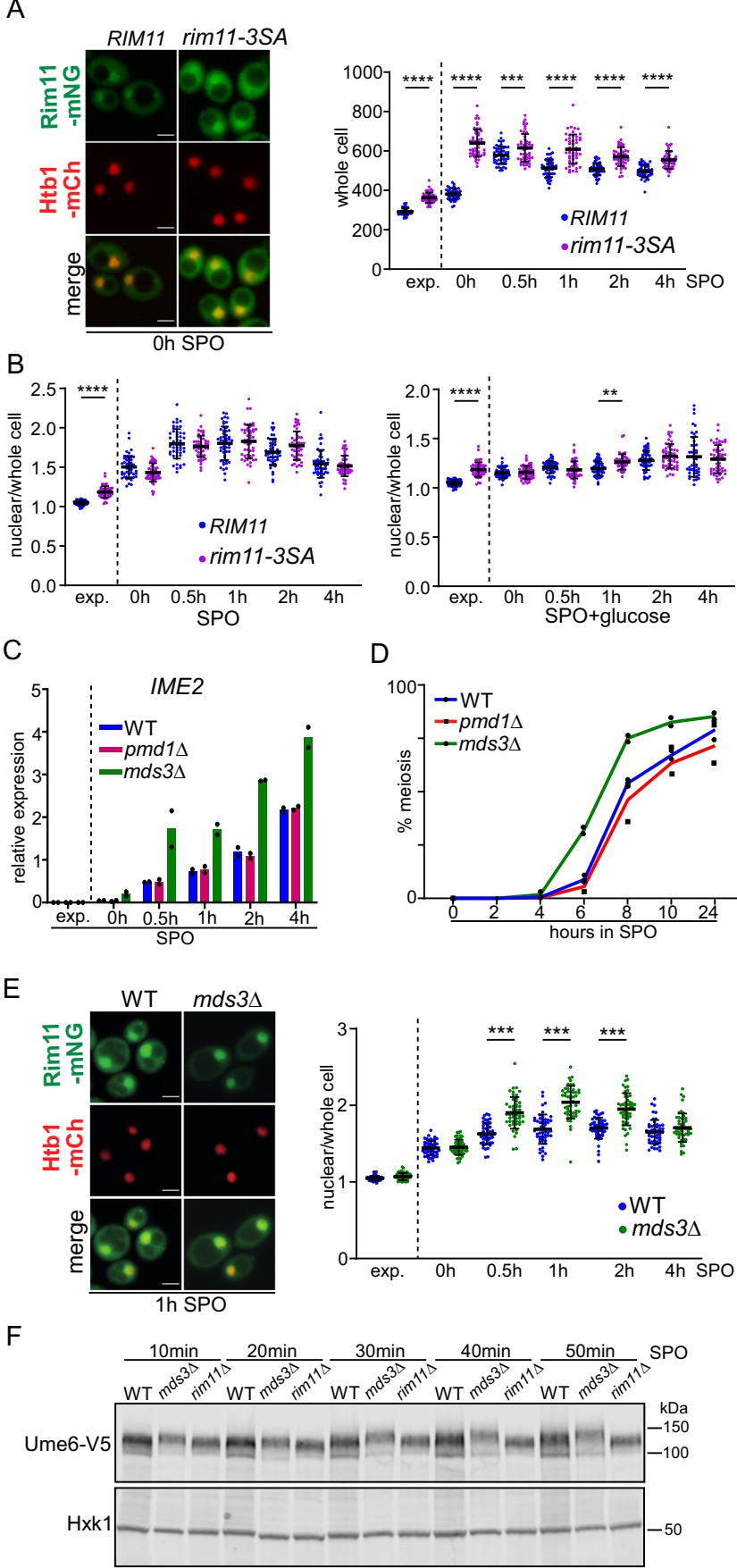

**Figure 4. PKA and TORC1 control Rim11 via distinct mechanisms.**

(A) Localization of Rim11-mNG. Cells harboring *rim11Δ* expressing and integration plasmid harboring *RIM11-mNG* and *rim11-3SA-mNG* (FW10776 and FW10778) were induced to enter meiosis and samples were taken at the indicated time points, whole-cell signal of Rim11-mNG was determined. Shown are example images for the 0-h time points (left), and the whole-cell quantification (right). At least $n = 50$ cells per condition were quantified. Black bars with error bars represent the mean+SD. One-way ANOVA; ***$P < 0.001$; ****$P < 0.0001$. Scale bar: 5 µm. (B) Similar analysis as in A but showing nuclear over whole-cell signal of cells induced to enter meiosis in SPO (left) or SPO plus 2% glucose (right). At least $n = 50$ cells per condition were quantified. Black bars with error bars represent the mean + SD. One-way ANOVA; **$P < 0.01$; ****$P < 0.0001$. (C) *IME2* expression in cell induced to enter meiosis in WT, *mds3Δ*, and *pmd1Δ*. (FW10297, FW10716, FW10718). *IME2* expression signals were normalized over *ACT1*. The mean signals and range of $n = 2$ repeats are shown. (D) Onset of meiosis of strains described in (C). The mean + range of $n = 2$ is shown, and least $n = 100$ cells per biological repeat were quantified. (E) Rim11-mNG nuclear over whole-cell localization in WT and *mds3Δ* cells induced to enter meiosis (FW10297, FW10718). Shown are example images for the 1-h time points (left), and the nuclear over whole-cell quantification (right). At least $n = 50$ cells per condition were quantified. Black bars with error bars represent the mean + SD. One-way ANOVA; ***$P < 0.001$. Scale bar: 5 µm. (F) Ume6-V5 migration as determined by western blotting in WT, *mds3Δ*, and *rim11Δ* (FW1208, FW11251, FW10033). Cells were induced to enter meiosis and samples were taken at the indicated time points. Membranes were probed with anti-V5 antibodies and Hxk1 antibodies as a loading control. Source data are available online for this figure.

proteins. Since both Ime1 and Ume6 operate within the nucleus, we investigated whether Rim11's kinase activity and its interaction with Ime1 play a role in determining Rim11's localization and expression.

First, we examined whether Ime1 and Rim11 expression were dependent on each other. We determined the stability of both proteins by blocking translation in cells with cycloheximide (Fig. EV4A). We observed that Ime1 is relatively unstable compared to Hxk1. In contrast, Rim11 remained unaffected by cycloheximide treatment, suggesting that Rim11 is a relatively stable protein. There was little difference in Ime1 and Rim11 stability in *rim11Δ* and *ime1Δ*, respectively. We conclude that the expression and stability of Ime1 and Rim11 are not dependent on each other.

To examine whether Rim11 kinase activity regulates its localization, we mutated its conserved autophosphorylation site Y199 to F (*rim11-Y199F*), which impairs Rim11 kinase activity (Zhan et al, 2000). As expected, *rim11-Y199F* was impaired for Ume6 phosphorylation (Fig. EV4B) and induction of *IME2* expression (Appendix Fig. S4A). While *rim11-Y199F* whole-cell levels were unaffected (Appendix Fig. S4B), the nuclear levels were significantly reduced (Fig. 5A). In *ime1Δ* cells, Rim11 nuclear localization was also significantly reduced (Fig. 5B). We conclude that both Rim11 kinase activity and Ime1 contribute to the accumulation of Rim11 in the nucleus.

## Ime1 is required for Rim11-dependent Ume6 phosphorylation

Rim11 phosphorylates both Ime1 and Ume6, which is necessary for the interaction between these two proteins and the subsequent initiation of EMG transcription (Bowdish et al, 1994; Rubin-Bejerano et al, 1996; Malathi et al, 1997). Additionally, there is a direct interaction between Ime1 and Rim11 (Malathi et al, 1999). This led us to investigate how Rim11 regulates Ime1 phosphorylation and whether Ime1 plays a role in Rim11-mediated Ume6 phosphorylation.

To ascertain the timing of Ime1 phosphorylation by Rim11, we induced Ime1 expression from the *CUP1* promoter (*pCUP1-HA-IME1*) in both WT and *rim11Δ* cells under pre-SPO and SPO conditions. In a Rim11-dependent manner, a significant portion of Ime1 protein exhibited higher migration patterns (Fig. EV4C,D), which was reversed when extracts were treated with a phosphatase

(Fig. EV4E). Notably, cells expressing Ime1 under nutrient-rich conditions (exp.) did not show a difference in the migration patterns of Ime1 between WT and *rim11Δ* (Fig. EV4C). This finding suggests that Rim11 is not active in nutrient-rich conditions. However, unlike Ume6, Ime1 is phosphorylated in a Rim11-dependent manner in cells grown in a presporulation medium.

Next, we determined whether Ime1 contributes to Rim11-mediated Ume6 phosphorylation. Nutrient starvation-exposed *ime1Δ* cells did not exhibit Rim11-dependent Ume6 migration shift, indicating that Ime1 is required for Rim11 phosphorylation of Ume6 (Fig. 5C). To further probe the importance of the Ime1 interaction with Rim11 for Ume6 phosphorylation, we used a previously characterized *IME1* missense mutant (*ime1-L321F*) known to impair the interaction with Rim11 (Bowdish et al, 1994). Like *ime1Δ* cells, *ime1-L321F* cells were also defective in Ume6 phosphorylation (Fig. 5D) and displayed no induction of *IME2* (Appendix Fig. S4C). We conclude that the Rim11-dependent phosphorylation of Ume6 requires Ime1, suggesting that Ime1 functions as an anchor or scaffold protein for Rim11. In addition to TORC1 and PKA signals, Ime1 possibly provides as a third signal input that promotes Rim11-dependent Ume6 phosphorylation and thus activation of EMG transcription.

## TurboID of Ume6 reveals nutrient-dependent interactions with co-repressors and coactivators

Given that Rim11 and Ime1 are essential for Ume6 phosphorylation and subsequent EMG transcription, we sought to understand how the Ime1-Rim11-Ume6 regulon activates EMG transcription. We employed a protein proximity labeling approach. Specifically, we tagged the carboxy-terminus of Ume6 with TurboID for proximity biotin labeling (Ume6-TID), which labels proteins within a 10-nanometer range of Ume6. (Kim et al, 2014). Cells were grown in the presence of biotin, and denatured protein extracts were generated, from which we purified biotinylated proteins using streptavidin beads. Subsequently, the associated proteins were analyzed by mass spectrometry (MS) using label-free quantification. The Ume6-TID exhibited a phosphorylation shift dependent on Rim11 and Ime1 (Fig. 5E). Ume6-TID strain displayed a small delay on meiotic entry compared to the control, indicating that the TID had no major impact on Ume6 function (Appendix Fig. S4D).

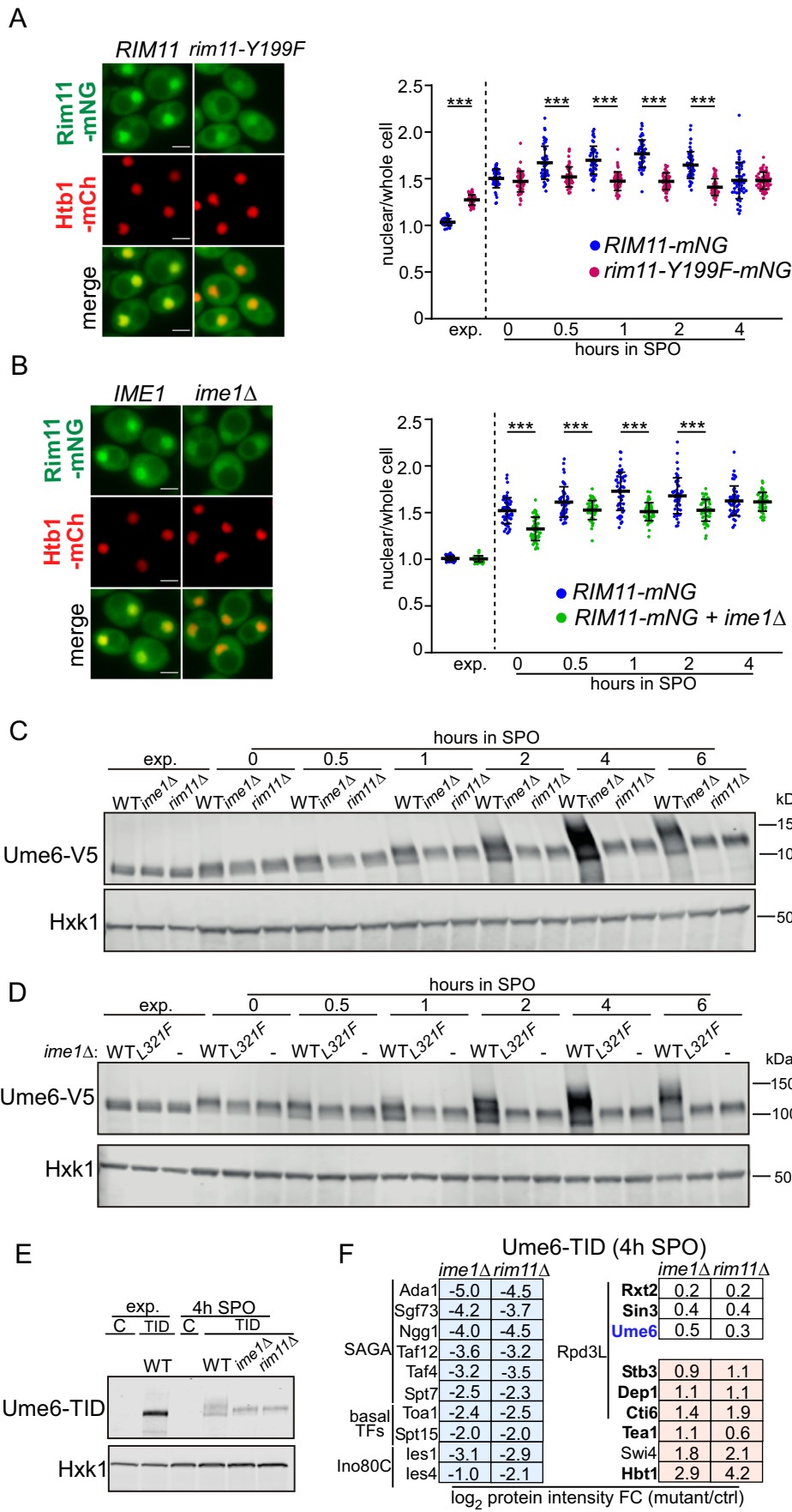

**Figure 5. Ime1 is required for Rim11-directed Ume6 phosphorylation.**

(A) Rim11-mNG nuclear over whole-cell localization in cells harboring *rim11Δ* expressing an integration plasmid harboring *RIM11-mNG* or *rim11-Y199F-mNG* (FW10776, FW10983). Example images are shown (left), and quantification (right). At least *n* = 50 cells per condition were quantified. Black bars with error bars represent the mean +SD. One-way ANOVA; ***P < 0.001. Scale bar: 5 μm. (B) The same analysis in A but comparing WT and *ime1Δ* cells (FW10776, FW10918). At least *n* = 50 cells per condition were quantified. Black bars with error bars represent the mean + SD. One-way ANOVA; ***P < 0.001. Scale bar: 5 μm. (C) Ume6-V5 expression and migration as determined by western blotting in *ime1Δ* and *rim11Δ* cells induced to enter meiosis (FW1208, FW11097, FW10033). Membranes were probed with anti-V5 antibodies and Hxk1 antibodies. (D) Similar analysis as (C) but in *ime1Δ* cells (FW11097) harboring an integration plasmid expressing *sfGFP-IME1* or *sfGFP-ime1-L321F* (FW11231, FW11233). (E, F) TurboID analysis of Ume6. WB showing expression of TurboID tagged Ume6 (Ume6-TID) in exponential phase in WT, and in cells induced to enter meiosis in WT, *ime1Δ* and *rim11Δ* (FW11422, FW11424, FW11426) in E. Log2 protein intensity of *ime1Δ* or *rim11Δ* over control in F. Listed are proteins involved in transcription or known to interact with Ume6. Significant proteins are identified by permutation-based *FDR*-corrected *t* test (threshold: *P* value < 0.05 and s0 = 0.1). Source data are available online for this figure.

First, we compared Ume6-TID profiles between cells grown in nutrient-rich conditions and cells entering meiosis (4 h SPO) (Fig. EV4F; Dataset EV1). For the analysis, we focused on proteins that exhibited significant enrichment in the Ume6-TID sample compared to the untagged control. We specifically selected proteins known to associate with Ume6 or involved in the regulation of chromatin and transcription. As expected, cells entering meiosis displayed a signature of proteins corresponding to the activation of transcription (Fig. EV4F; Dataset EV1). Six subunits of the coactivator SAGA (e.g., Spt7, Ada1), two subunits of the chromatin remodeller Ino80C (Ies1, Ies4), several basal transcription factors (Spt15, Toa1) and transcription elongation factors (FACT, Spt5, Spt6) were enriched in SPO. Conversely, the Tod6 subunit of Sin3-Rpd3L which is known to interact with Ume6, was relatively enriched in nutrient-rich conditions, while Sin3 and Rxt2 of the Sin3-Rpd3L co-repressor were not significantly changed. Tea1, Ty1 enhancer activator I and known interactor of Ume6, was strongly enriched in nutrient-rich conditions (Washburn and Esposito, 2001). Surprisingly, Rim11 and Ime1 were not detected by Ume6-TID. Plausible explanations are that Ime1 is an unstable protein and that Ime1 and Rim11 associate with the amino terminus of Ume6 which falls outside the range of TID at the carboxy-terminus of Ume6 (Kim et al, 2014; Rubin-Bejerano et al, 1996). We conclude that Ume6 proximity labeling can detect nutrient-dependent interactions with protein complexes that repress (Sin3-Rpd3L) or activate transcription (SAGA and Ino80C). Our results are also consistent with previous findings showing that Sin3-Rpd3L and SAGA can be recruited to the promoters of EMGs by Ume6 (Raithatha et al, 2021; Washburn and Esposito, 2001).

## Rim11 and Ime1 are both required for SAGA recruitment to EMGs

Second, we compared profiles of proteins labeled by Ume6-TID between WT, *ime1Δ* and *rim11Δ* in starved cells (4 h SPO). The *ime1Δ* and *rim11Δ* cells showed a decreased enrichment for six subunits of the coactivator SAGA, two subunits of chromatin remodeller Ino80C, and several basal transcription factors (Spt15, Toa1) compared to WT (Fig. 5F; Dataset EV1). Moreover, three subunits of Sin3-Rpd3L (Cti6, Dep1 and Stb3) were enriched in the *ime1Δ* and *rim11Δ* cells compared to the WT, while Sin3 and Rxt2 of Sin3-Rpd3L were not significantly changed, supporting the idea that Ume6 phosphorylation affects the interactions with Sin3-Rpd3L. We conclude that both Ime1 and Rim11 are required for the partial disassembly of Sin3-Rpd3L, and recruitment of SAGA coactivator and basal transcription factors to the promoters of EMGs.

## Tethering an activation domain to Ume6 bypasses the requirement for Rim11 in EMG induction and meiosis

The Rim11-dependent phosphorylated forms of Ume6 and Ime1 make up the transcription activator complex for EMG induction and initiation of meiosis (Bowdish et al, 1994). It is not known whether Rim11 and Ime1 act exclusively in EMG activation within the Ume6 regulon. Recent work showed that Ime1 can be made dispensable for entry into meiosis by tethering a transcription activation domain to Ume6 (Harris and Ünal, 2023). Using a similar approach, we investigated whether both the Rim11 and Ime1 requirements for meiosis can be made dispensable.

First, we measured EMG expression and meiosis in cells where Ime1 is tethered to Ume6 in the presence or absence of Rim11. Specifically, we co-expressed *sfGFP-IME1* and GFP nanobodies tagged *ume6-T99N* (*ume6-T99N-αGFP*) which enables tethering of Ime1 to *ume6-T99N-αGFP* (Harris and Ünal, 2023). The *ume6-T99N* mutation disrupts the interaction with Ime1, impairing EMG transcriptional activation (Bowdish et al, 1995). However, when *sfGFP-IME1/ume6-T99N-αGFP* are tethered, EMG expression is induced, and cells can enter meiosis as demonstrated previously (Harris and Ünal, 2023). We employed the *sfGFP-IME1/ume6-T99N-αGFP* system to determine whether Rim11 is required for EMG transcription by measuring *IME2* levels. We observed that *rim11Δ* cells containing *sfGFP-IME1/ume6-T99N-αGFP* exhibited no *IME2* activation (Fig. 6A) and failed to undergo meiosis (Fig. 6B). This aligns with the model proposing that Rim11 phosphorylation of Ime1 transforms it into a transcriptional activation domain required for initiating EMG transcription and meiotic entry (Bowdish et al, 1994).

To bypass the requirement for Ime1 phosphorylation by Rim11, we replaced the *IME1* locus with GFP fused to the *B112* activation domain (*GFP-B112*) (Harris and Ünal, 2023; Ottoz et al, 2014). Cells expressing *GFP-B112* and *ume6-T99N-αGFP* were able to activate *IME2* transcription (Fig. 6C) and entered meiosis (Fig. 6D) as also demonstrated previously (Harris and Ünal, 2023). *IME2* transcription was also induced in *rim11Δ* cells expressing *GFP-B112* and *ume6-T99N-αGFP* (Fig. 6C), and ~20% of cells entered meiosis (Fig. 6D). We conclude that both Rim11 and Ime1 can be made dispensable, in part, for EMG activation and meiosis.

## Overriding nutrient control of EMG expression

Next, we examined whether nutrient control of EMG transcription can also be bypassed with tethering of *GFP-B112* and *ume6-T99N-αGFP*. Instead of the *IME1* promoter, we expressed *GFP-B112* from

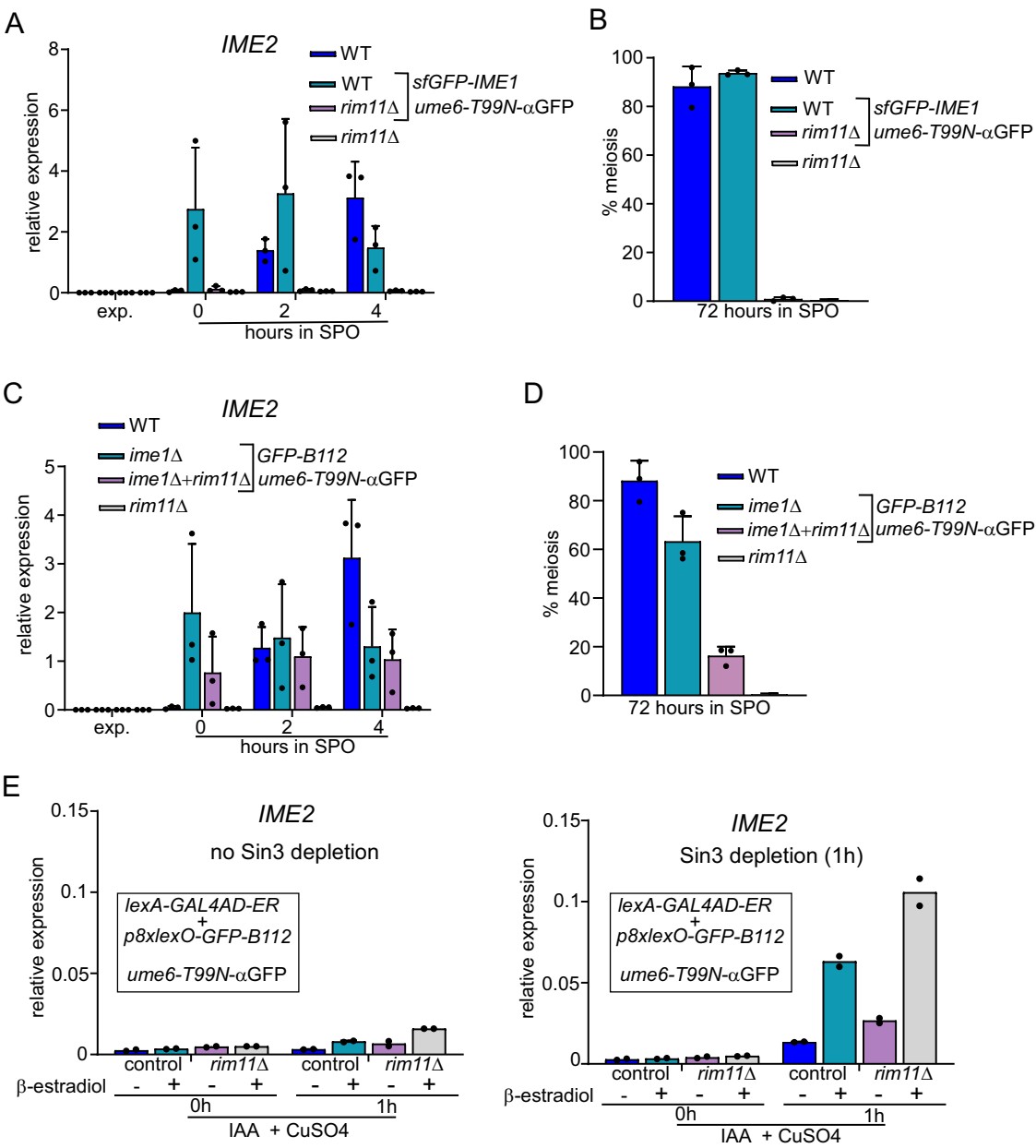

**Figure 6.  Rewiring of Ume6 regulon makes Rim11 and Ime1 partly dispensable.**

(**A**) *IME2* mRNA expression in cells induced to enter meiosis in WT or *rim11Δ* cells in the absence or presence of the *sfGFP-IME1* + *ume6-T99N-αGFP* alleles (FW1511, FW11432, FW10864, FW9277). *IME2* expression was normalized over *ACT1*, and *n* = 3 of biological repeats are shown. Black bars with error bars represent the mean + SEM. (**B**) The same strains as described in (**A**), but the percentage of cells that entered meiosis is plotted. The mean of *n* = 3 biological repeats are shown. Black bars with error bars represent the mean + SEM. (**C**) Same setup as (**A**), except that *sfGFP-IME1* was replaced with *GFP-B112* activation domain (FW1511, FW11414, FW11428, FW9277). The mean and SEM of *n* = 3 biological repeats are shown. (**D**) The same strains as C but plotted are the percentage of cells that entered meiosis. The mean and SEM of *n* = 3 biological repeats are shown. (**E**) *IME2* mRNA expression in cells grown in the exponential growth phase. Similar strains and setup as (**C**), with the exception that *GFP-B112* was under the control of a promoter featuring eight lexO sites (*p8xlexO*) and *lexA-GAL4AD-ER*. Additionally, cells also expressed *SIN3* tagged with the auxin-induced degron (*SIN3-AID*) in the absence (left) or presence (right) of an inducible OsTIR1 ligase (*pCUP1-OsTIR1*) construct (UB36868, UB37219, UB36870 and UB37221). Cells were treated with 3-indoleacetic acid (IAA) and copper sulfate to induce OsTIR1 for 1 h. *IME2* expression was normalized over *ACT1*, and the mean signal of *n* = 2 biological repeats is shown. Source data are available online for this figure.

a promoter harboring eight lex operator sequences (*p8xlexO-GFP-B112*) together lexA fused to the Gal4 activation domain and estrogen receptor (*lexA-GAL4AD-ER*) in *ume6-T99N-αGFP* cells (Harris and Ünal, 2023; Ottoz et al, 2014). When we induced *GFP-*

*B112* transcription with β-estradiol during exponential growth, we noted that *IME2* expression was only marginally increased, suggesting that the B112 tethered to Ume6 was not sufficient for EMG induction under these conditions (Fig. 6E, left panel). Given

the role of Sin3-Rpd3L in repressing EMGs via Ume6, we reasoned that alleviation of Sin3-Rpd3L repression is possibly a prerequisite of EMG activation (Carrozza et al, 2005; Washburn and Esposito, 2001). We generated an auxin-inducible degron tagged allele of *SIN3* (*SIN3-AID*) to conditionally deplete Sin3 (Appendix Fig. 5A) (Nishimura et al, 2009). *IME2* expression was induced in exponentially grown cells expressing GFP-B112 and depleted for Sin3 both in the absence or presence of Rim11 (Fig. 6E, right panel). Increased expression of *IME2* was not observed in cells not depleted for Sin3 (mock-treated or no Tir1 ligase) (Fig. 6E, left panel) or in cells that contained no *lexA-GAL4AD-ER* (Appendix Fig. S5B). We conclude that nutrient control of EMG transcription via Ime1 and Rim11 can, in part, be bypassed by tethering an activation domain to Ume6 in cells depleted for Sin3-Rpd3L.

### Single-cell analysis of the temporal coordination of the Rim11-Ime1-Ume6 regulon

We showed that Rim11 integrates signals from PKA, TORC1 and Ime1 to regulate induction of EMG transcription and entry into meiosis. The *IME1* promoter also integrates signals from TORC1 and PKA, which control its transcription. How does the variability in Rim11 and Ime1 expression affect the timing of meiosis? To address this question, we monitored Rim11, Ime1, and Ume6 nuclear intensities in single cells using live-cell imaging. We generated a strain with Rim11, Ime1, and Ume6 fluorophore-tagged at the endogenous loci, with mKOκ, sfGFP, and mTFP1, respectively (Tsutsui et al, 2008; Ai et al, 2006; Pédelacq et al, 2006). Additionally, the strain also carried a construct harboring an NLS-containing mRuby3 expressed from the *IME1* promoter to label nuclear meiotic divisions (Bajar et al, 2016). As a readout for EMG expression, we used Ume6 itself because Ume6 directly regulates its own promoter and thus, like EMGs, is transcriptionally induced in early meiosis (Harris and Ünal, 2023).

The four-color strain was able to undergo meiosis and form viable spores (92% viability) in the live-cell imaging setup (Movie EV2). First, we compared cells entering meiosis with cells that did not complete a meiotic division also here referred to as quiescent cells. As expected, nuclear levels of Rim11 were elevated in meiotic cells compared to non-meiotic cells (Fig. EV5A). Furthermore, Ime1 (Fig. 7A) and Ume6 (Fig. EV5B) nuclear levels were upregulated in cells entering meiosis but not in non-meiotic cells.

### Rim11 nuclear levels peak before Ime1 and Ume6

We assessed the nuclear accumulation dynamics of Rim11, Ime1, and Ume6 throughout meiosis. The nuclear accumulation of Rim11 preceded the peak nuclear accumulation of Ime1 and Ume6 regardless of whether cells were aligned to the first meiotic division (Fig. 7B) or the Rim11 peak (Fig. 7C). Ime1 levels peaked ~3 h after the Rim11 peak and then plateaued to decline slowly during MI. Interestingly, after reaching its peak, nuclear Rim11 levels declined significantly, reaching their lowest point at the onset of MI. The decline was also observed for Ume6, although with a delay of approximately 3 h from nuclear Rim11 peak time. We conclude that nuclear Rim11 levels peak well prior to Ime1 and Ume6. Our data suggest that cells setup the meiotic fate decision by altering Rim11 nuclear levels well prior to the onset of meiosis.

### Rim11 and Ime1 nuclear levels explain timing variability in meiosis

We used clustering analysis of the single-cell time series to determine whether the kinetics in the nuclear Rim11 intensities can explain the heterogeneity in timing observed in meiosis. We clustered ($k = 6$) single-cell traces for Rim11 and assessed whether the clustering according to Rim11 could reveal different but coordinated patterns for Ime1 and Ume6 accumulation or MI times (Fig. 7D). Strikingly, clustering according to Rim11 nuclear levels revealed a consistent pattern in which cells that accumulated Rim11 faster also were advanced in Ime1 and Ume6 accumulation and MI. For instance, clusters 3 (Fig. 7D) and 5 (Fig. EV5C) showed rapid nuclear accumulation of Rim11, Ime1, and Ume6, along with the meiotic entry. In contrast, cluster 4 (Fig. 7D) displayed the most delayed nuclear accumulation of Rim11, Ime1, and Ume6, as well as the onset of meiosis.

To confirm the temporal interdependence of Rim11, Ime1, and Ume6 peaks with respect to MI, we calculated the population correlation coefficients and obtained a good correlation between Rim11 nuclear peak and MI ($r^2 = 0.6$), despite the peak of Rim11 occurring approximately 7 h before MI (Fig. EV5D, left panel). By contrast, the size of meiotic cells was not correlated with MI times (Fig. EV5E). Both Ime1 (Fig. EV5C, middle) and Ume6 (Fig. EV5C, right) peak levels also showed a good correlation with MI ($r^2 = 0.83$ and $r^2 = 0.80$, respectively), which occurred ~4 h before MI. This suggests that the time of Rim11 peak accumulation can largely explain the timing of Ume6 accumulation and meiosis.

To evaluate whether the insights on Rim11 signal integration can explain the effects on EMG expression and meiosis, we constructed a mathematical model that incorporates a multi-signal-controlled network, including inferred interactions using ordinary differential equations (Fig. 7F and "Methods"). We defined three signals repressed by PKA/TORC1, (S1, S2, and S3) representing stimulation of Ime1 expression, Rim11 expression, and Rim11 nuclear localization. The model reproduced the mean single-cell time series for cells with meiosis (Fig. 7G, left panel). The model captured the timing of the peaks in nuclear Ime1, Ume6 and Rim11, as well as the amplitude of each signal. The model showed that the signal integration by Rim11 substantially impacts the timing and amplitude of the output signal (Ume6) (Fig. 7G, left panel). Removing the starvation signals to Rim11 (S2 and S3) shifted the Ume6 response in time and decreased its expression (Fig. 7G, right panel). It is worth noting that the model likely overestimated how much "leaky" Rim11 is present when TORC1 and PKA are active. At the single-cell level, when considering the timing of Rim11 and Ime1 nuclear peaks following our single-cell data, the model predicted the timing of EMG/Ume6 accumulation as observed in the data (Fig. EV5F). Overall, the model suggests that the signaling network (Rim11-Ime1-Ume6) architecture ensures sensitivity and robustness of EMG transcription, resulting in entry into meiosis.

## Discussion

Despite over five decades of research, the mechanisms by which cells integrate various signals to decide to enter meiosis and form spores remain poorly understood. Nutrient and mating-type signals

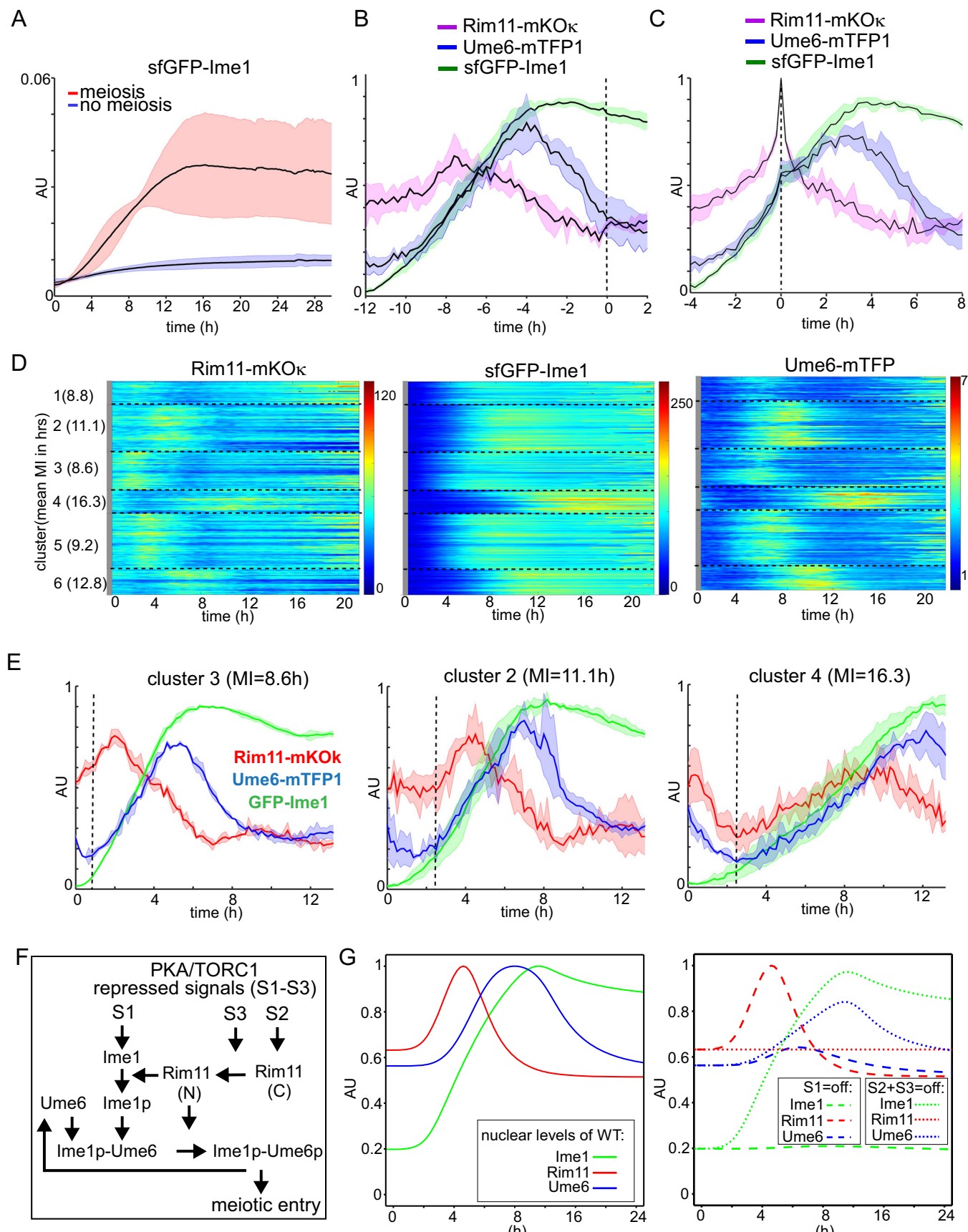

Figure 7.   Single-cell analysis and modeling reveal the timing dynamics Rim11, Ime1, Ume6, and meiosis.

Live-cell imaging of the dynamics of Rim11, Ime1, Ume6 in meiotic and non-meiotic cells. We generated a strain with *sfGFP-IME1*, *UME6-mTFP*, and *RIM11-mKOκ*, and *pIME1-NLS- mRuby3* (FW11243). (A) Nuclear concentrations of sfGFP-Ime1 in meiotic and non-meiotic cells. At least n = 500 cells were quantified. The bold line represents the mean, and the 95% intervals are highlighted by the thin lines. (B) Rim11-mKOκ, sfGFP-Ime1, Ume6-mTFP nuclear intensity in a time course aligned according to the MI division. (C) Time course data aligned according to the Rim11-mKOκ intensity peak is shown. (D) Clustering of single-cell traces according to Rim11-mKOκ nuclear intensity (k = 6). The signals for at least n = 40 cells per cluster are shown. (E) Mean traces of Rim11-mKOκ, sfGFP-Ime1, Ume6-mTFP nuclear intensity for cluster 3 (left), 2 (middle), and 4 (right). The vertical dashed lines indicate the points at which Rim11, Ime1, and Ume6 signals collectively begin to increase thereafter. (F) Scheme of the mathematical model. Ime1 expression is positively regulated by signal S1, Rim11 expression by S2, and Rim11 nuclear localization by S3. Ime1p and Ume6p represent the Rim11-phosphorylated forms. (G) Graph showing the nuclear signal of Rim11-mKOκ, sfGFP-Ime1, and Ume6-mTFP from the mean single-cell time series predicted by the model for the WT (left). Graph showing the mean single-cell time series predicted by the model when switching off the starvation signal to Ime1 (S1) or the starvation signals for Rim11 expression and localization (S2 and S3) (right). Source data are available online for this figure.

regulate transcription of the *IME1* promoter, which is central to the decision to enter meiosis (Tam and van Werven, 2020; Sagee et al, 1998; Rubin-Bejerano et al, 2004; van Werven and Amon, 2011). Moreover, the GSK-3β kinase Rim11 is an essential component of the regulon responsible for inducing EMG transcription (Bowdish and Mitchell, 1993). Our findings reveal a surprising aspect of Rim11 - namely, its ability to integrate multiple signals through distinct mechanisms, ultimately controlling Ume6 phosphorylation and the subsequent induction of EMG transcription (Fig. 7F).

## Rim11 acts as a nutrient signal integrator for meiotic entry

Our data showed that Rim11 integrates nutrient availability signals via PKA and TORC1 to control its expression and localization, respectively. Our data suggest that PKA destabilizes Rim11 protein levels. Previous work showed that PKA directly phosphorylates serines at the amino terminus of Rim11 (Rubin-Bejerano et al, 2004). However, the mechanism by which PKA destabilizes Rim11 remains unexplored. One possibility is that PKA phosphorylation of Rim11 recruits a ubiquitin ligase or the proteasome directly. Such a mechanism has been demonstrated for the RNA-binding protein Cth2. In response to iron depletion, Cth2 undergoes phosphorylation, leading to its destabilization through ubiquitin-mediated proteasome degradation (Romero et al, 2018).

How does TORC1 retain Rim11 outside the nucleus? We provide evidence that Mds3, in part, retains Rim11 in the cytoplasm. The molecular function of Mds3 in TORC1 signaling is not well characterized. There is evidence that Mds3 is part of the Sit4 phosphatase branch of TORC1 (Zacchi et al, 2010). Perhaps dephosphorylation of Rim11 by Sit4 keeps the Rim11 inactive and cytoplasmic. Given that *mds3Δ* had a partial effect on Rim11 localization, it is likely other effectors of TORC1 are involved in retaining Rim11 in the cytoplasm. The Sch9 kinase branch of TORC1 potentially plays a role in Rim11 localization because Sch9 also regulates *IME1* transcription and entry into meiosis (Weidberg et al, 2016). More work is needed to determine the molecular mechanism by which Rim11 localization is controlled by TORC1.

## Multi-layered regulation of Rim11 governs EMG transcription induction

Prior studies demonstrated physical interactions between Ime1 and Rim11, Rim11 and Ume6, and Ime1 and Ume6 (Bowdish et al, 1994; Malathi et al, 1997). In this work we showed that Ime1 acts as the anchor protein for Rim11-dependent phosphorylation of

Ume6. The Ime1 also contributes to Rim11 nuclear localization. The result is most surprising as previous work suggested that Rim11-dependent phosphorylation of Ume6 can occur in the absence of Ime1 (Zhan et al, 2000).

The scaffolding function of Ime1 in conjunction with Rim11 and Ume6 adds a new dimension to how EMG transcription is regulated in cells. Through its interaction with Ime1, Rim11 senses both nutrient availability and mating-type controls of *IME1* transcription regulation, which is necessary for Rim11-dependent Ume6 phosphorylation. Since Rim11 also independently senses both TORC1 and PKA signals, the expression of Ime1 alone is inadequate to induce EMG transcription and initiate meiosis. We propose that the dual role played by Rim11 and Ime1 in integrating nutrient signals via TORC1 and PKA contributes to the robustness and specificity of the regulatory network structure governing EMG transcription. The Rim11-Ime1-Ume6 network structure provides a resilient mechanism for initiating meiosis in response to various environmental signals.

Our findings align with previous work indicating that Ume6 serves as a binding platform for Ime1, facilitating the activation of EMG transcription (Harris and Ünal, 2023; Raithatha et al, 2021; Bowdish et al, 1995). An alternative model proposed that Cdc20-mediated degradation of Ume6 in early meiosis induces EMG transcription activation (Mallory et al, 2007). However, recent analyses suggest a different timeline, indicating that Ume6 degradation occurs later in meiosis. Consequently, our data support the idea that Ume6 serves as a binding hub for Ime1, thereby orchestrating EMG transcription and thus entry into meiosis (Harris and Ünal, 2023; Raithatha et al, 2021; Bowdish et al, 1995).

The single-cell data revealed a strong temporal regulation of Rim11, Ime1, and Ume6. During starvation, cells accumulate nuclear Rim11 ahead of Ime1 and Ume6. A plausible sequence of events following Rim11 nuclear accumulation involves Rim11-dependent phosphorylation of Ime1 within the nucleus, immediately followed by Rim11-dependent phosphorylation of Ume6, thereby initiating the formation of Ume6-Ime1 complexes (Fig. 7F). Once Rim11-phosphorylated Ume6-Ime1 complexes are formed, EMG transcription initiates, and Rim11 exits the nucleus. Exit from the nucleus of Rim11 is also triggered when cells are exposed to nutrient-rich conditions (Fig. 3E). Thus, the Rim11-Ime1-Ume6 regulon is rapidly reversible when environmental conditions change.

We constructed a mathematical model of the Rim11, Ime1, and Ume6 regulatory network. The model included three input signals to Ime1, Rim11 expression and Rim11 localization, and the Ime1 scaffolding function in Ume6 phosphorylation (Fig. 7F and 7G). The model captured the data accurately,

providing additional support for the proposed signaling network architecture. Modeling also indicates that the multi-signal regulation of Rim11 ensures a robust cellular decision-making process.

## GSK-3 kinases conserved signal integrators

In mammals, glycogen synthase kinase-3β (GSK-3β) has been shown to be required for cellular survival, differentiation, and metabolism (Beurel et al, 2015). The GSK-3 kinases have also been shown to play various roles in meiosis, spermatogenesis, and oocyte development, indicating that their roles have been conserved from yeast to humans (Malathi et al, 1999) (Takeo et al, 2012; Wen et al, 2019; Guo et al, 2003; Banerjee and Srayko, 2022). The Ime1 and Ume6 components of the regulatory network are not conserved in mammals.

The localization GSK-3β kinases has been shown to regulate differentiation of mouse embryonic stem cells (mESC) (Bechard and Dalton, 2009). Specifically, GSK-3β can translocate to the nucleus to target c-Myc for phosphorylation and degradation. Aberrant nuclear localization of GSK-3β has also been linked to a negative clinical cancer outcome (Ougolkov et al, 2006). Like TORC1 and Rim11 yeast, mTor has also been shown to be involved in retaining the localization of GSK-3β to the cytoplasm, and inhibiting mTor leads to nuclear GSK-3β where it activates transcription factors such as c-Myc and Snail (Bautista et al, 2018). Thus, TORC1/mTor controlled Rim11/ GSK-3β localization, and the targeting of nuclear transcription factors by Rim11/ GSK-3β is likely a conserved process from yeast to humans. In contrast to human GSK-3β, which has over 100 different substrate targets, the role of Rim11 in meiosis is predominantly centered on Ume6 and Ime1 phosphorylation because Rim11 can be made partly dispensable for meiosis when we rewired the Rim11-Ime1-Ume6 network (Beurel et al, 2015).

In conclusion, we showed that Rim11 is a central integrator of input signals representing the cell's metabolic and mating-type status (TORC1, PKA, and Ime1). The input signals determine whether Rim11 will phosphorylate Ume6 and allow for Ume6-Ime1 complexes to form, which in turn drive EMG transcription and entry into meiosis. Our findings present a new perspective on the regulation of meiotic initiation.

# Methods

## Yeast strain and plasmids

Experiments were carried out with a SK1 strain background of *Saccharomyces cerevisiae*. Gene deletions, depletions, anchor away, and epitope tagging of strains were achieved through a one-step PCR method (Longtine et al, 1998). Genomic integrations were verified by PCR with primers flanking the targeted region and, where feasible, further confirmed through western blotting. All strains used in the study underwent at least one round of genetic crossing. Unless stated differently, experiments were performed in liquid cultures at 30 °C with 300 rpm.

The anchor-away technique was described previously (Haruki et al, 2008). The SK1 strain with anchor-away modifications was kindly provided by the Adele Marston lab. The strain contains the ribosomal anchor protein *RPL13A* fused with two copies of the human FKBP12. The target protein *RIM11* was C-terminally tagged with FRB-GFP. Lastly, *FPR1* was deleted, allowing the human FKBP12 to bind efficiently to FRB.

For generating the *UME6* and *ume6-T99N* plasmids, we used the vector backbone of the single-copy integration plasmid which originates from the pNH605 plasmid (Mitchell et al, 2015). The pNH605 plasmid was digested with BamHI and NotI. The digested pNH605 plasmid and the *UME6* PCR fragment (~700 bp upstream of the START codon, and ~700 bp downstream of the STOP codon) were assembled (NEB, Gibson assembly). To generate *ume6-T99N*, the pNH605 plasmid and two overlapping PCR fragments with *T99N* mutation were assembled (NEB, Gibson assembly). To generate, the *rim11-Y199F* mutation, the genomic *RIM11* locus was first C-terminally tagged with mNeonGreen (mNG). The *RIM11-mNG* locus, including ~700 bp upstream of the START codon, was amplified and assembled with the digested pNH605 plasmid (*pNH605-RIM11-mNG*) (NEB, Gibson assembly). The *pNH605-RIM11-mNG* was used for PCR site-directed mutagenesis (NEB, Q5® Site-Directed Mutagenesis Kit) to generate *pNH605-rim11-Y199F-mNG*. The *ime1-L321F* plasmid was generated by site-directed mutagenesis (NEB, Q5® Site-Directed Mutagenesis Kit) using *pNH604-pIME1-sfGFP-IME1* described previously as the template (Tam and van Werven, 2020). Plasmids were verified by sequencing. The list of the yeast strains and plasmids used in Dataset EV2 and Table EV1.

*UME6* was C-terminally tagged with the TurboID tag including 3 tandem copies of the Myc epitope (TurboID-3xMyc) (Larochelle et al, 2019). For tagging of Rim11 with mKOk and Ume6 with mTFP1, cassette plasmids were used as described previously (Arguello-Miranda et al, 2018).

## Growth conditions

For a standard meiosis time course, cells were inoculated in liquid YPD (1% yeast extract, 2% peptone, 2% glucose, 22.4 mg/L uracil, and 80 mg/L tryptophan) grown to saturation overnight. The cells were transferred to pre-SPO medium (BYTA) (1% yeast extract, 2% bactopeptone, 1% potassium acetate, 50 mM potassium phthalate) $OD_{600} = 0.4$. After 16–18 h, the cells were pelleted and washed once with sterile water before being transferred to liquid sporulation medium (SPO) $OD_{600}$ 2.5. For the mid-log phase samples, cells were inoculated in YPD, grown overnight and then diluted back to $OD_{600} = 0.5$. the next morning. The samples were taken when the cells reached $OD_{600} = 1$.

To inhibit the TORC1, the cells were grown to mid-log phase $OD_{600}$ 1, as described in the standard meiosis protocol. The cells were then treated with 1 μM rapamycin (Sigma-Aldrich, R0395). To inhibit the PKA pathway, a strain containing the *TKP1* analog-sensitive allele (*tpk1-as*) with *TPK2* and *TPK3* deleted was used (Weidberg et al, 2016). PP1 analog 1NM-PP1 was used to inhibit PKA activity (Calbiochem, Merck Millipore). Cells were grown to mid-log phase $OD_{600}$ 1, and subsequently cells were treated with 5 μM of 1NM-PP1.

For the anchor-away experiment, yeast cultures were grown to induce meiosis using a synchronous meiosis protocol (Moretto et al, 2021). After 16 h in BYTA, cells were treated with 1 μM rapamycin (Sigma-Aldrich, R0395) and incubated for 1 h. Then, cells were washed and transferred to SPO, $OD_{600} = 2.5$, and treated with 1 μM rapamycin.

For the Ume6-TID experiment, cells were grown to saturation overnight and diluted back to 50 ml fresh YPD $OD_{600} = 0.25$. $OD_{600} = 0.5$, the cells were treated with 50 µM biotin (Sigma-Aldrich, B4501) and harvested when the cultures reached $OD_{600} = 1$. For sporulation conditions, cells were grown following the standard meiosis protocol. Cells were transferred to 50 ml SPO $OD_{600} = 1.8$ and treated immediately with 50 µM biotin. At 4 h in SPO, cells were harvested.

For mitotic Sin3 depletion assays using auxin-induced degron *SIN3-AID*, cells were grown in YPD. After overnight incubation, cultures were back diluted to an $OD_{600} = 0.2$ and grown for 3 h to exponential phase ($OD_{600} \geq 0.5$). Once the exponential phase was reached (0 h), all cultures were treated with CuSO4 (50 µM final) and 3-indoleacetic acid (IAA; 200 µM final).

To assess the relative protein stability of Rim11 and Ime1, we used the *pCUP1-IME1* and *RIM11-V5* strains. To induce Ime1, cells were treated at 2 h in SPO with CuSO4 (50 µM final) for 2 h, after which cells were subsequently treated with cycloheximide (100 µg/ml). Samples were collected at the specified time points, fixed with TCA, and processed for western blot analysis.

## Western blotting

For western blot (WB) analysis, 5 $OD_{600}$ units were pelleted and resuspended in 1 ml 5% trichloroacetic acid (TCA). Cells were then pelleted and resuspended in 1 ml 100% acetone and centrifuged. The pellets were dried before being resuspended in protein breakage buffer (50 mM Tris pH 7.5, 1 mM EDTA supplemented with 2.7 mM DTT) plus glass beads (Ø 0.5 mm, BioSpec). Cells were lysed in the bead-beater, and subsequently SDS loading buffer (187.5 mM Tris pH 6.8, 5.7% β-mercaptoethanol, 30% glycerol, 9% SDS, and 0.05% Bromophenol Blue) was added and samples were boiled at 95 °C.

To dephosphorylate Ime1, protein extracts were treated with Quick CIP (NEB) for 10 min, which is a heat-labile version of calf intestinal alkaline phosphatase (CIP) purified from a recombinant source, before sample buffer was added and samples were boiled.

The WB samples were loaded on a 4-20% or 7% Midi CriterionTM TGXTM Precast gel (BioRad) and ran at 180 V for 45 min in 1× running buffer (190 mM glycine, 25 mM and 3.4 mM SDS). Trans-Blot SD Semi-Dry Transfer Cell (BioRad) or the Trans-Blot Turbo Transfer System (BioRad) were used for transfer to membranes. The former one was performed in a semi-dry transfer buffer (48 mM Tris Base, 39 mM glycine, 0.04% SDS and 10% methanol) onto a methanol-activated PVDF membrane. The latter one in 1× of commercially available transfer buffer (BioRad) onto a nitrocellulose membrane. The membrane blocked (1% BSA and 1% non-fat powdered milk in 1× phosphate-buffered saline (PBS) buffer supplemented with 0.01% Tween-20 (0.01% PBST), washed 3× in 0.01% PBST buffer and incubated with the primary antibodies added to the blocking buffer overnight at 4 °C. Hexokinase (Hxk1) was used as the loading control for WB. Membranes were washed three times with 0.01% PBST buffer and incubated with secondary antibodies. Membranes were scanned on the Li-COR Odyssey® CLx after incubations with secondary antibodies IRDye 800CW anti-mouse and IRDye 680RD anti-rabbit (LI-COR), or scanned on Amersham Imager 600 Instrument (GE Healthcare) after incubation with anti-mouse and anti-rabbit IgG horseradish peroxidase (HRP)-linked antibodies (Amersham)

or streptavidin HRP-conjugated antibodies (Sigma-Aldrich) and ECL detection Reagents (Sigma-Aldrich). Image processing and quantification were done with the Image StudioTM Lite software.

## Reverse transcription and quantitative PCR

Total RNA was extracted from 5 $OD_{600}$ units using TES buffer (10 mM Tris HCl pH 7.5, 10 mM EDTA (pH8), 0.5% SDS) and acid-phenol:chloroform (Ambion) at 65 °C, followed by ethanol precipitation, DNAse treatment (rDNAsa), and column-purified using NucleoSpin RNA kit (Macherey-Nagel). 500 ng of total RNA was used for reverse transcription using ProtoScript II First Strand cDNA Synthesis Kit (New England BioLabs). For quantification, the QuantStudioTM 7 Real-Time PCR. The gene expression values are relative to the housekeeping gene expression *ACT1*.

For Fig. 6, 5 µg of purified total RNA was then treated with DNase (TURBO DNA-free kit, Thermo Fisher (MA, USA) according to the manufacturer, and 4 µL (<1 µg) of DNase treated total RNA was then reverse transcribed into cDNA with the use of random hexamers (Superscript III Supermix, Thermo Fisher) according to manufacturer's instructions. cDNA was then quantified using the SYBR green mix (Life Technologies (CA, USA)) and measured using the Applied Biosystem StepOnePlusTM Real-Time PCR system (Thermofisher—4376600). Signals were normalized to *ACT1*. PCR primers used are listed in Table EV2.

## Microscopy analysis for fixed images

Images were taken with an Eclipse Ti-E inverted microscope system (Nikon) using the 100X oil objective. Proteins tagged with mNeonGreen (mNG) and superfolder GFP (sfGFP) were imaged with a GFP filter with an exposure time of 500 msec and GFP at 1000 msec, respectively. Proteins tagged with mCherry were imaged with a mCherry filter with an exposure time of 100 msec. Image analysis was carried out using Fiji, where whole-cell selection was determined manually, and nuclei selection was determined via segmentation (Schindelin et al, 2012). The mean values (mean intensity) were calculated by taking the sum of the gray values of all pixels within the selected object over the number of pixels. To investigate on nuclear localization of proteins, the ratio between the nuclear mean intensity over the whole mean intensity was taken.

## DAPI counting for determining meiosis

Cells were resuspended in 1 µg/mL 4',6-diamidino-2-phenylindole (DAPI) in 1× phosphate-buffered saline (PBS) buffer and visualized by fluorescent microscopy. 200 cells were counted per sample. Each cell was assessed for the number of nuclei. Cells with one DAPI mass were considered to have not entered meiosis. Cells with two or more DAPI masses were considered to have entered meiosis. The percentage of cells that completed the first and second meiotic division over the total number of cells was calculated.

## Proximity labeling using Ume6-TurboID

Cells were pelleted, washed two times in sterile water, and four times in RIPA buffer 0.4% SDS (50 mM Tris HCL pH 7.5, 150 mM NaCl, 1.5 mM MglCl$_2$, 1 mM EGTA, 0.04%SDS, 1% NP-40). Cells were resuspended in 2 ml 5% TCA per 5 $OD_{600}$ units incubated

4 °C overnight, pelleted, washed in 100% acetone, resuspended in in TE buffer (100 mM Tris pH 7.5, 1 mM EDTA) supplemented with 2.7 mM DTT, per 18OD units. 500 µl of zirconia/silicate beads (Ø 0.5 mm) were added, and the cells were lysed in the bead-beater. 3×SDS sample buffer (187.5 mM Tris pH 6.8, 30% glycerol, 3% SDS) was added and the samples were boiled. For affinity purification, 400 µl of streptavidin M280 beads (Thermo Fisher) were used per sample, incubated at 4 °C for 3 h. Beads were washed once with wash buffer, four times with RIPA 0.4% SDS buffer and five times with 20 mM ammonium bicarbonate. Beads were resuspended in 50 µl of 50 mM ammonium bicarbonate.

Samples were prepared using on-beads tryptic digestion protocol prior to mass spectrometry. Samples were reduced with 5 mM DTT for 1 h at 37 °C in a thermomixer. Proteins were then alkylated with 10 mM IAA for 30 min in the dark at room temperature. After reduction/alkylation beads were trypsinised in order to digest biotinylated proteins (0.4 µg/µl trypsin in 50 mM $NH_4HCO_3$). Digestion was performed overnight at 37 °C in a thermomixer at 450 rpm. The reaction was stopped with 10% formic acid (FA). Peptides were recovered and the beads were discarded utilizing a magnetic rack. Finally, a C18 clean-up was performed using EV2018 EVOTIP PURE. The desalting was done according to the manufacturer's instructions and protocol. Briefly, the Evotips tips were conditioned with 0.1% Formic acid in acetonitrile; equilibrated with 0.1% Formic acid in water and ~1 µg of each digested sample was loaded onto Evotips. Peptides were eluted from the Evotips with 50% acetonitrile into a vial and vacuum-dried by SpeedVac to remove any traces of organic solvents. Finally, the dried peptides were resuspended in 0.1% formic acid.

## Mass spectrometry (MS)

The resulting peptides were analyzed by nano-scale capillary LC-MS/MS using an Ultimate U3000 HPLC (ThermoScientific Dionex, San Jose, USA) to deliver a flow of ~300 nL/min. A C18 Acclaim PepMap100 5 µm, 75 µm × 20 mm nanoViper (Thermo-Scientific Dionex, San Jose, USA), trapped the peptides prior to separation on an EASY-Spray PepMap RSLC 2 µm, 100 Å, 75 µm × 500 mm nanoViper column (ThermoScientific Dionex, San Jose, USA). Peptides were eluted with a 90 min gradient of acetonitrile (2%v/v to 80%v/v). The analytical column outlet was directly interfaced via a nano-flow electrospray ionization source, to a hybrid quadrupole orbitrap mass spectrometer (Lumos Tribrid Orbitrap mass spectrometer, ThermoScientific, San Jose, USA). A data-dependent analysis was carried out, using a resolution of 120,000 for the full MS spectrum, followed by an MS/MS spectra acquisition in the linear ion trap using "TopS" mode. MS spectra were collected over a m/z range of 300–1800. MS/MS scans were collected using a threshold energy of 32% for collision-induced dissociation.

## MS data analysis

LC-MS/MS raw files were processed in MaxQuant (version 2.0.3.1). The LFQ algorithm and match between runs settings were selected. Data were searched against the reviewed UniProt Saccharomyces cerevisiae proteome using the Andromeda search engine embedded in MaxQuant.

Trypsin was set as the digestion enzyme (cleavage at the C-terminal side of lysine and arginine amino acid residues unless proline is present on the carboxyl side of the cleavage site) and a maximum of two missed cleavages were allowed. Cysteine carbamidomethylation was set as a fixed modification, while oxidation of methionine and acetylation of protein N-termini were set as variable modifications. The "match between runs" feature was used with a matching time window of 0.7 min and an alignment time window of 20 min. Label-free quantification was performed using the MaxLFQ feature included in MaxQuant according to default LFQ parameters. The minimum peptide length was set at 7 amino acid units. FDR, determined by searching a reverse sequence database, of 0.01 was used at both the protein and peptide level.

The MaxQuant protein groups output file was imported into Perseus software (version 1.4.0.2) for further statistical analysis and data visualization. Contaminant and reverse protein hits were removed. LFQ intensities were log2 transformed. Missing values (NaN) were imputed from a normal distribution with default values. For each sample, the triplicates were grouped. Data were filtered for at least two out of three replicates LFQ intensity values in at least one group. Protein LFQ intensities were normalized, and missing values were imputed by values simulating noise around the detection limit using the default parameters. A protein was considered significantly differentially expressed when FDR < 0.05.

The mass spectrometry proteomics data have been deposited to the ProteomeXchange Consortium via the PRIDE partner repository with the dataset identifier PXD049212 (Perez-Riverol et al, 2022).

## Microfluidic cell culture

Y04C CellASIC microfluidic devices (http://www.cellasic.com/) were used for cell culture at 25 °C, with 0.6 psi isobaric flow rate. To promote meiosis entry, cells were grown in yeast peptone dextrose (YPD) at 30 °C in an orbital shaker at 295 rpm for 24 h; once $OD_{600}$ reached ~2–3, 50 µl of cells were sonicated for 4–6 s at 3 W, and loaded in the microfluidic device using 1–2 pulses at 20 psi for 5 s. Meiotic induction was triggered by exposing the cells for at least 20 h to sporulation medium (SPO) composed of 0.6% Potassium Acetate, 2% sorbitol, 40 mg/L Adenine, 40 mg/L Uracil, 20 mg/L Histidine, 20 mg/L Leucine, 20 mg/L Tryptophan, 0.02% Raffinose (from a 20% w/v stock solution) and pH adjusted to 8.5 using a 0.25 M sodium carbonate solution. For the return to growth experiments, cells were transferred to a synthetic complete medium (SCD: 1% succinic acid, 0.6% sodium hydroxide, 0.5% ammonium sulfate, 0.17% yeast nitrogen base without amino acids or ammonium sulfate, 0.13% amino acid dropout powder (complete), 2% glucose) after 24 h of microfluidic exposure to SPO at 25 °C maintaining a flow rate of 0.6 psi using the CellASIC™ ONIX Microfluidic pump system to trigger isobaric medium transfer.

## Time-lapse microscopy

Microfluidics experiments were performed using at least two different microscopy setups for reproducibility. Both setups used an automated Zeiss Observer Z1 microscope controlled by ZEN pro software (Zeiss); images were acquired at a 12 min sampling rate using an 40× Zeiss EC Plan-Neofluar 40×1.3 NA oil Ph 3 M27 immersion objective. Image focus was controlled using Definite Focus 2.0 or 3.0. Images were registered using an AxioCam HR Rev

3 camera or an AxioCam 712 monochrome. At least five fields of view were imaged for each biological replicate. Light sources used were COLIBRI 2.0 light-emitting diodes (LED) or an X-CITE XYLIS XT720S lamp (Excelitas Technologies). Sequential non-phototoxic imaging of four minimally cross-talking fluorophores during meiotic development was achieved through tailored dichroic mirrors and bandpass filters using the following fluorophores: Teal-cyan fluorescent protein (mTFP1), green-yellow fluorescent protein (mNeonGreen, Superfolder GFP (sfGFP), mKusabira-Orange κappa fluorescent protein (mKOκ), eqFP611 red fluorescent protein variant (mRuby3), mRFP1.5-derived red fluorescent protein (mCherry) and phase-contrast (Shaner et al, 2013; Ai et al, 2006; Pédelacq et al, 2006; Tsutsui et al, 2008; Bajar et al, 2016; Shaner et al, 2004). Spectral unmixing of strains containing red and orange fluorophores (FK10696 and FK11243) created images where the final mKOκ image was equal to the original mKOκ image minus 0.51 of the red image.

## Image processing and quantification of cellular features

Image analysis code is available at https://github.com/alejandrolvido/ Quiescence-Entry and https://github.com/alejandrolvido/Spectral-Imaging. Images were acquired using a ZEN pro software (Zeiss) with a $2 \times 2$ binning configuration in uncompressed TIFF format. Images were converted to double format and de-noised using the functions *double(), medfilt2()*. Background-correction and cell segmentation was done according to (Arguello-Miranda et al, 2018). A Gaussian fit method was used to define the nuclear area of single cells according to (Arguello-Miranda et al, 2018). Nuclear and cytoplasmic concentrations were obtained using the custom function *Get_Sphere_Vol()*, which labels the objects within a cell mask, computes the equivalent diameter for each object based on its pixel area, and calculates the total nuclear or cytoplasmic volume.

The arbitrary units displayed in the figures for LCI represent the total or average fluorescence signal intensity registered in a fluorescent channel after correction for background, autofluorescence, and bleed-through, as described in the fluorescence extraction code as described previously (Kamat, 2019; Arguello-Miranda et al, 2018; Acuña-Rodriguez et al, 2022).

## Time series and clustering analyses

The presence of the first meiotic nuclear division, which marks the point of no-return in the meiotic program, was used to classify cells as meiosis or no meiosis (Tsuchiya et al, 2014). To obtain MI times, the first derivative of the fluorescence intensity times series of a nuclear marker, such as the histone Htb1-mCherry or pIME1-mRuby3, was calculated and the point with the highest inflection was assessed as the first meiotic division, this assignment was manually confirmed using custom MATLAB code to score the onset of MI nuclear division by directly inspecting fluorescent image time series (Arguello-Miranda et al, 2018). Rim11 peaks were determined as the highest intensity point in the Rim11-mNG or Rim11-mKOκ nuclear fluorescence intensity time series over a window of 60 time points before the time of MI nuclear division. For locating Ume6-mTFP1 and sfGFP-Ime1 peaks, the time

window was restricted to 40 time points before the first meiotic nuclear division.

Single-cell meiotic time series clusters were created by applying the function *kmeans()* to the first 80 time points after starvation onset using correlation distance, 10 replicates, a maximum iteration limit of 10000, and with an experimentally defined number of clusters.

## Statistical analyses

Statistical analyses were performed using GraphPad Prism Version 9.0.0 for Windows, GraphPad Software, www.graphpad.com. One-way or two-way ANOVA tests were carried out, depending on the number of conditions tested. Statistical analyses were conducted in MATLAB, treating each independent microfluidic device as a biological replicate. Kolmogorov–Smirnov tests (K–S test) determined statistical significance set at $P < 0.05$ using the function *kstest2()*. Line plots correspond to the biological replicates' average surrounded by the 95% confidence intervals, represented as shaded areas. Scatter plots depicted R-squared values calculated using *corrcoef()*.

## Mathematical model

We employ a system of ordinary differential equations (ODEs) to model the regulatory network that controls the entry into meiosis. Within this framework, we capture the impact of TORC1 and PKA into three starvation signals, called $S_1$, $S_2$, and $S_3$, that lead to the localization of Rim11 and regulate the Ime1 and Rim11 levels. All three starvation signals are assumed to follow a sigmoidal behavior and thus vary between 0 and 1. To get this we use a simple logistic growth model with delay:

$$\frac{dS_1}{dt} = \theta_1(1 - S_1)S_1, \text{ if } t \geq t_1, \text{ else } \frac{dS_1}{dt} = 0 \text{ and } S_1 = 0.01$$

where $\theta_1$ denotes a growth rate, and $t_1$ denotes any delay in the onset of starvation. Similar equations govern $S_2$ and $S_3$.

Our model strives to recover the main trends and properties of our single-cell time series data. For simplicity, we assume that all Rim11 present in the nucleus (Rim11$_N$) plays an active role in phosphorylation, while the Rim11 in the cytosol (Rim11$_C$) is assumed to be an inactive reservoir. Moreover, we only consider and aim to recover the concentrations of Ime1 and Ume6 in the nucleus. Based on these assumptions, the concentration of the unphosphorylated Ime1 in the nucleus as a function of time ($t$) is governed by the following ODE:

$$\frac{d\text{Ime1}}{dt} = s_1 \cdot S_1 - d_1 \cdot \text{Ime1} - p_1 \frac{\text{Ime1} \cdot \text{Rim11}_N}{c_1 + \text{Ime1}} + u_1 \cdot \text{Ime1}_P + b_1.$$

Here, $s_1$, $d_1$, $p_1$, $c_1$, $u_1$, and $b_1$ denote model parameters: $s_1$ is connected to starvation, $d_1$ is a decay rate, $p_1$ and $c_1$ are associated with phosphorylation, $u_1$ is associated with dephosphorylation of Ime1 and $b_1$ is a base transcription rate. The amount of unphosphorylated Ime1 is thus dictated by $S_1$ as well and its phosphorylated counterpart (Ime1$_P$) and thus by Rim11$_N$. Meanwhile, Ime1$_P$ can combine with Ume6 to form a complex (C). As we

observe some adaptation to the stress in the level of Ime1, we introduce a decay term for $\text{Ime1}_P$ that depends on $S_1$ at an earlier time (that is at $t\text{-}k_1\tau_1$), this is a simple way of modeling an incoherent feedforward loop that is known to produce an adaptive response:

$$\frac{d\text{Ime1}_P}{dt} = p_1 \frac{\text{Ime1} \cdot \text{Rim11}_N}{c_1 + \text{Ime1}} - u_1 \cdot \text{Ime1}_P - a_1 \cdot \text{Ime1}_P \cdot \text{Ume6}$$
$$+ a_2 \cdot C - d_2 \cdot \text{Ime1}_P \cdot S_1(t - k_1\tau_1)$$

Here, $\tau_1$ denotes the time at which Ime1 peaks for the mean single-cell time series, while $a_1$, $a_2$, $d_1$, and $k_1$ are model parameters: $a_1$ and $a_2$ denote rates that describe the formation of the complex, and $d_1$ is a decay rate.

For Rim11, $S_2$ controls the concentration in the cytosol (modeling transcriptional regulation), while $S_3$ governs the localization to the nucleus:

$$\frac{d\text{Rim11}_C}{dt} = s_2 \cdot S_2 - l_{cn} \cdot S_3 \cdot \text{Rim11}_C + l_{nc} \cdot \text{Rim11}_N$$
$$- d_3 \cdot \text{Rim11}_C + b_2,$$
$$\frac{d\text{Rim11}_N}{dt} = l_{cn} \cdot S_3 \cdot \text{Rim11}_C$$
$$- l_{nc} \cdot \text{Rim11}_N + l_b \cdot \text{Rim11}_C$$
$$- d_4 \cdot \text{Rim11}_N \cdot S_3(t - k_2\tau_2).$$

Here, $l_{cn}$, $l_{nc}$ and $l_b$ are model parameters that are associated with localization. Both, d3 and d4 are decay rates, while b2 is a basal transcription rate. Again, we have introduced a decay term that depends on the starvation signal ($S_3$) at an earlier time, at $t\text{-}k_2\tau_2$, based on the time ($\tau_2$) at which $\text{Rim11}_N$ peaks in the mean single-cell time series, which is to produce an adaptive response to stress as observed in the data.

We have found that Ime1 acts as a scaffold for Ume6 phosphorylation, so we assume that the complex C can get phosphorylated by $\text{Rim11}_N$ on Ume6 producing a new complex ($C_P$) of phosphorylated Ime1 and phosphorylated Ume6. The concentration of the complex C formed by $\text{Ime1}_P$ and Ume6 is given by

$$\frac{dC}{dt} = a_1 \cdot \text{Ime1}_P \cdot \text{Ume6} - a_2 \cdot C - p_2 \cdot \frac{\text{Rim11}_N \cdot C}{c_2 + C} + u_2 * C_P$$

where $p_2$ and $c_2$ are model parameters that are associated with the phosphorylation and $u_2$ is a model parameter associated with dephosphorylation of the Ume6 in the complex. The concentration of this new complex is given by

$$\frac{dC_P}{dt} = p_2 \cdot \frac{\text{Rim11}_N \cdot C}{c_2 + C} - u_2 \cdot C_P.$$

Finally, it is known that $C_P$ regulates transcription of the Ume6 promoter (in addition to the other early miotic genes, such as Ime2). So, the concentration of Ume6 is dictated by the formation of the complex C and the concentration of the phosphorylated

complex through a positive feedback loop (based on the model parameters $p_3$, $c_3$, and $q$):

$$\frac{d\text{Ume6}}{dt} = p_3 \cdot \frac{C_P^q}{c_3^q + C_P^q} - d_5 \cdot \text{Ume6} + b_3$$
$$- a_1 \cdot \text{Ime1}_P \cdot \text{Ume6} + a_2 \cdot C$$

where $b_3$ is a base transcription rate, and $d_5$ is a decay rate.

To recover the mean single-cell time series qualitatively, we tuned the parameters using a combination of manual adjustment and fitting. We set $k_1$ to 0.85, $k_2$ to 0.55, $d_1$ to 0.22 per hour, $d_2$ to 3.0 per hour, $d_3$ to 18 per hour, $d_4$ to 0.53 per hour, $d_5$ to 3.0 per hour, $p_1$ to 0.6 per hour, $p_2$ to 0.4 per hour, $p_3$ to 150 per hour, $c_1$ to 80, $c_2$ to 35, $c_3$ to 25, $s_1$ to 24 per hour, $s_2$ to 70 per hour, $a_1$ to 1.5 per hour, $l_{cn}$ to 0.63 per hour, $l_{nc}$ to 0.15 per hour and $\theta_1$ and the corresponding rates for $S_2$ and $S_3$ to 2 per hour. The remaining seven parameters ($a_2$, $b_1$, $b_2$, $b_3$, $l_b$, $u_1$ and $u_2$) were fixed by requiring a steady state when no increased starvation signal was present, i.e., when $S_1$, $S_2$ or $S_3$ are 0.01.

## Data availability

The mass spectrometry proteomics data have been deposited to the ProteomeXchange Consortium via the PRIDE partner repository with the dataset identifier PXD049212.

The source data of this paper are collected in the following database record: biostudies:S-SCDT-10_1038-S44318-024-00149-7.

## Peer review information

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

## Acknowledgements

The authors thank Frank Uhlmann and Katrin Rittinger for reading of the manuscript. The authors also thank the four reviewers for their comments and suggestions. This work was supported by the Francis Crick Institute (CC2043), which receives its core funding from Cancer Research UK (CC2043), the UK Medical Research Council (CC2043), and the Wellcome Trust (CC2043). OAM and NT are supported by grant R00GM135487 from the National Institute of General Medical Sciences of the National Institutes of Health, USA. ACSJ is supported by the Eric and Wendy Schmidt AI in Science Postdoctoral Fellowship, a Schmidt Futures program.

## Author contributions

**Johanna Kociemba**: Conceptualization; Data curation; Formal analysis; Validation; Investigation; Methodology; Writing—original draft; Writing—review and editing. **Andreas Christ Sølvsten Jørgensen**: Formal analysis; Investigation; Visualization; Methodology; Writing—review and editing. **Nika Tadic**: Data curation; Formal analysis; Investigation; Visualization; Methodology; Writing—review and editing. **Anthony Harris**: Data curation; Formal analysis; Validation; Investigation; Visualization; Methodology; Writing—review and editing. **Theodora Sideri**: Data curation; Formal analysis; Validation; Investigation; Methodology; Writing—review and editing. **Wei Yee Chan**: Data curation; Formal analysis; Investigation; Methodology; Writing—review and editing. **Fairouz Ibrahim**: Data curation; Formal analysis; Investigation; Methodology; Writing—review and editing. **Elcin Ünal**: Resources; Supervision; Funding acquisition; Investigation; Methodology; Writing—review and editing. **Mark Skehel**: Data curation; Formal analysis; Supervision; Visualization; Writing—review and editing. **Vahid Shahrezaei**: Conceptualization; Resources; Formal analysis; Supervision; Funding acquisition; Investigation; Methodology; Writing—original draft; Writing—review and editing. **Orlando Arguello-Miranda**: Data curation; Formal analysis; Supervision; Funding acquisition; Investigation; Visualization; Methodology; Writing—original draft; Writing—review and editing. **Folkert Jacobus van Werven**: Conceptualization; Resources; Data curation; Formal analysis; Supervision; Funding acquisition; Investigation; Writing—original draft; Writing—review and editing.

Source data underlying figure panels in this paper may have individual authorship assigned. Where available, figure panel/source data authorship is listed in the following database record: biostudies:S-SCDT-10_1038-S44318-024-00149-7.

## Funding

## Disclosure and competing interests statement

The authors declare no competing interests.

# Expanded View Figures

**Figure EV1.  Rim11 expression and localization dynamics prior and during meiosis.**

(**A**) Onset of meiosis in WT (7794) and cells harboring *RIM11* tagged with mNeonGreen (*RIM11*-mNG) (FW10297). Cells were induced to sporulate, and samples were taken at the indicated time points, fixed, and stained with DAPI. Cells that contained two more DAPI masses were considered to have entered meiosis. The error bars represent the mean $+$ SEM of $n=3$, and least $n=100$ cells per biological repeat were quantified. (**B**) Nuclear concentrations of Rim11 in exponential growth (exp.) and in cells induced to enter meiosis in SPO and expressing *RIM11*-mNG $+$ *HTB1*-mCh (FW10297). At least $n=50$ cells per condition were quantified. The error bars represent the mean$+$SD. (**C**) *RIM11* expression determined by RT-qPCR in WT cells (FW1511) grown until the exponential phase (exp.) and cells induced to enter meiosis. Samples were normalized to the time point 0 h. $n=2$ biological repeats were performed. The *RIM11* signals were normalized over *ACT1*. (**D**) Quantification of whole-cell concentrations of Rim11-mNG using live-cell imaging setup using the strain described in (**A**). Indicated are the mean traces of cells that entered meiosis (meiosis) and cells that did not enter meiosis (no meiosis). Shown are the Rim11-mNG cytoplasmic concentrations.

▶

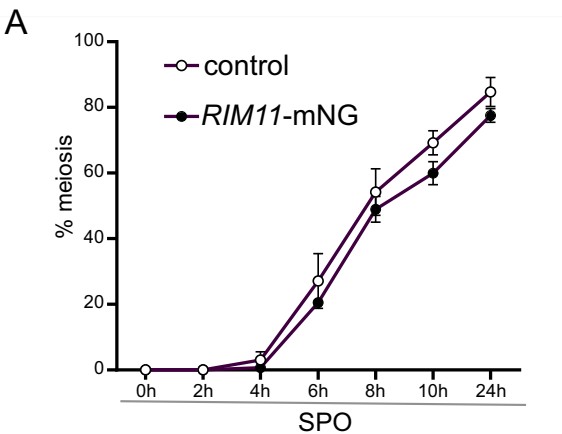

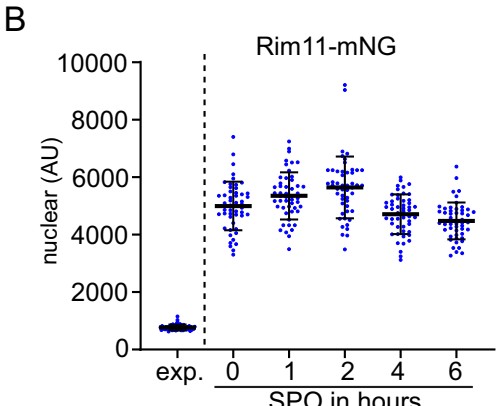

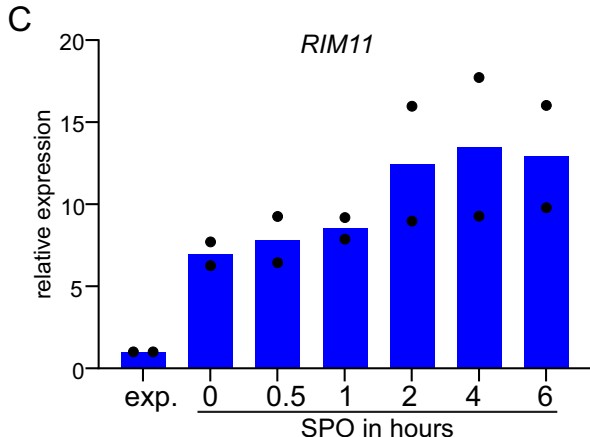

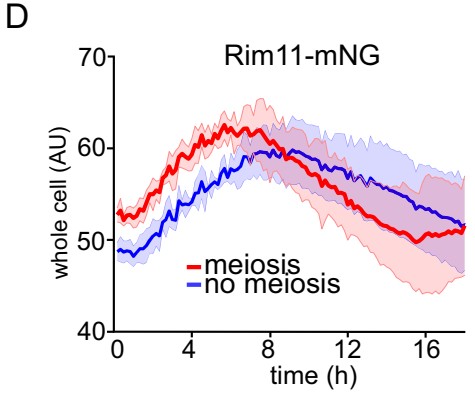

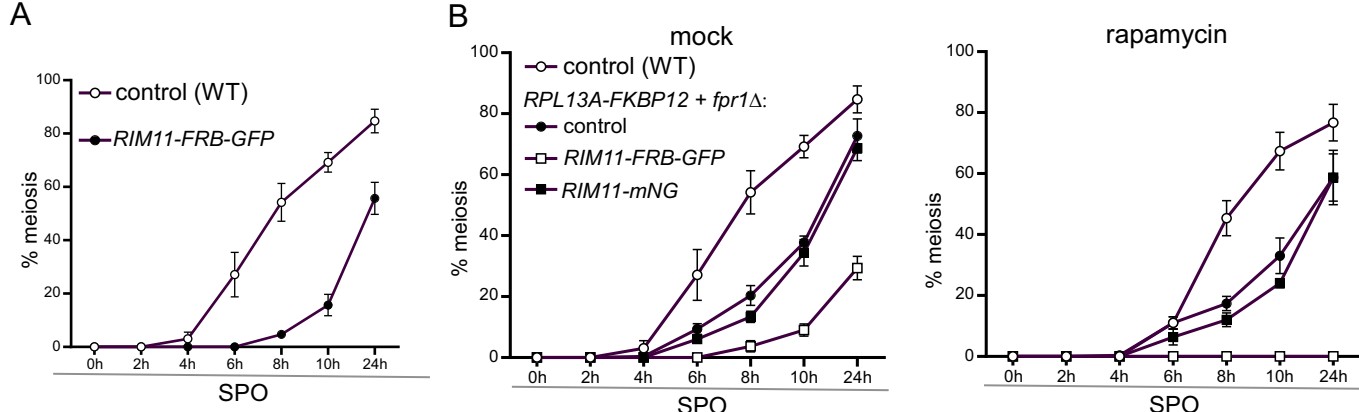

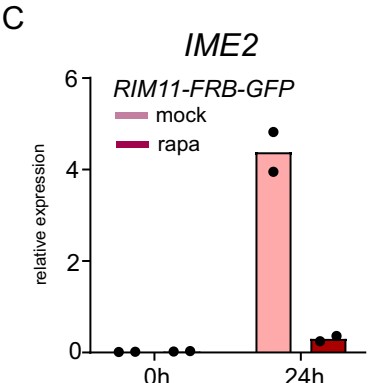

**Figure EV2.  Rim11 nuclear localization is required for meiosis.**

(**A**) Onset of meiosis in WT (7794) and *RIM11-FRB-GFP* (FW11208). Cells were induced to sporulate, and samples were taken at the indicated time points, fixed, and stained with DAPI. Cells that contained two more DAPI masses were considered to have entered meiosis. The error bars represent the mean + SEM of *n* = 3, and at least *n* = 100 cells per biological repeat were quantified. (**B**) Onset of meiosis in WT cells (FW1511), and in cells harboring *RPL13A-FKBP12* and *fpr1Δ* (FW11257) by itself or together with *RIM11-FRB-GFP* (FW11124) or *RIM11-mNG* (FW11126). Cells were mock-treated or untreated with rapamycin at 0 h in SPO. The mean and SEM of 3 independent repeats is shown. (**C**) *IME2* expression was determined in *RIM11-FRB-GFP* (FW11124) after 24 h in SPO in mock and rapamycin-treated cells.

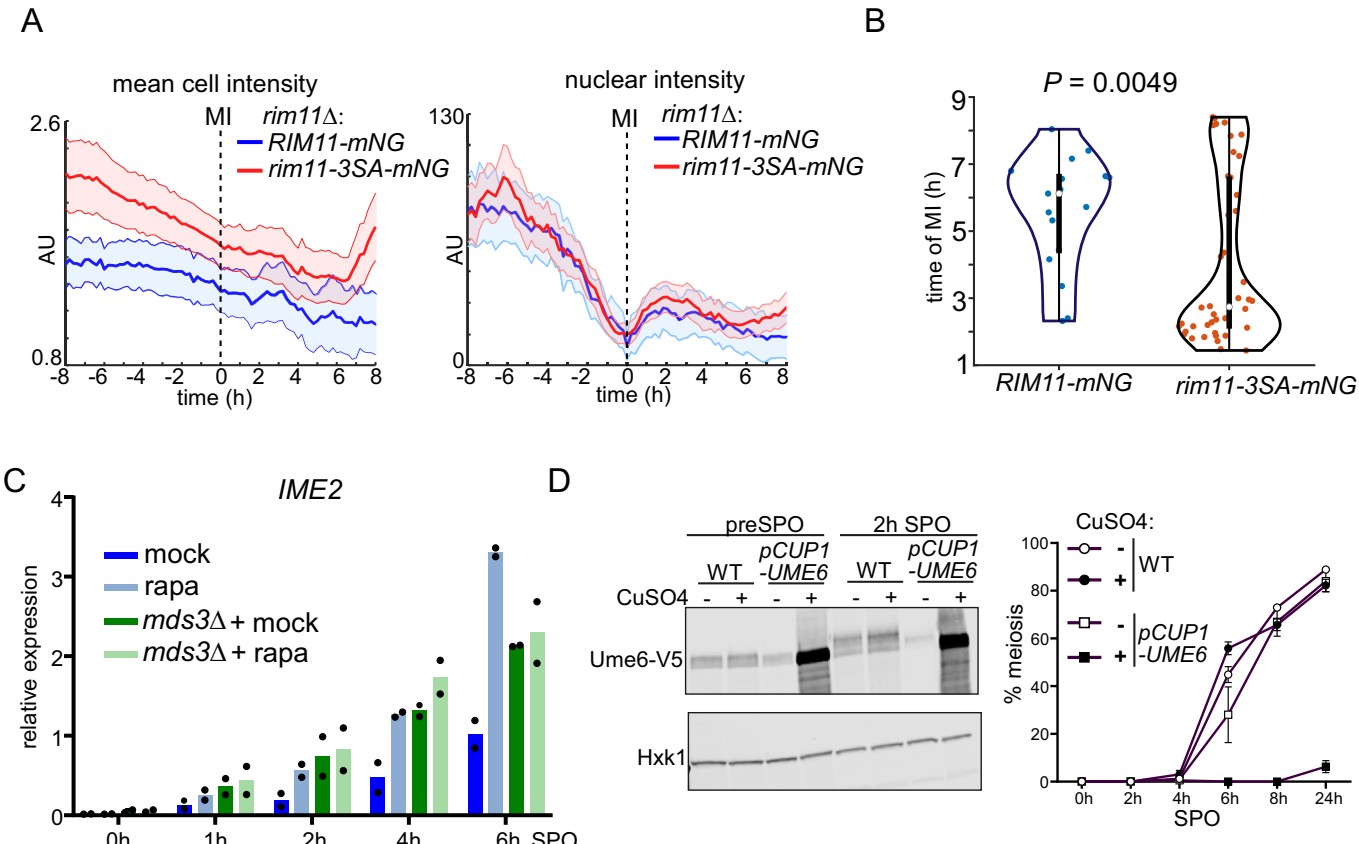

**Figure EV3.   PKA and TORC1 control Rim11 via distinct mechanisms.**

(A) Quantification of live-cell imaging of Rim11-mNG and rim11-3SA-mNG. Shown are whole-cell concentration (left) and nuclear intensity (right). Cells harboring *rim11Δ* expressing a single-copy integration plasmid harboring *RIM11-mNG* and *rim11-3SA-mNG* (FW10776 and FW10778). Time points were aligned according to the MI division. At least *n* = 500 cells were quantified. The bold colored line represents the mean, and the 95% intervals are highlighted by the thin colored lines. (B) The same experiment as in A but showing the timing of meiosis in Rim11-mNG and rim11-3SA-mNG. *N* = 17 (Rim11-mNG) and *n* = 52 (rim11-3SA-mNG) cells were quantified. The minima and maxima represent the whole distribution range, the center dot the median of the distribution, the bounds of the box represents limits of the 1.5 interquartile range, whiskers represent the whole distribution leaving out the outliers using a 1.5 interquartile outlier criterion. The area around the whisker plot shows the shape of the distribution represented as the kernel density. (C) *IME2* expression in cells induced to enter meiosis in WT and *mds3Δ* that were either untreated or treated with rapamycin (FW10297 and FW10718). *IME2* expression signals were normalized over *ACT1*. The mean signals of n = 2 biological repeats are shown. (D) Expression and onset of meiosis in *UME6-V5* (FW1208) and *pCUP1-UME6-V5* (FW10562). Cells were grown till the presporulation medium, treated with CuSO4 or untreated, shifted to SPO. Samples were taken at the indicated time points for western blot and DAPI staining. Membranes were probed with anti-V5 antibodies and Hxk1 antibodies as a loading control (left). Cells that contained two more DAPI masses were considered to have entered meiosis. The error bars represent the mean + SEM of *n* = 3, and at least *n* = 100 cells per biological repeat were quantified (right).

## A

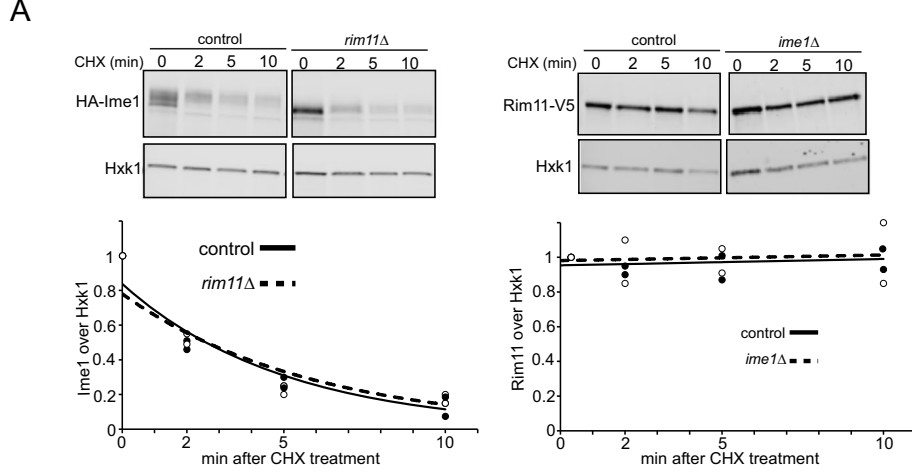

## B

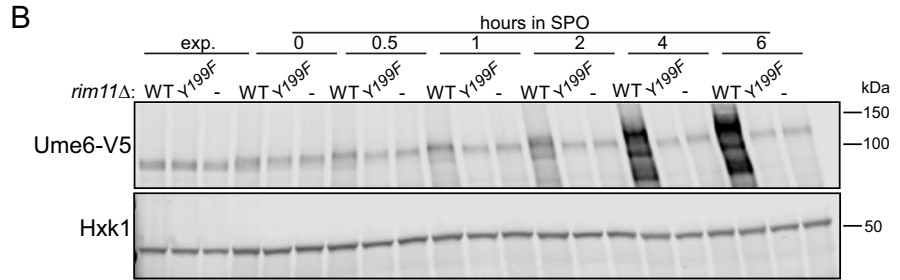

## C

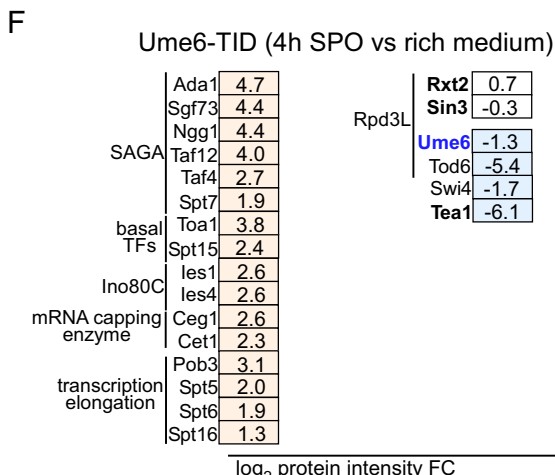

## D

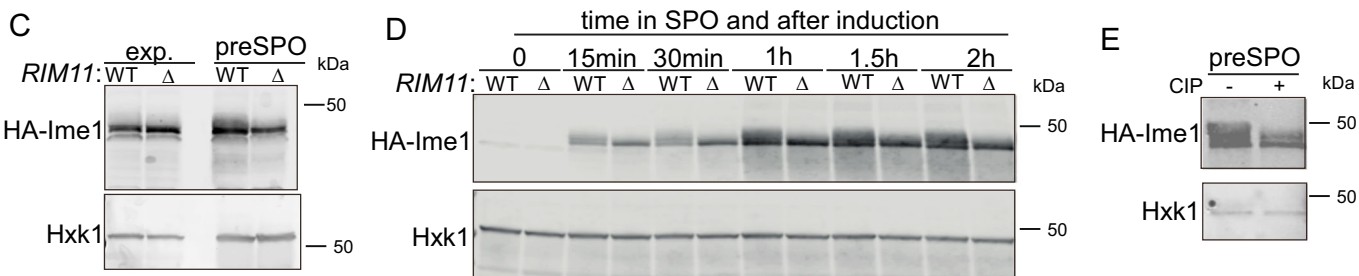

## E

## F

Ume6-TID (4h SPO vs rich medium)

| | | | | | |
|---|---|---|---|---|---|
| SAGA | Ada1 | 4.7 | Rpd3L | Rxt2 | 0.7 |
| | Sgf73 | 4.4 | | Sin3 | -0.3 |
| | Ngg1 | 4.4 | | Ume6 | -1.3 |
| | Taf12 | 4.0 | | Tod6 | -5.4 |
| | Taf4 | 2.7 | | Swi4 | -1.7 |
| | Spt7 | 1.9 | | Tea1 | -6.1 |
| basal TFs | Toa1 | 3.8 | | | |
| | Spt15 | 2.4 | | | |
| Ino80C | Ies1 | 2.6 | | | |
| | Ies4 | 2.6 | | | |
| mRNA capping enzyme | Ceg1 | 2.6 | | | |
| | Cet1 | 2.3 | | | |
| transcription elongation | Pob3 | 3.1 | | | |
| | Spt5 | 2.0 | | | |
| | Spt6 | 1.9 | | | |
| | Spt16 | 1.3 | | | |

log$_2$ protein intensity FC

◀  **Figure EV4.  Ime1 is required for Rim11-directed Ume6 phosphorylation.**

(**A**) Relative protein stability of Ime1 and Rim11 in control and in *rim11Δ* or *ime1Δ* cells, respectively. *pCUP-HA-IME1* (control and *rim11Δ*, FW2444 and FW10373) or *RIM11-V5* (control and *ime1Δ*, FW10446 and FW11682) cells were induced to enter meiosis. At 2 h in SPO Ime1 expression was induced with CuSO4. At 4 h cells were treated with cycloheximide (CHX). Samples were taken at the indicated time points for western blotting. Membranes were probed with anti-HA or anti-V5 antibodies. As a loading control Hxk1 was used. Relative quantification with respect Hxk1 and the 0-hour time point is shown for $n = 2$ biological repeats. (**B**) Ume6-V5 expression and migration as determined by western blotting in *rim11Δ* expressing an integration plasmid harboring *RIM11-mNG* or *rim11-Y199F-mNG* cells induced to enter meiosis (FW11186, FW11184, FW10033). Membranes were probes with anti-V5 antibodies and Hxk1 antibodies. (**C–E**) Ime1 expression and migration in exponential growth, and in cells induced to enter meiosis. Ime1 tagged with HA was expressed from the *CUP1* promoter and induced in WT and *rim11Δ* cells (FW2444, FW10373). Samples were taken at the indicated time points, and probed with anti-HA antibodies, and Hxk1 antibodies as a control. In E protein extracts from samples grown in presporulation medium were treated with alkaline phosphatase (CIP). (**F**) Proximity labeling using Ume6-TID comparing 4 h SPO to rich medium conditions. Log2 protein intensity fold change (FC) comparing 4 h SPO to rich medium. Proteins involved in transcription or that are known to interact with Ume6 are shown.

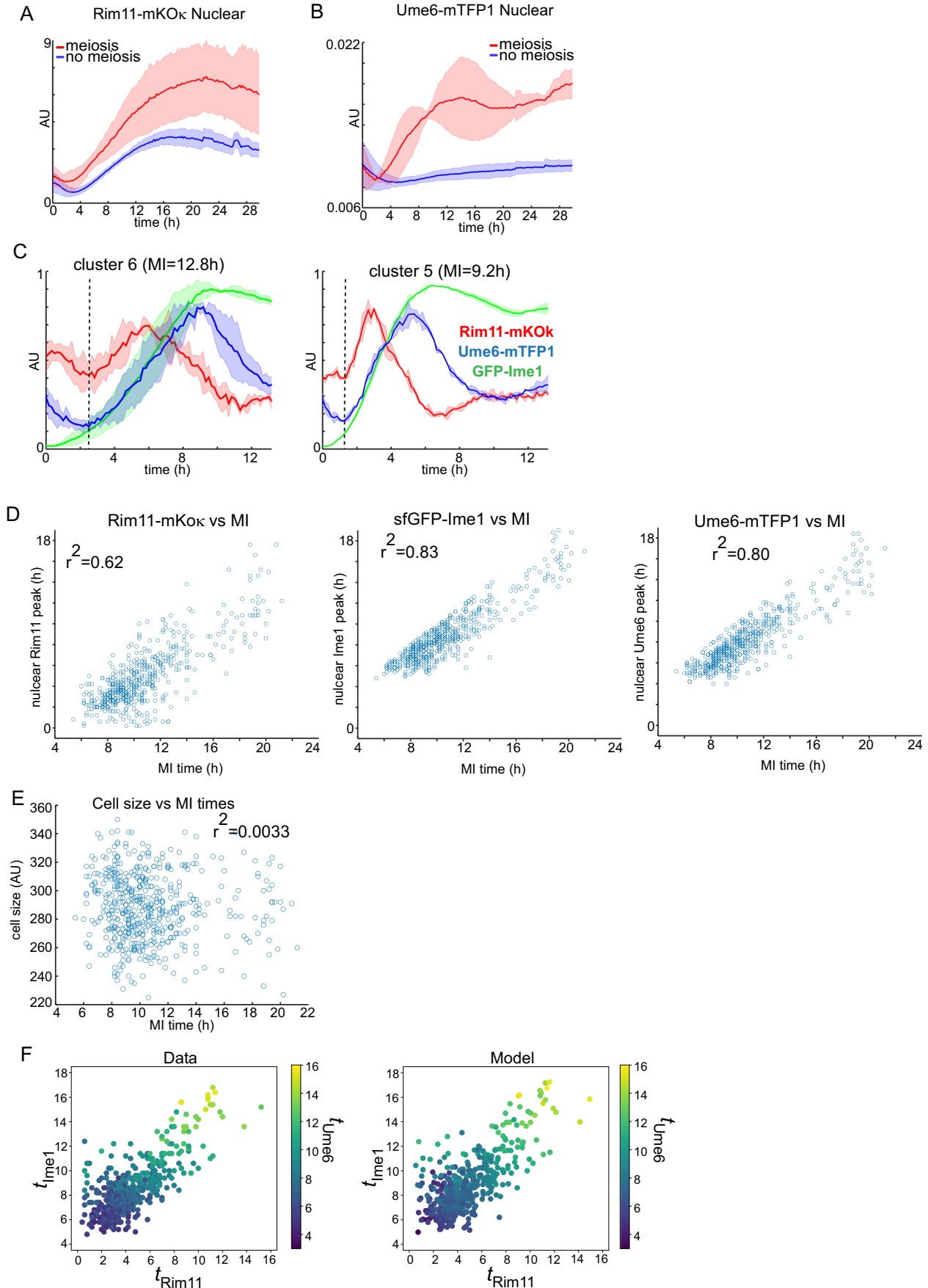

◀  **Figure EV5.   Single-cell analysis and modeling reveals the timing dynamics Rim11, Ime1, Ume6 and meiosis.**

Live-cell imaging of the dynamics of Rim11-mKOκ, sfGFP-Ime1, Ume6-mTFP in meiotic and non-meiotic cells (FW11243). (**A**) Nuclear concentrations of Rim1-mKOκ in meiotic and non-meiotic cells. (**B**) Nuclear concentrations of Rim11-mTFP in meiotic and non-meiotic cells. (**C**) Mean traces of Rim11-mKOκ, sfGFP-Ime1, and Ume6-mTFP nuclear intensity for clusters 6 (left), and 5 (right) described in Fig. 7D. (**D**) Scatter plot showing the single-cell data comparing the timing of Rim11-mKOκ peak versus MI (left), sfGFP-Ime1 peak versus MI (middle), Ume6-mTFP peak versus MI (right). (**E**) Scatter plot showing the single-cell data comparing the timing of cell size versus MI. (**F**) Time of Ume6-mTFP peak as a function of sfGFP-Ime1 and Rim11-mKOκ peak timings for the cells that enter meiosis. The left panel shows the results from the time series data, while the right panel illustrates our model predictions. Besides the peak time in Ime1 and Rim11, the model reproduces the amplitudes of the peaks in Ime1 and Rim11 as well as the final value of Rim11 in the single-cell data. The plot contains 461 out of 524 cells—of the remaining 63 cells, most were excluded because the maximal values in Ime1 and Rim11 of the model lie outside the first 18 h after $t = 0$.

