## [Peer Review File · The EMBO Journal]

Multi-signal regulation of the GSK-3 β homolog Rim11 controls meiosis entry in budding yeast

Folkert van Werven, Johanna Kociemba, Andreas Jørgensen, Nika Tadic, Anthony Harris, Theodora Sideri, Wei Chan, Fairouz Ibrahim, Elcin Ünal, Mark Skehel, Vahid Shahrezaei, and Orlando Arguello-Miranda

Corresponding author(s): Folkert van Werven (folkert.vanwerven@crick.ac.uk) , Vahid Shahrezaei (v.shahrezaei@imperial.ac.uk), Orlando Arguello-Miranda (oargell@ncsu.edu)

Review Timeline:

Submission Date:	28th Sep 23
Editorial Decision:	9th Nov 23
Revision Received:	6th Feb 24
Editorial Decision:	21st Feb 24
Revision Received:	14th Mar 24
Editorial Decision:	15th Apr 24
Revision Received:	22nd Apr 24
Accepted:	27th May 24

Editor: Ioannis Papaioannou

Transaction Report:

Dear Dr. van Werven,

Thank you for submitting your manuscript for consideration by the EMBO Journal. It has been seen by four experts in the field, and we have received the full set of their reports, which I have already shared with you (they are included again below). I would also like to thank you for your point-by-point response to their comments and your provisional revision plan, according to which you are able and willing to address their concerns in a reasonable amount of time.

Given the referees' largely positive comments and recommendations, I would like to invite you to submit a revised version of your manuscript, addressing the comments of all four reviewers. I should add that it is EMBO Journal policy to allow only a single round of major revision, and acceptance of your manuscript will therefore depend on the completeness of your responses in this revised version. If you have any questions or comments, we can also discuss the revisions in a video chat, if you like.

We generally allow three months as standard revision time (8th February 2024). As a matter of policy, competing manuscripts published during this period will not negatively impact our assessment of the conceptual advance presented by your study. However, we request that you contact us as soon as possible upon publication of any related work, to discuss how to proceed. Should you foresee a problem in meeting this three-month deadline, please let us know in advance and we may be able to grant an extension.

Thank you for the opportunity to consider your work for publication in the EMBO Journal. I look forward to your revision.

Yours sincerely,

Instructions for preparing your revised manuscript

1. When you are ready to submit the revision, please upload:

- A Word file of the manuscript text (including legends of main Figures, EV Figures and Tables). Please make sure that changes are highlighted (or "tracked") to be clearly visible.

- Individual production-quality figure files (one file per figure). When assembling your figures, please refer to our figure preparation guidelines in order to ensure proper formatting and readability in print as well as on screen:

If the data shown in a figure are obtained from n {less than or equal to} 2, please use scatter plots showing the individual data points.

i. the name of the statistical test used to generate error bars and P values

ii. the number (n) of independent experiments (please specify technical or biological replicates) underlying each data point (discussion of statistical methodology can be reported in the Materials and Methods section, but figure legends should contain a basic description of n , P , and the test applied)

iii. the nature of the bars and error bars (s.d., s.e.m.).

- A point-by-point response to the referees' comments, with a detailed description of the changes made (as a word file). All referees' concerns must be fully addressed and their suggestions taken on board. When preparing your letter of response to the referees' comments, please bear in mind that this will form part of the Review Process File and will therefore be available online to the community. Please note that you have the possibility to opt out of the transparent process at any stage prior to publication by letting the editorial office know (contact@embojournal.org); if you do opt out, the Review Process File link will point to the following statement: "No Review Process File is available with this article, as the authors have chosen not to make the review process public in this case.". For more details on our Transparent Editorial Process, please visit our website: <https://www.embopress.org/page/journal/14602075/authorguide#transparentprocess>

- Expanded View (EV) files (replacing Supplementary Information) that are collapsible/expandable online. A maximum of 5 EV Figures can be typeset. EV Figures should be cited as "Figure EV1, Figure EV2" etc. in the text, and their respective legends should be included in the manuscript file after the legends of regular figures. See detailed instructions regarding Expanded View

files here:

- For the figures that you do NOT wish to display as Expanded View figures, they should be bundled together with their legends in a single PDF file called "Appendix", which should start with a short Table of Contents (including page numbers). Appendix figures should be referred to in the main text as: "Appendix Figure S1, Appendix Figure S2" etc. Please see detailed instructions here: <https://www.embopress.org/page/journal/14602075/authorguide#expandedview>

- A complete author checklist, which you can download from our author guidelines (<https://www.embopress.org/page/journal/14602075/authorguide>). Please note that the checklist will also be part of the Review Process File.

2. Please note that no statistics should be calculated if $n=2$.

3. Before submitting your revision, primary datasets (and computer code, where appropriate) produced in this study need to be deposited in appropriate public databases (see <https://www.embopress.org/page/journal/14602075/authorguide#dataavailability>). Specifically, we would kindly ask you to provide public access to the following datasets/data:

- Mass spectrometry data.

The accession numbers and database should be listed in a formal "Data availability" section (placed after Materials and Methods) that follows the model below (see also <https://www.embopress.org/page/journal/14602075/authorguide#dataavailability>):

Data availability

- RNA-seq data: Gene Expression Omnibus GSE46843 (<https://www.ncbi.nlm.nih.gov/geo/query/acc.cgi?acc=GSE46843>)
- [data type]: [name of the resource] [accession number/identifier/doi] ([URL or identifiers.org/DATABASE:ACCESSION])

*** Note: all links should resolve to a page where the data can be accessed. ***

*** Note: the Data Availability Section is restricted to new primary data that are part of this study. ***

4. Please check that the title and the abstract of the manuscript are brief, yet explicit, even to non-specialists. The length of the title should not exceed 100 characters (including spaces), and the abstract should be a single paragraph not exceeding 175 words.

5. Please also note our reference format: <https://www.embopress.org/page/journal/14602075/authorguide#referencesformat>.

7. Please remember: digital image enhancement is acceptable practice, as long as it accurately represents the original data and conforms to community standards. If a figure has been subjected to significant electronic manipulation, this must be noted in the figure legend or in the "Materials and Methods" section. The editors reserve the right to request original versions of figures and the original images that were used to assemble the figure.

8. Our journal encourages inclusion of data citations in the reference list to directly cite datasets that were obtained from public databases. Data citations in the article text are distinct from normal bibliographical citations and should directly link to the database records from which the data can be accessed. In the main text, data citations are formatted as follows: "Data ref: Smith et al, 2001" or "Data ref: NCBI Sequence Read Archive PRJNA342805, 2017". In the Reference list, data citations must be labeled with "[DATASET]". A data reference must provide the database name, accession number/identifiers, and a resolvable link to the landing page from which the data can be accessed at the end of the reference. Further instructions are available at: <https://www.embopress.org/page/journal/14602075/authorguide#referencesformat>.

9. We request authors to consider both actual and perceived competing interests. Please review our policy (<https://www.embopress.org/page/journal/14602075/authorguide#conflictsofinterest>) and update your competing interests statement if necessary. Please name this section 'Disclosure and competing interests statement' and place it after the Acknowledgements section.

10. Please note that all corresponding authors are required to provide an ORCID ID upon submission of a revised manuscript (<https://orcid.org/>). Please find instructions on how to link your ORCID ID to your account in our manuscript tracking system in our Author guidelines (<https://www.embopress.org/page/journal/14602075/authorguide#authorshipguidelines>).

11. We use CRediT to specify the contributions of each author in the journal submission system. CRediT replaces the author contribution section, which should be removed from the manuscript. Please use the free text box to provide more detailed descriptions. See also guide to authors: <https://www.embopress.org/page/journal/14602075/authorguide#authorshipguidelines>.

13. We would also welcome the submission of cover suggestions or motifs to be used by our Graphics Illustrator in designing a cover.

14. Please use the link below to submit your revision:
<https://emboj.msubmit.net/cgi-bin/main.plex>

Referee #1:

The paper by Kociemba et al uses a variety of orthogonal approaches to convincingly show that the Rim11 kinase acts to integrate a variety of signals to enable Ime1-mediated transcription of early genes. They show that the starvation conditions required to induce meiosis result in increased Rim11 levels and nuclear localization of the kinase and that nuclear localization of Rim11 is required to induction of early genes like IME2. Abundance and nuclear localization are independently regulated by PKA and TORC1, respectively. In addition they demonstrate that that Ime1 promotes Rim11 localization to the nucleus, is phosphorylated by Rim11 prior to Ume6 phosphorylation and that interaction between Rim11 and Ime1 is required for Ume6 phosphorylation. They incorporated their findings into a mathematical model that illustrates how Rim11 affects the timing and amount of Ume6 phosphorylation. This is a very thorough analysis of a complicated problem which elucidates how different signals can be used to regulate transcription of a set of genes that dramatically alters cell fate (going from vegetative growth to meiosis).

While the science in this manuscript is excellent, the document was very difficult to read because of numerous typos, grammatical errors, incorrect use of genetic nomenclature, overuse of jargon, and some unexplained inconsistencies in their results. I gave detailed comments for the Figure 1 legend, but will leave it to the authors to carefully read the remaining legends and correct them.

Major comments:

1. This work directly contradicts the findings published in Molecular Cell by the Strich lab where they showed that Ume6 was degraded during meiosis by Cdc20/APC. This contradiction should be mentioned and if the authors have a potential explanation for the different results that would be helpful.
2. Page 8 bottom, The sentence "...while a large fraction of untreated cells...completed meiosis" is misleading". According to the figure legend the plotted values include binucleate cells which have only completed the first meiotic division. Only tetranucleate cells should be counted as completing meiosis. The same comment is true for Figure S2B, 4D, S4D. Instead of saying that "two or more DAPI masses were considered to have undergone meiosis" it would be more accurate to say "...to have entered meiosis". If the data are broken down to graph binucleates (MI) separately from tetranucleates (MII), are there any differences?
3. I'm confused by the data presented in Figure 2F and S2F. There is no Ume6 phosphorylation at 24 hrs in the Rim11-AA - rap strain in Figure 2F, yet cells from this strain enter Meiosis I at 10 hours (Figure S2B). Could the authors please explain this discrepancy?
4. In Figure 1A, Ume6 phosphorylation increases between 1 and 2 hours after transfer to Spo medium. In Figure 4F, Ume6 is highly phosphorylated within 50 minutes after transfer to Spo medium. Please explain this discrepancy.
5. In Figure 4C, IME2 expression is elevated at 0.5 hours in mds3Δ compared to WT and stays elevated at 1 hour (at ~2 arbitrary units). In Figures S4H, the 0.5 hr timepoint wasn't taken, even though that is where the difference in kinetics was previously observed. The relative expression of IME2 in mds3Δ + mock is much lower at the 1 hour timepoint than was observed for mds3Δ in Figure 4C. Please explain the discrepancy.

Minor comments

Title page, affiliation 3 writes out United Kingdom while #5 uses UK; Similarly #1 using United States and #4 uses USA-be consistent.

Page 4, middle, "The kinase Rim11 and DSBk-3b homolog also play a critical role..."

Throughout add a space between the citations at the end of a sentence.

Define acronyms the first time they are used, for example PKA and TORC1 (this definition is given on page 10 but should be given the first time TORC1 is used).

Page 5, line 6-7: During starvation Rim11 expression is increased and the protein is localized to the nucleus.

Page 6 near bottom: ...we also expressed histone H2B fused to mCherry (not sure why "signal" is present)

Page 8, line 7, ...increased nuclear concentrations of Rim11 not only correlate....

Figure S2B, The mNG tagged allele appears to be hypomorphic as it is greatly delayed in meiotic progression compared to WT. The authors should mention this. Also there is a typo in the figure legend for this panel: Rim11-mNG, not nNG.

Page 9, line 8, ...on IME2 expression in cells....

Page 10, line 2, ...by fusing a nuclear...

Page 10, line 7,forcing Rim11 to the nucleus..

Page 10, middle, ...to either SPO or SPO plus glucose.

Page 10, bottom, can be written either "...showed increased Rim11 localization to the nucleus.." or "...showed an increase in Rim11 localization to the nucleus"..

Page 11, line 1, ...and a more rapid increase in Ume6....

Page 11, middle, 1-NM-PP1 is not an analog of ATP as it lacks the sugar and phosphate groups of ATP. It is an analog of the PP1 inhibitor which resembles a derivatized adenine.

Page 12 top, use the correct allele designation for the analog sensitive kinase: PKA^{AS} is not correct nomenclature.

Figures S4A-please provide p values for those differences that are significant.

Page 12, bottom, ...Rim11 depletion was much faster in the nucleus than the cytoplasm....

Page 13, top. A reference is needed for the statement that PKA phosphorylates specific residues on Rim11 and the fact that the RIM11^{S3A} mutant exhibits increased Rim11 kinase activity in glucose.

Page 13 bottom, We conclude that...of Rim11 likely destabilizes...and has little effect...(keep the verb tenses the same).

Heading: Mds3 promotes Rim11 localization via TORC1

Why is the synergistic effect of combining inactivation of PKA with inhibition of TORC not seen if Figure S3B?

Page 14 middle, ...we examined IME2 expression and meiotic progression...

The timepoint shown in Figure 4E should be indicated on the figure and in the figure legend.

Page 15 top, Please indicate where the data are that show that mds3Δ cells still display Rim11 nuclear localization changes upon starvation...

Page 15 bottom, ...the transcription activation domain function of Ime1 that drives...

Page 16 top, ...interaction with Ime1 contribute to the Rim11 localization pattern....

Page 16 top.. mutated tyrosine 199 to phenylalanine (T199F). Since this mutation impairs Rim1 kinase activity, it is likely a recessive mutant. The proper way to write the genotype is rim11-Y199F (italicized). All capital letters (RIM11^{Y199F}) indicates the allele is dominant. If that is the case, then please explicitly state this.

The nomenclature should be corrected in Figure 5 and S5 as well.

Page 16 bottom, In *Saccharomyces cerevisiae* nomenclature, wild-type alleles are indicated by all capital letters with no allele designation, so RIM11 not RIM11^{WT}.

Page 17: *ime1-L321F* (italicized), not IME1^{L321F}. A reference is needed for the fact that this mutation impairs the interaction between Ime1 and Rim11.

Page 17, end of middle paragraph, This surprising result suggests....

Page 18, mass spectrometry (MS)

Page 18 middle, Conversely, the Tod6 subunit that is known to interact with Ume6 was more... Also, ...likely because Ime1 associates with the amino-terminus region of Ume6....

Page 20: *ume6-T99N*, not UME6^{T99N}. Needs a reference for the fact that this allele impairs the interaction between Ume6 and Ime1.

Page 20, middle: we replaced the IME1 locus with GFP fused to the B112 activation domain.

Figure 6D, again, the data overstate the number of cells that "completed meiosis" since binucleate cells are included.

Page 20 bottom, "Rewiring of the Ume6 regulon is sufficient to drive meiosis in the absence of Ime1 and Rim11" sounds like a heading and should be written in bold.

Page 21, Should cite the Benjamin and Herskowitz paper for GAL4-ER unless this is different from the system they developed in which case the reference for the construct used by the authors should be given.

Page 21, middle: we reasoned that alleviation of this repression by depletion of ... To achieve this, we introduced an auxin.... SIN3-AID is an allele and should be italicized. A reference for the AID system is needed.

Figure 6E, *lexA* is a bacterial gene and the first letter should not be capitalized: *lex-GAL4-ER* (italicized).

Page 22 top, reference for the various fluorophore tags should be given

Page 22, How many tetrads were dissected to determine spore viability?

Page 22, end of middle paragraph, Figure 7A , S7A and B should be cited, not 6A and S6A and B

Page 22, middle heading: Rim11 nuclear levels peak before Ime1 and Ume6

Page 24 top: need a reference for the "previous work"...

The first several paragraphs of the discussion are redundant with the results section.

Page 28 top, ...expression of Ime1 is not sufficient to drive... and thus can be rate limiting for ...

Page 28 middle, Once Rim11...complexes are formed, Rim11 exits from the nucleus. Exit from the nucleus..

Page 29, Thus TORC1...localization and the targeting of nuclear...factors...are likely conserved processes from yeast to humans. In contrast to...

Page 30 top ...is predominantly centered...

Page 30 middle, ...whether Rim11 will phosphorylate Ume6 and allow for...

Page 31, Table S3. Plasmids used

Page 31, How were the various genomic manipulations verified?

Define BYTA.

Page 32, I don't understand this sentence: The *ume6-T99N* point mutation was generated by closing and site-directed mutagenesis.

Page 33, BYTA medium, sporulation medium (medium is singular, media is plural).

Page 34, define TCA

Page 35, Line 4, delete "in this thesis" from the sentence about the loading control.

What does it mean that the research was funded "in whole, or in part, by the Wellcome Trust". Doesn't it have to be one or the other?

References

Only the first word and proper nouns should be capitalized in the titles.

Genes and genus species names should be italicized.

Figure 1 legend. The A text is confusing. Were the exponential cells grown in a different medium than the cells used for sporulation? Should indicate that Hxk1 is Hexokinase I and is being used as a loading control. Typo in B text. "We strain was used..." I can't figure out what the authors meant. Rather than "control" it would be clearer to label the exp. Column "Rim11" as this tells the reader that the control was untagged Rim11 (as opposed to rim11Δ for example). What were the statistical tests used for C and D and what p value do three asterisks indicate? In E, "underwent" should not be used after "did not". Instead say "did not undergo meiosis". Need to define AU-I assume that means "arbitrary units". These arbitrary units are not defined in the methods section on image processing.

Referee #2:

In the submitted ms, Kociemba et al investigate the interrelationship of Ume6, Ime1, Rim11, TORC1 and PKA in controlling the meiotic program. They identify Rim11 as a central signal integrator with PKA suppressing Rim11 levels and TORC1 retaining Rim11 in the cytoplasm. Consequentially, inhibition of PKA using a nucleotide analogue sensitive PKA variant increases Rim11 levels, while TORC1 inhibition increases nuclear accumulation of Rim11. Within this pathway, Ime1 acts as an anchor important for Ume6 phosphorylation. Upon complex formation, Ume6-Ime1 drive EMG transcription. This is a very interesting study. All experiments are thoroughly planned and executed. Presented data are of high quality. This reviewer recommends publication, once the authors have addressed the following points:

- Fig 1A: compared to exponentially growing cells, Ume6 is also slightly slower migrating in SDS-PAGE upon starvation in the rim11Δ cells. Is this also a relevant phosphorylation event? Can the authors speculate on the kinase(s) acting on Ume6 under these conditions?
- Fig. 1E-G: it is interesting that the cells not undergoing meiosis seem to have slightly lower amounts of Rim11 at the onset of starvation. Do the authors think there is a correlation? What could be the cause for reduced Rim11 levels in these cells?
- Fig 1H-I: from the figure and figure legend it is not clear if total of nuclear Rim11 is quantified here
- Page 7, last sentence refers to Fig. 1I and states that nuclear Rim11 peaks 3-6h before MI. However, the time of MI is only given as a reference point in Fig. 1H
- Fig. 2F: in this figure, the expression level of Ume6 correlates with expression of Rim11, but seems to be independent from Rim11's subcellular localization. Can the authors explain this observation, especially considering that the RIM11 AA cells do not enter meiosis (Fig. 2D) and thus should not upregulate Ume6-dependent Ume6 expression?
- Fig. 3: the authors show that inhibition of TORC1 and PKA is sufficient to increase Rim11 levels and nuclear localization, which by itself is not sufficient to drive meiotic entry (as shown in Fig. 2). Is TORC1/PKA inhibition sufficient to activate EMG transcription and meiotic entry?
- Page 13: the text states that "previous work suggested that PKA also directly phosphorylates...", for which the authors should provide the reference
- Fig. 4: the authors show that Rim11 expression levels are higher when three alanine residues are mutated, in line with their PKA inhibition assays in Fig. 3. However, the timing of meiosis onset was not affected (S4B) likely because the S3A mutant did not accumulate in the nucleus, in line with the fact that this is controlled by TORC1. Have the authors tried to combine the Rim11 S3A mutation with rapamycin treatment?
- Fig 4F and S4G: for me, it is hard to tell if there is a difference in phosphorylation dynamics between WT and mds3Δ conditions, partly because the expression levels of Ume6-V5 are much lower in the mds3delta mutant. Do the authors know why the Ume6-V5 expression is lower after deletion of Mds3?
- Furthermore the consequences of Mds3 deletion on Ime2 expression (Fig. S4H) are somewhat puzzling in that differences in Ume6 phosphorylation can only be observed very early, whereas differences in Ime2 expression are much more pronounced at later time points, when there is no discernable difference in phosphorylation? Could the effect on Ime2 expression rather be related to the expression level of Ume6-V5 than its phosphorylation?
- S5C and S5D: the mobility shift in SDS-PAGE, from which the authors deduce Rim11-dependent phosphorylation of Ime1, is rather small. Have the authors tested if this shift is sensitive to phosphatase treatment?
- Fig. 5E and 5F: in the experiments presented so far, the expression of Ume6-V5 was significantly higher in WT compared to ime1Δ or rim11Δ strains after 4h in SPO medium (e.g. Fig. 1A or 5C), as might be expected if Ume6 stimulates its own expression in meiosis (page 22). The Ume6-TID was expressed in the same genetic background and differences in the Ume6

expression levels could potentially impact labelling efficiencies in the BioID assays. In Fig. 5E, the expression of Ume6-TID seems to be more or less equal between conditions. Was this representative for all experiments? If yes, how do the authors explain the different responses to Rim11 deletion for Ume6-V5 compared to Ume6-TID?

- Page 22: middle paragraph explains data from Fig. 7, but refers to Fig. 6

- Fig. 7H is not labelled in the figure

- One conclusion of Fig. 7 is that the timing of nuclear Rim11 accumulation can be rate-limiting for meiosis in many cells. However, in Fig. 2H forced nuclear localization of Rim11 did not considerably accelerate Ume6 phosphorylation or meiosis, although this Rim11 version was even higher expressed than the WT. How do the authors explain these two apparently contradictory observations?

- Page 28: the authors discuss that "...once phosphorylated Ume6-Ime1 complexes are formed, Rim11 exists in the nucleus." The authors further state that "Cells accumulate nuclear Rim11 ahead of Ime1 and Ume6." How can the second statement be reconciled with the first statement that phosphorylation of Ume6 and Ime1 in the nucleus is a prerequisite for Rim11 to exist in the nucleus?

Referee #3:

EMBOJ-20230115737

The mechanism determining cell fate is central to development. The authors employed yeast meiosis as a model to unravel the molecular mechanisms behind fate determination. During nutritional starvation, budding yeast diploid cells enter meiosis as meiosis-specific transcriptional mechanisms activate. Three major proteins, Ime1, Ume6, and Rim11, play essential roles in the transcriptional activation of early meiotic genes (EMGs). The authors demonstrated that meiosis-specific phosphorylation of Ume6 depends on Rim11. They also showed the significance of Ume6's nuclear localization, which is independently regulated by the PKA and TORC1 pathways. Additionally, it was revealed that Ime1 is crucial for the nuclear localization of Rim11. Through a combination of experiments and mathematical modeling, the authors propose that metabolic and mating type statuses converge through Rim11 to regulate meiotic entry by activating transcriptional programs.

While there is no doubt that cell fate determination is generally important, I found that the conceptual advance made in this paper was relatively minor. Although the introduction and discussion are well-written, the results section is poorly narrated, making it challenging to follow. I noticed the absence of some key experiments (see below). A major drawback is that many assays rely on RIM11-mNG, whose function seems significantly compromised (Fig. S2B). Additionally, please include line numbers.

Major points:

1. P.6 We fused Rim11 to mNeogreen (Rim11-mNG) and determined its expression and localization in nutrient rich conditions and in cells entering meiosis.

It is essential to state the functionality of the Rim11-mNG fusion when the fusion gene is first introduced. Based on the result shown in Fig. S2B, Rim11-mNG is not very functional. The authors should acknowledge this limitation initially. Following this, the behavior of Rim11-mNG in wild-type cells should be closely examined by comparing the localization kinetics of Rim11-mNG with native Rim11, the latter visualized by immunofluorescence.

2. P.8 To disrupt nuclear Rim11 accumulation, we adopted the anchor-away method as previously described (Haruki et al, 2008).

Apparently, Rim11-AA is far from functional either, already phenocopying the null mutant since it takes 48 hours to observe phospho-Ume6 (and this is in the SK1 background). At least, an attempt should be made to improve the situation, possibly by creating a GFP-less version of RIM11-AA. The legend of Fig. 2D should state the incubation duration for meiosis measurements.

3. P.16 To examine whether Rim11 kinase activity is important for its localization, we mutated its autophosphorylation site Y199 to F (RIM11Y199F), which impairs Rim11 kinase activity.

Protein stability of rim11-Y199F needs to be checked by Western blot. The phenotype could be due to the compromised integrity of the Rim11 protein, which cannot be addressed by fluorescence.

4. P.16 We found that in ime1 cells Rim11 nuclear localization was also significantly reduced (Figure 5B).

It is worth checking the protein integrity of Rim11 in the absence of Ime1. It is critical to establish that the protein stability of Rim11 is not affected by the absence of Ime1 and vice versa. Then, the authors can focus on the mechanism that controls their nuclear localization.

5. P.16 Strikingly, our data showed that Ime1 can be phosphorylated by Rim11 "as soon it is expressed",

A band shift does not directly indicate phosphorylation. Experiments are necessary to examine if the shift is indeed due to phosphorylation. Please check the part in quotes.

6. P.18 The Ume6-TID showed a Rim11 and Ime1 dependent phosphorylation shift demonstrating that the TurboID tag on Ume6 did not affect phosphorylation (Figure 5E).

Is this the same level of phosphorylation as that of untagged Ume6? The legend does not specify the antibody used for Western blot analysis.

7. P.20 We conclude that Rim11 and Ime1 can be made dispensable for EMG activation and meiosis, suggesting that Rim11 and Ime1 primarily function in the Ume6 regulon.

Firstly, given that sporulation is < 20% at 72 hours, Rim11 is hardly dispensable for meiosis. Secondly, assuming B112 is a strong, bona fide transcription activator (as implied from the context, given no citation here), is there any surprise if it targets the same locations as Ume6, and IME2 transcription is indeed induced? In other words, if a strong, foreign transcription factor is overriding the native system, what are readers supposed to learn from this result?

Minor points:

8. P.5 we measured Rim11 dependent phosphorylation by western blotting of Ume6, ...

Is Ume6 phosphorylation documented in the literature? If so, please provide a proper citation

9. P.9 As expected, treatment with rapamycin had no negative effect on IME2 expression in cell harbouring RIM11-mNG instead of RIM11-AA (Figure S2D)

It appears that adding rapamycin also induces IME2 expression in the strain expressing Rim11-mNG.

10. Fig. 2G

What does the asterisk next to "+" mean? The figure legend does not provide an explanation.

11. Fig. 2H

The experimental procedure is unclear. Were copper ions added to the SPO medium at the indicated time point and incubated further? Although this may be detailed in the method section, a brief description in the legend would be beneficial.

12. p.13 A mutant of Rim11 with the three serines substituted with alanines (RIM113SA) displayed increased Rim11 kinase activity in glucose medium, which mis-regulates the onset of meiosis.

Is this finding from the current study, or has it been reported previously? If it's the latter, please provide a proper literature citation.

13. P.13 Both "Rim11" and RIM113SA cells displayed reduced nuclear localization in SPO with glucose (Figure 4B and S4E). We conclude that PKA mediated phosphorylation of Rim11 "is likely destabilizes" Rim11 and reduces its protein levels and propensity to undergo meiosis but had little effect on Rim11 nuclear localization.

Please check the part in quotes.

14. P.17 ...we used a previously characterized IME1 missense mutant (IME1L321F) which impairs the interaction with Rim11

Please provide a proper literature citation for the characterization of this mutant.

15. P.18 Specifically, we tagged Ume6 at the carboxy terminus with TurboID for proximity biotin labelling (Ume6-TID)

Please show data where the functionality of Ume6-TID was examined.

16. P.20 UME6T99N allele impairs the interaction with Ime1, impairing EMG transcriptional activation.

If this allele was not identified in this work, a proper literature citation is needed.

17. P.20 With the use sfGFP-Ime1 and GFP nanobodies tagged UME6T99N (UME6T99N- GFP), it is possible to tether Ime1 to Ume6T99N- GFP and thereby drive EMG transcription and meiosis as shown previously (Figure 6A and 6B).

It is essential to clarify that only IME2 is monitored in this context, rather than making a broad statement about EMG transcription

and meiosis.

18. P.20 However, rim11 cells harbouring sfGFP-IME1/UME6T99N- GFP displayed no EMG activation and no meiosis, which is consistent with the model that Rim11 phosphorylation of Ime1 turns Ime1 into transcriptional activation domain that is essential for driving EMG transcription and meiosis (Bowdish et al, 1994).

How about the involvement of Rim11-dependent phosphorylation of Ume6 as well?

19. P.20 To bypass the requirement for Ime1 phosphorylation by Rim11, we replaced the IME1 locus with GFP fused B112 activation domain (GFP-B112), which allows for phosphorylation independent activation of transcription.

Please provide a proper literature citation for this approach.

20. P.20, 21 We expressed GFP-B112 from a promoter harbouring eight lex operator sequences (p8lexO-GFP-B112) together with the -estradiol regulated GAL4-AD- ER in UME6T99N-aGFP cells.

It should read "lexA-GAL4-AD-ER," as "GAL4-AD-ER" might confuse readers.

21. P.20 Sin3-Rpd3L in repressing EMGs, we reasoned that alleviation of this "repression depletion" of Sin3-Rpd3L is possibly a prerequisite of EMG activation

To achieve this, we "introduced generated" an auxin inducible degron tagged allele of Sin3 (SIN3-AID) to conditionally deplete Sin3

Please check the part in quotes.

22. P.20 Depletion of Sin3 or presence of Tir1 ligase in cells did not result in increased expression of IME2 (Figure 6E, left panel, and Figure S6).

This sentence is confusing. Figure 6E, left panel represents the condition where Sin3 is not depleted.

23. P.20 Despite nutrient rich conditions, IME2 expression was strongly induced in RIM11WT or rim11 cells expressing GFP-B112 and depleted for Sin3 (Figure 6E, right panel).

It is necessary to provide actual data showing that Sin3 is indeed down-regulated upon IAA treatment (possibly by Western blot).

24. Fig.6S

How is this different from Fig. 6E right, and why is there no IME2 induction seen in this experiment? The figure legend needs to be more informative.

25. P.22 First, we compared cells entering meiosis with cells that did not complete a meiotic division also here referred to as quiescent cells ("Figure 6A, and Figure S6A and S6B"). As expected, nuclear levels of Rim11 were elevated in meiotic cells compared to non-meiotic cells ("Figure S6B").

I believe these are all from Fig. 7.

26. P.23 For example, cluster 2 was the most delayed in Rim11 nuclear accumulation, and so were Ime1 and Ume6 nuclear signals and the onset of MI (Figure 7D and 7E).

Isn't cluster 4 more delayed in Rim11 accumulation?

27. Fig. 7E

Please explain what the vertical broken lines represent in the figure legend.

28. P.24 We noted that in cluster 3, nuclear Rim11 accumulated sharply from the 0-hour time point onwards, while Ume6 and Ime1 (to a lesser extent) were slightly delayed, suggesting that Ime1 was rate limiting in cluster 3 for Ume6 induction
P.24 In clusters 4 and 6, we noted that both Rim11 and Ume6 levels initially drop, and subsequently increase at the same time (Figure 7E, right panel; Figure S7E). Ime1 levels, on the other hand, gradually increased over time, suggesting that Rim11 nuclear levels were rate limiting for clusters 4 and 6.

The logic regarding what is rate-limiting is not clear.

29. P.24 In clusters 2 and 5 we noted that Ime1 levels increased well before nuclear Rim11 and Ume6 levels increased,

indicating the Rim11 was the rate limiting step in cluster 2 and 5 (Figure 7E, middle panel and Figure S7E).

The sentence is challenging to understand. If you mean the early increase of Ime1 (0 to ~2 hours), please clarify.

30. P.25 ...using ordinary differential equations (Figure 7F and "Supplementary File 3").

This should refer to Supplementary File 1.

Referee #4:

Previous studies have shown that the induction of meiosis in yeast is transcriptionally regulated by the DNA-binding protein, Ume6, a transcriptional activator Ime1, and the GSK3 β homolog, Rim11 protein kinase. A model for meiotic induction proposed by the Mitchell lab and others in the 1990's/early 2000's posited that Ume6 recruits repressive factors to about 450 so called early meiotic genes (EMGs) in mitotically growing cells and that the transcriptional activator, Ime1, expressed when diploids are starved for key nutrients, replaces the repressive factors bound to Ume6 in a regulated pathway that is controlled by the Rim11 protein kinase and other signaling molecules. Although challenged by a series of papers published in the mid-2000's suggesting that Ume6 is degraded as EMGs are induced, this challenge to has more recently been refuted by published studies from several laboratories. Based on the totality of the data it now seems very likely that Ume6 is indeed the "landing pad" for either repressor proteins in mitotic cells or. Ime1 in starved diploids as proposed in the original model. In the early studies it was shown that Ume6 and Ime1 are phosphorylated by the protein kinase, Rim11 when diploid cells are starved. It has also been shown that Rim11 is controlled by PKA and the van Werven group has shown that inhibition of PKA and TORC are sufficient to induce EMG expression and entry into meiosis.

This paper does a thoughtful job of tying together TORC, PKA, Rim11, Ume6, and Ime1 while focusing on Rim11 and how it connects this signaling network. An impressive series of approaches that support the hypotheses being tested are described and these lead to a model in which PKA and TORC regulate Rim11 transcription and entry into nuclei respectively where Rim11 phosphorylates Ume6 in an Ime1-dependent manner as well as Ime1 itself to activate EMGs. Overall, the work puts a new focus on Rim11 as a key integrator of signals that guide this decision. On the plus side, the article paints the most complete picture of this regulatory transition yet. The work is carefully designed and rigorous. I was unable to find any major flaws or technical problems (although there are some issues pointed out that should be addressed). On the minus side, because meiotic induction in yeast has been studied for many years, many of the interactions in this model have previously been described in the published literature. While the authors should be congratulated on the rigorous and thoughtful analysis of the system, a criticism that could be levied is that the broader conclusions in the paper are somewhat confirmatory of what has already been worked out. Another criticism of the manuscript is that it feels as if there are a half dozen different projects that have been shoe-horned into a single paper. As such, the paper has a lot of transitions and it requires a lot of effort to get through. A less important criticism is that there are a number of awkward transitions and grammatical issues throughout the text of the paper - the text of this article should be tightened up throughout.

Listed below are the key findings of this study with associated comments.

1. The paper starts out by characterizing the phosphorylation of Ume6 by Rim11 (the phosphorylation of Ume6 by Rim11 target was initially reported by in Malathi K, et al. [1997] Mol Cell Biol 17:7230-6) and others.
2. The paper then moves on to convincingly show that Rim11 expression is upregulated by nutritional signals and that its localization to the nucleus is increased in meiotically induced cells.
3. An anchor away technology is used to inducibly eliminate Rim11 from nuclei and thereby show that nuclear localization of Rim11 is essential for activation of early meiotic gene (EMG) transcription and entry into meiosis. This important contribution is fully supported by the data presented.
4. I do not understand the experiment in Fig. 2F - why are 24h and 48h time points, which are long after the entire program has been completed in wild-type cells being tested here? The description of this experiment needs to be clarified.
5. The paper shows that increased expression of Rim11 and entry into the nucleus is not sufficient to cause EMG transcription/meiotic induction. This is not surprising since increased production of Ime1 is also required for EMG transcription and meiotic induction.
6. It is next shown that Rim11 is inhibited by PKA (using an analog-sensitive form of Tpk1 in a *tpk2 Δ tpk3 Δ* background) and by TORC (using rapamycin). The PKA/Rim11 connection has previously been described. Data supporting previously published work suggesting that PKA regulates Rim11 by phosphorylating Rim11's amino terminus is presented. Most importantly, it is next shown that rapamycin promotes entry of Rim11 into nuclei.
7. The paper investigates the role of two genes, MDS3 and PMD1, previously shown to be involved in TORC signaling and also involved in regulating EMGs in the regulation of Rim11. These experiments suggest that Mds3 connects TORC to the localization of Rim11, thereby influencing EMG expression. These data are suggestive and more work is required to elucidate how this might be happening. These findings do not seem well integrated/connected with the rest of the paper.
8. Next, it is shown that Rim11 nuclear abundance is reduced and Ume6 is underphosphorylated in meiotically induced cells when the Rim11 kinase is mutationally inactivated or when IME1 is deleted. The Ime1-L321F mutant that has previously been

shown to be defective in interacting with Rim11 is also shown to be defective in Ume6 phosphorylation. These findings are consistent with Rim11 activity and Ime1 contributing to Rim11 nuclear localization. However, because these mutations decrease meiotic induction cause/effect relationships are not proven. (Note that references describing Rim11-Y209F are missing and the reference for the Ime1-L321F mutant is also missing from this paragraph).

9. Turbo ID of Ume6 is next used to identify a set of substrates in rich medium and SPO. The *ime1* Δ and *rim11* Δ mutants in SPO are also assayed. The authors state that the data were sorted for proteins known to interact with Ume6 or that are associated with transcriptional regulation. How the set of "proteins known to interact with Ume6/associated with transcriptional regulation" was not specified. The set of proteins identified were fairly minor in abundance (only one of the proteins ranked in the top 100 in terms of raw abundance) which creates concerns about how robust the Ume6 Turbo ID system is. Nevertheless, the comparisons of various strains and conditions do support the conclusions. These data support the prevailing model for Ime1 and Rim11 in EMG induction and further suggest that Rim11 and Ime1 are important in recruiting the SAGA complex. The explanation for why Ime1 and Rim11 were not detected in these experiments is not compelling and raise further concerns about the robustness of the system.

10. Mechanistically significant work in the paper uses the Ime1-binding defective UME6T99N-allele fused to GFP (GFP-binding nanobody) in cells expressing GFP-IME1 or GFP-B112 (a transcriptional activator) in the presence or absence of Rim11. These studies show that Rim11 is still essential for EMG induction/meiosis in the GFP-Ime1 system but not in the GFP-B112 system. This indicates that at least one role of Rim11 in inducing EMGs is via Ime1, consistent with a model in which the known phosphorylation of Ime1 by Rim11 converts it into an activator. These experiments connect Ume6 to Ime1 to Rim11 and nicely tie these pieces of the system together. The paper then describes experiments in which GFP-B112 and UME6T99N- GFP were co-expressed in mitotic diploids growing in rich media and show that while EMGs and meiosis were not induced, that the further depletion of Sin3 (important for Ume6-dependent repression) allows EMGs to be induced. These experiments, showing that recruiting an activation domain to Ume6 in cells lacking Rpd3/Sin3 (the repressor proteins normally recruited to Ume6 in mitotic cells) can bypass the requirement of Rim11 in meiotic gene induction show a masterful application of genetic strategies to the system.

11. Clustering analyses is applied to single cell time courses and these data are used to support a correlation between Rim11 in nuclei and the timing of Ime1/increased Ume6 in nuclei and completion of MI. Further follow-up analyses suggest that Rim11 can be rate limiting for MI in some cells while Ime1 can be rate limiting in others.

12. Finally, a mathematical model is constructed that incorporates the regulatory inputs (TORC and PKA) into three regulated steps (Rim11 production, import of Rim11 into nuclei, and Ime1 production). This model seems to explain the variation of single cells in the system.

Referee #1

The paper by Kociemba et al uses a variety of orthogonal approaches to convincingly show that the Rim11 kinase acts to integrate a variety of signals to enable Ime1-mediated transcription of early genes. They show that the starvation conditions required to induce meiosis result in increased Rim11 levels and nuclear localization of the kinase and that nuclear localization of Rim11 is required to induction of early genes like IME2. Abundance and nuclear localization are independently regulated by PKA and TORC1, respectively. In addition they demonstrate that Ime1 promotes Rim11 localization to the nucleus, is phosphorylated by Rim11 prior to Ume6 phosphorylation and that interaction between Rim11 and Ime1 is required for Ume6 phosphorylation. They incorporated their findings into a mathematical model that illustrates how Rim11 affects the timing and amount of Ume6 phosphorylation. This is a very thorough analysis of a complicated problem which elucidates how different signals can be used to regulate transcription of a set of genes that dramatically alters cell fate (going from vegetative growth to meiosis).

While the science in this manuscript is excellent, the document was very difficult to read because of numerous typos, grammatical errors, incorrect use of genetic nomenclature, overuse of jargon, and some unexplained inconsistencies in their results. I gave detailed comments for the Figure 1 legend, but will leave it to the authors to carefully read the remaining legends and correct them.

Major comments:

1. This work directly contradicts the findings published in *Molecular Cell* by the Strich lab where they showed that Ume6 was degraded during meiosis by Cdc20/APC. This contradiction should be mentioned and if the authors have a potential explanation for the different results that would be helpful.

We have added a section to the discussion (see below) explaining the difference between our data and the Strich lab publication. Our results are consistent with two recent publications which show that Ume6 is not degraded until later in meiosis (Harris and Ünal 2023; Raithatha et al. 2021).

“Our findings align with previous work indicating that Ume6 serves as a binding platform for Ime1, facilitating the activation of EMG transcription (Harris & Ünal, 2023; Raithatha et al, 2021; Bowdish et al, 1995). An alternative model proposed that Cdc20 mediated degradation of Ume6 in early meiosis is essential for EMG transcription activation (Mallory et al, 2007). Recent analyses suggest a different timeline, indicating that Ume6 degradation occurs later in meiosis (Harris and Ünal 2023). Consequently, our data support the idea that Ume6 serves as a binding platform for Ime1 that drives EMG transcription and thus entry into meiosis (Harris & Ünal, 2023; Raithatha et al, 2021; Bowdish et al, 1995).”

2. Page 8 bottom, The sentence "...while a large fraction of untreated cells...completed meiosis" is misleading". According to the figure legend the plotted values include binucleate cells which have only completed the first meiotic division. Only tetranucleate cells should be counted as completing meiosis. The same comment is true for Figure S2B, 4D, S4D. Instead of saying that "two or more DAPI masses were considered to have undergone meiosis" it would be more accurate to say "...to have entered meiosis". If the data are broken down to graph binucleates (MI) separately from tetranucleates (MII), are there any differences?

We have revised the text as suggested and now use the term 'entered meiosis' instead of 'completed meiosis.' Our observations did not reveal a distinct difference between binucleates and tetranucleates. Given the manuscript's primary focus on the entry into

meiosis, we have chosen not to incorporate the differentiation between binucleates and tetranucleates throughout the manuscript.

3. I'm confused by the data presented in Figure 2F and S2F. There is no Ume6 phosphorylation at 24 hrs in the Rim11-AA - rap strain in Figure 2F, yet cells from this strain enter Meiosis I at 10 hours (Figure S2B). Could the authors please explain this discrepancy?

We appreciate the reviewer's comment. The combination of Rim11-FRB-GFP and RPL13-FKBP12/*fpr1*Δ resulted in a significant delay in meiosis, with only a fraction of cells entering meiosis at 24 hours (Figure EV 2B). Upon further investigation, we identified that both Rim11-FRB-GFP and RPL13-FKBP12/*fpr1*Δ contribute to this delay. Consequently, we observed a corresponding delay in Ume6 phosphorylation. Most cells still entered meiosis in the Rim11-FRB-GFP combined with RPL13-FKBP12/*fpr1*Δ after 72 hours. Despite that Rim11-FRB-GFP and RPL13-FKBP12/*fpr1*Δ is delayed, we are confident that the data presented in Figure 2 are valid.

4. In Figure 1A, Ume6 phosphorylation increases between 1 and 2 hours after transfer to Spo medium. In Figure 4F, Ume6 is highly phosphorylated within 50 minutes after transfer to Spo medium. Please explain this discrepancy.

In Figure 4F, a Ume6 phosphorylation shift is visible at 50 min in SPO for WT. However, this is not the maximum phosphorylation of Ume6. Instead, it highlights the first timepoint where a difference was observed. We highlighted the difference between WT and *mds3*Δ for the early timepoint as we observed by using 7% SDS-PAGE, running the samples for an extended time. We have added the information to the figure legend. For the later time-points at 4 and 6 hours in SPO we did not observe differences in Ume6 phosphorylation between WT and *mds3*Δ (Appendix Figure S3D).

5. In Figure 4C, *IME2* expression is elevated at 0.5 hours in *mds3*Δ compared to WT and stays elevated at 1 hour (at ~2 arbitrary units). In Figures S4H, the 0.5 hr timepoint wasn't taken, even though that is where the difference in kinetics was previously observed. The relative expression of *IME2* in *mds3*Δ + mock is much lower at the 1 hour timepoint than was observed for *mds3*Δ in Figure 4C. Please explain the discrepancy.

We noted that *IME2* was consistently higher in the *mds3*Δ compared to WT about 2-fold in both Figure 4C and EV3C. Indeed, the induction of *IME2* in Figure 4C in both WT and *mds3*Δ was somewhat slower. Although we try to control for variations between experiments as much as possible, time course experiments can vary to some degree in absolute number and kinetics. In this case there was a significant time gap between the experiment performed in figure 4C and EV3C. New batches of medium or small fluctuations in temperature can affect experimental outcome. To control for this, we always include the WT as a control for every time course experiment. The *IME2* expression in the WT was also lower in Figure S4G but the difference between WT and *mds3*Δ was consistent, indicating that the results are consistent between the two figure panels.

Minor comments

Title page, affiliation 3 writes out United Kingdom while #5 uses UK; Similarly #1 using United States and #4 uses USA-be consistent.

Corrected accordingly.

Page 4, middle, "The kinase Rim11 and DSBk-3b homolog also play a critical role..."
Throughout add a space between the citations at the end of a sentence.
Corrected accordingly.

Define acronyms the first time they are used, for example PKA and TORC1 (this definition is given on page 10 but should be given the first time TORC1 is used).
Corrected accordingly.

Page 5, line 6-7: During starvation Rim11 expression is increased and the protein is localized to the nucleus.
Corrected accordingly.

Page 6 near bottom: ...we also expressed histone H2B fused to mCherry (not sure why "signal" is present)
Corrected accordingly.

Page 8, line 7, ...increased nuclear concentrations of Rim11 not only correlate....
Corrected accordingly.

Figure S2B, The mNG tagged allele appears to be hypomorphic as it is greatly delayed in meiotic progression compared to WT. The authors should mention this. Also there is a typo in the figure legend for this panel: Rim11-mNG, not nNG.
We have dissected the meiosis onset in strains harbouring the various tags on Rim11. We found that Rim11-mNG by itself is not hypomorphic but FRB-GFP on Rim11 is (Figure EV1A, 2A and 2B). Also, cells harbouring *fpr1Δ* and RPL13A-2xFKBP12 display a delay in meiosis (Figure EV2B). We have added these data and revised the manuscript accordingly.

We have corrected the typo.

Page 9, line 8, ...on IME2 expression in cells....
Corrected accordingly.

Page 10, line 2, ...by fusing a nuclear...
Corrected accordingly.

Page 10, line 7, ...forcing Rim11 to the nucleus..
Corrected accordingly.

Page 10, middle, ...to either SPO or SPO plus glucose.
Corrected accordingly.

Page 10, bottom, can be written either "...showed increased Rim11 localization to the nucleus.." or "...showed an increase in Rim11 localization to the nucleus"..
Corrected accordingly.

Page 11, line 1, ...and a more rapid increase in Ume6....
Corrected accordingly.

Page 11, middle, 1-NM-PP1 is not an analog of ATP as it lacks the sugar and phosphate groups of ATP. It is an analog of the PP1 inhibitor which resembles a derivatized adenine. We thank the reviewer for identifying the error. Corrected accordingly. We removed “analog of ATP”

Page 12 top, use the correct allele designation for the analog sensitive kinase: PKA^{AS} is not correct nomenclature.

We named the allele *tpk1-as*.

Figures S4A-please provide p values for those differences that are significant.

We assessed the data presented in Figure S4A. We observed that this pertains to a published microarray experiment where the number of replicates used is not explicitly stated. Since the data are not crucial to the conclusions of the manuscript, we have made the decision to exclude this experiment from the manuscript.

Page 12, bottom, ...Rim11 depletion was much faster in the nucleus than the cytoplasm.... Corrected accordingly.

Page 13, top. A reference is needed for the statement that PKA phosphorylates specific residues on Rim11 and the fact that the RIM11^{S3A} mutant exhibits increased Rim11 kinase activity in glucose.

Reference is included.

Page 13 bottom, We conclude that...of Rim11 likely destabilizes...and has little effect...(keep the verb tenses the same).

We re-wrote the sentence.

Heading: Mds3 promotes Rim11 localization via TORC1

Corrected accordingly.

Why is the synergistic effect of combining inactivation of PKA with inhibition of TORC not seen if Figure S3B?

Appendix Figure S2B represents the whole cell signal. TORC1 mostly controls the localization of Rim11, while PKA controls Rim11 levels. Hence only the PKA effect is visible in Appendix Figure S2B. Our data suggest that TORC1 controls localization and not the levels of Rim11.

Page 14 middle, ...we examined IME2 expression and meiotic progression...

The timepoint shown in Figure 4E should be indicated on the figure and in the figure legend. Corrected accordingly.

Page 15 top, Please indicate where the data are that show that *mds3Δ* cells still display Rim11 nuclear localization changes upon starvation...

Corrected accordingly.

Page 15 bottom, ...the transcription activation domain function of Ime1 that drives...

Corrected accordingly.

Page 16 top, ...interaction with Ime1 contribute to the Rim11 localization pattern....

Corrected accordingly.

Page 16 top.. mutated tyrosine 199 to phenylalanine (T199F). Since this mutation impairs Rim1 kinase activity, it is likely a recessive mutant. The proper way to write the genotype is

rim11-Y199F (italicized). All capital letters (RIM11^{Y199F}) indicates the allele is dominant. If that is the case, then please explicitly state this.

The nomenclature should be corrected in Figure 5 and S5 as well.

Corrected accordingly.

Page 16 bottom, In *Saccharomyces cerevisiae* nomenclature, wild-type alleles are indicated by all capital letters with no allele designation, so RIM11 not RIM11^{WT}.

Corrected accordingly.

Page 17: ime1-L321F (italicized), not IME1^{L321F}. A reference is needed for the fact that this mutation impairs the interaction between Ime1 and Rim11.

Corrected accordingly.

Page 17, end of middle paragraph, This surprising result suggests....

Corrected accordingly.

Page 18, mass spectrometry (MS)

Corrected accordingly.

Page 18 middle, Conversely, the Tod6 subunit that is known to interact with Ume6 was more...

Also, ...likely because Ime1 associates with the amino-terminus region of Ume6....

Corrected accordingly.

Page 20: ume6-T99N, not UME6^{T99N}. Needs a reference for the fact that this allele impairs the interaction between Ume6 and Ime1.

Corrected accordingly.

Page 20, middle: we replaced the IME1 locus with GFP fused to the B112 activation domain.

Corrected accordingly.

Figure 6D, again, the data overstate the number of cells that "completed meiosis" since binucleate cells are included.

Corrected accordingly. We used the term entered meiosis.

Page 20 bottom, "Rewiring of the Ume6 regulon is sufficient to drive meiosis in the absence of Ime1 and Rim11" sounds like a heading and should be written in bold.

Corrected accordingly.

Page 21, Should cite the Benjamin and Herskowitz paper for GAL4-ER unless this is different from the system they developed in which case the reference for the construct used by the authors should be given.

Corrected accordingly. The construct contains lexA fused to the Gal4 activation domain and estradiol receptor responsive domain (lexA-GAL4AD-ER)

Page 21, middle: we reasoned that alleviation of this repression by depletion of ...

To achieve this, we introduced an auxin.....

SIN3-AID is an allele and should be italicized. A reference for the AID system is needed.

Corrected accordingly.

Figure 6E, lexA is a bacterial gene and the first letter should not be capitalized: lexA-GAL4-ER (italicized).

Corrected accordingly.

Page 22 top, reference for the various fluorophore tags should be given

Corrected accordingly.

Page 22, How many tetrads were dissected to determine spore viability?
200 spores (50 tetrads)

Page 22, end of middle paragraph, Figure 7A , S7A and B should be cited, not 6A and S6A and B

Corrected accordingly.

Page 22, middle heading: Rim11 nuclear levels peak before Ime1 and Ume6

Corrected accordingly.

Page 24 top: need a reference for the "previous work"...

Corrected accordingly.

The first several paragraphs of the discussion are redundant with the results section.

Corrected accordingly.

Page 28 top, ...expression of Ime1 is not sufficient to drive... and thus can be rate limiting for

...

Corrected accordingly.

Page 28 middle, Once Rim11...complexes are formed, Rim11 exits from the nucleus. Exit from the nucleus..

Corrected accordingly.

Page 29, Thus TORC1...localization and the targeting of nuclear...factors...are likely conserved processes from yeast to humans. In contrast to...

Corrected accordingly.

Page 30 top ...is predominantly centered...

Corrected accordingly.

Page 30 middle, ...whether Rim11 will phosphorylate Ume6 and allow for...

Corrected accordingly.

Page 31, Table S3. Plasmids used

Corrected accordingly.

Page 31, How were the various genomic manipulations verified?

Genomic integrations were verified by PCR with primers flanking the targeted region and, where feasible, further confirmed through western blotting. We added this to the methods.

Define BYTA.

Page 32, I don't understand this sentence: The ume6-T99N point mutation was generated by closing and site-directed mutagenesis.

Corrected accordingly. We re-wrote the section.

Page 33, BYTA medium, sporulation medium (medium is singular, media is plural).

Corrected accordingly.

Page 34, define TCA

Corrected accordingly.

Page 35, Line 4, delete "in this thesis" from the sentence about the loading control.

Corrected accordingly.

What does it mean that the research was funded "in whole, or in part, by the Wellcome Trust". Doesn't it have to be one or the other?

Corrected accordingly.

References

Only the first word and proper nouns should be capitalized in the titles.

Genes and genus species names should be italicized.

Corrected accordingly.

Figure 1 legend. The A text is confusing. Were the exponential cells grown in a different medium than the cells used for sporulation? Should indicate that Hxk1 is Hexokinase I and is being used as a loading control. Typo in B text. "We strain was used..." I can't figure out what the authors meant. Rather than "control" it would be clearer to label the exp. Column "Rim11" as this tells the reader that the control was untagged Rim11 (as opposed to rim11 Δ for example). What were the statistical tests used for C and D and what p value do three asterisks indicate? In E, "underwent" should not be used after "did not". Instead say "did not undergo meiosis". Need to define AU-I assume that means "arbitrary units". These arbitrary units are not defined in the methods section on image processing.

We rewrote the legends and added the information to methods. We have checked and corrected all the legends.

Image analysis of fixed cells was carried out using Fiji, where whole cell selection was determined manually and nuclei selection was determined via segmentation (Schindelin et al, 2012). The mean values (mean intensity) were calculated by taking the sum of the grey values of all pixels within the selected object over the number of pixels. To investigate on nuclear localisation of proteins, the ratio between the nuclear mean intensity over the whole mean intensity was taken.

The arbitrary units of LCI represent the total or average fluorescence signal intensity registered in a fluorescent channel after correction for background, autofluorescence, and bleed-through, as described in the fluorescence extraction code as described previously (Kamat 2019; Arguello-Miranda et al. 2018; Acuña-Rodríguez, Mena-Vega, and Argüello-Miranda 2022).

Referee #2

In the submitted ms, Kociemba et al investigate the interrelationship of Ume6, Ime1, Rim11, TORC1 and PKA in controlling the meiotic program. They identify Rim11 as a central signal integrator with PKA suppressing Rim11 levels and TORC1 retaining Rim11 in the cytoplasm. Consequentially, inhibition of PKA using a nucleotide analogue sensitive PKA variant increases Rim11 levels, while TORC1 inhibition increases nuclear accumulation of Rim11. Within this pathway, Ime1 acts as an anchor important for Ume6 phosphorylation. Upon complex formation, Ume6-Ime1 drive EMG transcription. This is a very interesting study. All experiments are thoroughly planned and executed. Presented data are of high quality. This reviewer recommends publication, once the authors have addressed the following points:

Fig 1A: compared to exponentially growing cells, Ume6 is also slightly slower migrating in SDS-PAGE upon starvation in the *rim11Δ* cells. Is this also a relevant phosphorylation event? Can the authors speculate on the kinase(s) acting on Ume6 under these conditions?

The reviewer correctly noted that in Figure 1A a small shift of Ume6 can be observed between YPD and BYTA which occurs in a Rim11-independent manner. There are indeed several candidate kinases that could phosphorylate Ume6. For example, Mck1 and Rim15 are known to target Ume6 as well during meiosis (Xiao and Mitchell 2000). We added this information to the manuscript.

Fig. 1E-G: it is interesting that the cells not undergoing meiosis seem to have slightly lower amounts of Rim11 at the onset of starvation. Do the authors think there is a correlation? What could be the cause for reduced Rim11 levels in these cells?

We believe that the lower Rim11 levels are a characteristic feature of cells that have not entered meiosis. It is indeed interesting that the levels are already lower at the earliest time point. Previous studies have also reported that meiotic and quiescent/non-meiotic cells can differ at early stages (Arguello-Miranda et al. 2018). The signal integration to Rim11 via TORC1 and PKA is likely affected in cells not entering, as we have elaborated upon in more detail throughout manuscript.

Fig 1H-I: from the figure and figure legend it is not clear if total of nuclear Rim11 is quantified here

We have clarified the legend and figure accordingly. Mean nuclear Rim11 levels are displayed in this figure.

Page 7, last sentence refers to Fig. 1I and states that nuclear Rim11 peaks 3-6h before MI. However, the time of MI is only given as a reference point in Fig. 1H

The reference should be indeed Fig. 1H. We corrected the section accordingly.

Fig. 2F: in this figure, the expression level of Ume6 correlates with expression of Rim11, but seems to be independent from Rim11's subcellular localization. Can the authors explain this observation, especially considering that the RIM11 AA cells do not enter meiosis (Fig. 2D) and thus should not upregulate Ume6-dependent Ume6 expression?

We do not have an explanation for why Ume6 levels in the *rim11Δ* are lower. Possibly, Rim11-FRB-GFP stabilizes Ume6, which remains to be investigated. We speculate that some residual Rim11-FRB-GFP may be present in the nucleus when the cells were treated with rapamycin, which might be sufficient to stabilize Ume6. More importantly we reported that Ume6 is not phosphorylated in the Rim11-FRB-GFP cells treated with rapamycin.

Fig. 3: the authors show that inhibition of TORC1 and PKA is sufficient to increase Rim11 levels and nuclear localization, which by itself is not sufficient to drive meiotic entry (as shown in Fig. 2). Is TORC1/PKA inhibition sufficient to activate EMG transcription and meiotic entry?

Yes, inhibition of TORC1 and PKA is sufficient to drive EMG and meiosis. We will provide the reference accordingly, which is (Weidberg et al. 2016).

Page 13: the text states that "previous work suggested that PKA also directly phosphorylates...", for which the authors should provide the reference

We have included the reference, which is (Ifat Rubin-Bejerano et al. 2004).

Fig. 4: the authors show that Rim11 expression levels are higher when three alanine residues are mutated, in line with their PKA inhibition assays in Fig. 3. However, the timing of meiosis onset was not affected (S4B) likely because the S3A mutant did not accumulate in the nucleus, in line with the fact that this is controlled by TORC1. Have the authors tried to combine the Rim11 S3A mutation with rapamycin treatment?

The rim11-S3A mutant leads to upregulation of Rim11 levels. We indeed observed little difference on the onset of meiosis in population-based assay (Appendix Figure S3A). However, we did observe increased meiosis in LCI (Appendix Figure S3C). It is worth noting that in Appendix Figure S1F we monitored overexpression of *NLS-RIM11*, which was not sufficient to enter meiosis. We think that both Ime1 and Rim11 are rate limiting. Therefore, when Rim11 is induced earlier Ime1 will be rate limiting.

- Fig 4F and S4G: for me, it is hard to tell if there is a difference in phosphorylation dynamics between WT and *mds3Δ* conditions, partly because the expression levels of Ume6-V5 are much lower in the *mds3Δ* mutant. Do the authors know why the Ume6-V5 expression is lower after deletion of Mds3?

We have no explanation for the slight decrease in Ume6-V5 levels observed in the *mds3Δ* strain. Consistently across multiple time course experiments, we have noted a discernible difference in Ume6 migration between WT and *mds3Δ*. Consequently, we have concluded that there is a consistent, albeit small, distinction in Ume6 phosphorylation kinetics between WT and *mds3Δ*.

Furthermore the consequences of Mds3 deletion on Ime2 expression (Fig. S4H) are somewhat puzzling in that differences in Ume6 phosphorylation can only be observed very early, whereas differences in Ime2 expression are much more pronounced at later time points, when there is no discernable difference in phosphorylation? Could the effect on Ime2 expression rather be related to the expression level of Ume6-V5 than its phosphorylation?

While maximum expression is at later timepoint, we do note that differences *IME2* levels are already observed at the early time points between WT and *mds3Δ* (e.g. 1 hour timepoint Figure EV3C or 0.5 hour timepoint in Figure 4C). There is a consistent 2-fold difference between *mds3Δ* and WT, suggesting that *IME2* expression is higher in the *mds3Δ* from early time point onwards.

We do not think that the lower levels of Ume6 affects meiotic entry because when we lower Ume6 levels by using a weaker promoter (uninduced *CUP1*- promoter), cells can still undergo meiosis with nearly WT kinetics (Figure EV3D).

- S5C and S5D: the mobility shift in SDS-PAGE, from which the authors deduce Rim11-dependent phosphorylation of Ime1, is rather small. Have the authors tested if this shift is sensitive to phosphatase treatment?

We have now included a Ume6 migration shift assay of Ime1 in Figure EV4E. We noted that Ime1 protein migrated faster in CIP treated extracts indicating the shift is Rim11-dependent.

- Fig. 5E and 5F: in the experiments presented so far, the expression of Ume6-V5 was significantly higher in WT compared to *ime1Δ* or *rim11Δ* strains after 4h in SPO medium (e.g. Fig. 1A or 5C), as might be expected if Ume6 stimulates its own expression in meiosis

(page 22). The Ume6-TID was expressed in the same genetic background and differences in the Ume6 expression levels could potentially impact labelling efficiencies in the BioID assays. In Fig. 5E, the expression of Ume6-TID seems to be more or less equal between conditions. Was this representative for all experiments? If yes, how do the authors explain the different responses to Rim11 deletion for Ume6-V5 compared to Ume6-TID?

It is possible that the TID-tag makes Ume6 slightly less stable. However, Ume6 is fully functional as cells enter meiosis in the Ume6-TID strain (Appendix Figure S4D). Also, lower levels of Ume6 do not affect meiosis, hence we do not think that this is an issue (Figure EV3D).

- Page 22: middle paragraph explains data from Fig. 7, but refers to Fig. 6
Corrected accordingly.

- Fig. 7H is not labelled in the figure
Corrected accordingly.

- One conclusion of Fig. 7 is that the timing of nuclear Rim11 accumulation can be rate-limiting for meiosis in many cells. However, in Fig. 2H forced nuclear localization of Rim11 did not considerably accelerate Ume6 phosphorylation or meiosis, although this Rim11 version was even higher expressed than the WT. How do the authors explain these two apparently contradictory observations?

We propose in the manuscript that both Ime1 and Rim11 can be rate-limiting. In Fig 2H we force Rim11 to the nucleus, however cells do not enter meiosis faster indicating the nuclear Rim11 is not sufficient for entry into meiosis. In these cells Ime1 is likely the rate-limiting step. We have clarified in the text that Ime1 can also be rate limiting.

- Page 28: the authors discuss that "...once phosphorylated Ume6-Ime1 complexes are formed, Rim11 exists in the nucleus." The authors further state that "Cells accumulate nuclear Rim11 ahead of Ime1 and Ume6." How can the second statement be reconciled with the first statement that phosphorylation of Ume6 and Ime1 in the nucleus is a prerequisite for Rim11 to exist in the nucleus?

We thank the reviewer for pointing out the error. It should say "exit" in the first statement and not "exist". We have corrected this accordingly.

Referee #3

EMBOJ-20230115737

The mechanism determining cell fate is central to development. The authors employed yeast meiosis as a model to unravel the molecular mechanisms behind fate determination. During nutritional starvation, budding yeast diploid cells enter meiosis as meiosis-specific transcriptional mechanisms activate. Three major proteins, Ime1, Ume6, and Rim11, play essential roles in the transcriptional activation of early meiotic genes (EMGs). The authors demonstrated that meiosis-specific phosphorylation of Ume6 depends on Rim11. They also showed the significance of Ume6's nuclear localization, which is independently regulated by the PKA and TORC1 pathways. Additionally, it was revealed that Ime1 is crucial for the nuclear localization of Rim11. Through a combination of experiments and mathematical modeling, the authors propose that metabolic and mating type statuses converge through Rim11 to regulate meiotic entry by activating transcriptional programs.

While there is no doubt that cell fate determination is generally important, I found that the conceptual advance made in this paper was relatively minor. Although the introduction and discussion are well-written, the results section is poorly narrated, making it challenging to follow. I noticed the absence of some key experiments (see below). A major drawback is that many assays rely on RIM11-mNG, whose function seems significantly compromised (Fig. S2B). Additionally, please include line numbers.

Major points:

1. P.6 We fused Rim11 to mNeogreen (Rim11-mNG) and determined its expression and localization in nutrient rich conditions and in cells entering meiosis.

The mNG on Rim11 itself does not affect entry into meiosis. We have included the data in Figure S1A. In Figure S2C, the Rim11-mNG also harbours *frp1* Δ and RPL13A-2xFKBP12. We now show that *frp1* Δ and RPL13A-2xFKBP12 cause a delay in meiosis (Figure EV2B).

It is essential to state the functionality of the Rim11-mNG fusion when the fusion gene is first introduced. Based on the result shown in Fig. S2B, Rim11-mNG is not very functional. The authors should acknowledge this limitation initially. Following this, the behavior of Rim11-mNG in wild-type cells should be closely examined by comparing the localization kinetics of Rim11-mNG with native Rim11, the latter visualized by immunofluorescence.

See previous point. The mNG on Rim11 itself does not affect entry into meiosis. We have included the data in Figure EV1A.

2. P.8 To disrupt nuclear Rim11 accumulation, we adopted the anchor-away method as previously described (Haruki et al, 2008).

Apparently, Rim11-AA is far from functional either, already phenocopying the null mutant since it takes 48 hours to observe phospho-Ume6 (and this is in the SK1 background). At least, an attempt should be made to improve the situation, possibly by creating a GFP-less version of RM11-AA. The legend of Fig. 2D should state the incubation duration for meiosis measurements.

The Rim11-FRB-GFP affects the timing of the onset of meiosis (Figure EV2A and 2B), but the majority of cells enter eventually after 72 hours (Figure 2D). Even though, there is a delay in entry in meiosis, cells were able to enter meiosis to a similar degree as the WT when mock-treated. Hence, we believe that our conclusions are valid, despite the meiotic delay in the Rim11- FRB-GFP strain.

It is worth noting that we have made several unsuccessful attempts to optimize the Rim11-FRB-GFP experiment by generating different versions of the Rim11- FRB-GFP without GFP, which required a significant amount of time. Together the delay in entry into meiosis is caused by a combination of both Rim11- FRB-GFP, *frp1* Δ and RPL13A-2xFKBP12.

3. P.16 To examine whether Rim11 kinase activity is important for its localization, we mutated its autophosphorylation site Y199 to F (RIM11Y199F), which impairs Rim11 kinase activity.

Protein stability of rim11-Y199F needs to be checked by Western blot. The phenotype could be due to the compromised integrity of the Rim11 protein, which cannot be addressed by fluorescence.

In previous studies, the *rim11-Y199F* mutant has been described and well characterized biochemically and in yeast meiosis. These works showed that during sporulation, the rim11-Y199F protein was well detected, and kinase assays with *rim11-Y199F* showed comparable Rim11 levels to the WT (Zhan et al. 2000; Bowdish, Yuan, and Mitchell 1994). *In vitro* it has

been shown that rim11-Y199F mutant can undergo autophosphorylation but does not phosphorylate Ume6, see Figure 6 of (Zhan et al. 2000). Based on these previous studies, there is no reason to think that the rim11-Y199F protein integrity is affected in our study. We noted that whole cell levels of the rim11-Y199F-mNG were comparable with WT Rim11-mNG at most time points (Figure S4B). We used the rim11-Y199F-mNG mutant to determine whether localization is regulated by Rim11's ability to phosphorylate substrates, which was not the case.

4. P.16 We found that in ime1• cells Rim11 nuclear localization was also significantly reduced (Figure 5B).

It is worth checking the protein integrity of Rim11 in the absence of Ime1. It is critical to establish that the protein stability of Rim11 is not affected by the absence of Ime1 and vice versa. Then, the authors can focus on the mechanism that controls their nuclear localization.

To address the point raised by the reviewer, we expressed Rim11/Ime1 from the *CUP1* promoter in WT and *ime1Δ* or *rim11Δ* cells and treated cells with cycloheximide (in Figure EV4A). We observed no difference in Rim11 expression and no difference in Rim11 relative stability between *IME1* and *ime1Δ* cells. Conversely, we observed no difference in Ime1 stability between *RIM11* and *rim11Δ* cells. It is worth noting that Ime1 is a relatively unstable protein.

5. P.16 Strikingly, our data showed that Ime1 can be phosphorylated by Rim11 "as soon it is expressed",

A band shift does not directly indicate phosphorylation. Experiments are necessary to examine if the shift is indeed due to phosphorylation. Please check the part in quotes.

We have tested phosphatase sensitivity of Ime1. We observed that the higher migrating Ime1 was strongly reduced upon CIP treatment (Figure EV4E).

6. P.18 The Ume6-TID showed a Rim11 and Ime1 dependent phosphorylation shift demonstrating that the TurboID tag on Ume6 did not affect phosphorylation (Figure 5E).

Is this the same level of phosphorylation as that of untagged Ume6? The legend does not specify the antibody used for Western blot analysis.

The TID has a Myc tag, which we used to detect Ume6 and phosphorylation shifts of Ume6. By western blotting it is challenging to obtain quantitative information regarding the phosphorylation status of Ume6. However, we noted that the TID tag on Ume6 does not interfere with entry into meiosis (Appendix Figure S4D).

7. P.20 We conclude that Rim11 and Ime1 can be made dispensable for EMG activation and meiosis, suggesting that Rim11 and Ime1 primarily function in the Ume6 regulon.

Firstly, given that sporulation is < 20% at 72 hours, Rim11 is hardly dispensable for meiosis. Secondly, assuming B112 is a strong, bona fide transcription activator (as implied from the context, given no citation here), is there any surprise if it targets the same locations as Ume6, and *IME2* transcription is indeed induced? In other words, if a strong, foreign transcription factor is overriding the native system, what are readers supposed to learn from this result?

We have re-worded the conclusion and state that Rim11 is “partly” dispensable. We believe the experiment is informative since it demonstrates that we understand how the system works showing that Rim11 is acting primarily through Ume6 and Ime1.

Minor points:

8. P.5 we measured Rim11 dependent phosphorylation by western blotting of Ume6, ...

Is Ume6 phosphorylation documented in the literature? If so, please provide a proper citation
We have included the relevant citations.

9. P.9 As expected, treatment with rapamycin had no negative effect on IME2 expression in cell harbouring RIM11-mNG instead of RIM11-AA (Figure S2D)

It appears that adding rapamycin also induces IME2 expression in the strain expressing Rim11-mNG.

We agree that rapamycin treatment induces *IME2* expression. Therefore, we stated that rapamycin has “no negative effect” on *IME2* expression.

10. Fig. 2G

What does the asterisk next to "+" mean? The figure legend does not provide an explanation.

We used two different concentrations of copper sulphate. We have clarified this in the legend accordingly.

11. Fig. 2H

The experimental procedure is unclear. Were copper ions added to the SPO medium at the indicated time point and incubated further? Although this may be detailed in the method section, a brief description in the legend would be beneficial.

We have clarified this in legend.

12. p.13 A mutant of Rim11 with the three serines substituted with alanines (RIM113SA) displayed increased Rim11 kinase activity in glucose medium, which mis-regulates the onset of meiosis.

Is this finding from the current study, or has it been reported previously? If it's the latter, please provide a proper literature citation.

We have included the relevant citation.

13. P.13 Both "Rim11" and RIM113SA cells displayed reduced nuclear localization in SPO with glucose (Figure 4B and S4E). We conclude that PKA mediated phosphorylation of Rim11 "is likely destabilizes" Rim11 and reduces its protein levels and propensity to undergo meiosis but had little effect on Rim11 nuclear localization.

Please check the part in quotes.

We have corrected the textual error accordingly.

14. P.17 ...we used a previously characterized IME1 missense mutant (IME1L321F) which impairs the interaction with Rim11

Please provide a proper literature citation for the characterization of this mutant.

We have included the citation.

15. P.18 Specifically, we tagged Ume6 at the carboxy terminus with TurboID for proximity biotin labelling (Ume6-TID)

Please show data where the functionality of Ume6-TID was examined.

We have included data showing that Ume6-TID can enter meiosis (Appendix Figure S4D).

16. P.20 UME6T99N allele impairs the interaction with Ime1, impairing EMG transcriptional activation.

If this allele was not identified in this work, a proper literature citation is needed.

We included the citation.

17. P.20 With the use sfGFP-Ime1 and GFP nanobodies tagged UME6T99N (UME6T99N-•GFP), it is possible to tether Ime1 to Ume6T99N-•GFP and thereby drive EMG transcription and meiosis as shown previously (Figure 6A and 6B).

It is essential to clarify that only IME2 is monitored in this context, rather than making a broad statement about EMG transcription and meiosis.

We have amended the text accordingly and explicitly state *IME2*.

18. P.20 However, rim11• cells harbouring sfGFP-IME1/UME6T99N-•GFP displayed no EMG activation and no meiosis, which is consistent with the model that Rim11 phosphorylation of Ime1 turns Ime1 into transcriptional activation domain that is essential for driving EMG transcription and meiosis(Bowdish et al, 1994).

How about the involvement of Rim11-dependent phosphorylation of Ume6 as well?

It is not likely that Ume6 itself has a transcriptionally activating function. First, the T99 is a key phosphorylation site of Ume6. The T99N mutant is completely impaired in EMG activation and meiosis. The T99N mutant can be bypassed by tethering Ime1 to Ume6, suggesting that the T99N site is not involved in activating transcription, but rather the recruitment of Ime1. Two recent manuscripts dissected the model further and supported the hypothesis that Ume6 acts as a binding platform for repressors and activators (Harris and Ünal 2023; Raithatha et al. 2021).

19. P.20 To bypass the requirement for Ime1 phosphorylation by Rim11, we replaced the IME1 locus with GFP fused B112 activation domain (GFP-B112), which allows for phosphorylation independent activation of transcription.

Please provide a proper literature citation for this approach.

We have included the citations.

20. P.20, 21 We expressed GFP-B112 from a promoter harbouring eight lex operator sequences (p8 \times lexO-GFP-B112) together with the \bullet -estradiol regulated GAL4-AD- ER in UME6T99N-aGFP cells.

It should read "lexA-GAL4-AD-ER," as "GAL4-AD-ER" might confuse readers. We have amended the text accordingly.

21. P.20 Sin3-Rpd3L in repressing EMGs, we reasoned that alleviation of this "repression depletion" of Sin3-Rpd3L is possibly a prerequisite of EMG activation To achieve this, we "introduced generated" an auxin inducible degron tagged allele of Sin3 (SIN3-AID) to conditionally deplete Sin3

Please check the part in quotes. We have amended the text accordingly.

22. P.20 Depletion of Sin3 or presence of Tir1 ligase in cells did not result in increased expression of IME2 (Figure 6E, left panel, and Figure S6).

This sentence is confusing. Figure 6E, left panel represents the condition where Sin3 is not depleted.

We have amended the text accordingly, see below.

To achieve this, we generated an auxin inducible degron tagged allele of Sin3 (SIN3-AID) to conditionally deplete Sin3 (Appendix Figure S5A) (Nishimura et al, 2009). Despite nutrient rich conditions, *IME2* expression was strongly induced in *rim11* Δ cells expressing GFP-B112 and depleted for Sin3 (Figure 6E, right panel). Control cells, whether Sin3 was not depleted (mock-treated or expressing no Tir1 ligase) or expressing no lexA-GAL4AD-ER, did not show increased expression of IME2 (Figure 6E, left panel, and Appendix Figure S5B). We conclude that nutrient control of EMG transcription via *lme1* and *Rim11* can be bypassed, in part, by tethering an activation domain to *Ume6* in cells depleted for Sin3-Rpd3L.

23. P.20 Despite nutrient rich conditions, IME2 expression was strongly induced in RIM11WT or *rim11* \bullet cells expressing GFP-B112 and depleted for Sin3 (Figure 6E, right panel).

It is necessary to provide actual data showing that Sin3 is indeed down-regulated upon IAA treatment (possibly by Western blot).

We have provided the western blot in Figure S6A.

24. Fig.6S

How is this different from Fig. 6E right, and why is there no IME2 induction seen in this experiment? The figure legend needs to be more informative.

Sin3 was not depleted in Figure 6E left panel, while it was depleted in right panel. In figure S6B cells did not express lexA-GAL4AD-ER. We have added the information to the figure legends.

25. P.22 First, we compared cells entering meiosis with cells that did not complete a meiotic division also here referred to as quiescent cells ("Figure 6A, and Figure S6A and S6B"). As expected, nuclear levels of Rim11 were elevated in meiotic cells compared to non-meiotic cells ("Figure S6B").

I believe these are all from Fig. 7.

We have amended the text accordingly.

26. P.23 For example, cluster 2 was the most delayed in Rim11 nuclear accumulation, and so were Ime1 and Ume6 nuclear signals and the onset of MI (Figure 7D and 7E).

Isn't cluster 4 more delayed in Rim11 accumulation?

In cluster 4 is more delayed. We have amended the text accordingly.

27. Fig. 7E

Please explain what the vertical broken lines represent in the figure legend.

We added an explanation to the legend. The vertical dashed lines indicate the points at which Rim11, Ime1, and Ume6 signals collectively begin to increase thereafter.

28. P.24 We noted that in cluster 3, nuclear Rim11 accumulated sharply from the 0-hour time point onwards, while Ume6 and Ime1 (to a lesser extent) were slightly delayed, suggesting that Ime1 was rate limiting in cluster 3 for Ume6 induction

P.24 In clusters 4 and 6, we noted that both Rim11 and Ume6 levels initially drop, and subsequently increase at the same time (Figure 7E, right panel; Figure S7E). Ime1 levels, on the other hand, gradually increased over time, suggesting that Rim11 nuclear levels were rate limiting for clusters 4 and 6.

The logic regarding what is rate-limiting is not clear.

Ume6 is an EMG. Hence, an increase or decrease in Ume6 reflects EMG expression. The facts that Rim11 or Ime1 (depending on the cluster) show the same increase or decrease as the Ume6 expression suggests that nuclear levels Rim11 or Ime1 were rate limiting for the Ume6 expression. We have clarified this in the main text.

29. P.24 In clusters 2 and 5 we noted that Ime1 levels increased well before nuclear Rim11 and Ume6 levels increased, indicating the Rim11 was the rate limiting step in cluster 2 and 5 (Figure 7E, middle panel and Figure S7E).

The sentence is challenging to understand. If you mean the early increase of Ime1 (0 to ~2 hours), please clarify.

We have amended the text accordingly.

In clusters 2 and 5, we observed an early increase in Ime1 levels, preceding the elevation of nuclear Rim11 and Ume6 levels (Figure 7E, middle panel and Figure EV7E). This suggests that Rim11 serves as the rate-limiting step in clusters 2 and 5.

30. P.25 ...using ordinary differential equations (Figure 7F and "Supplementary File 3").

This should refer to Supplementary File 1.

We have amended the error.

Referee #4

Previous studies have shown that the induction of meiosis in yeast is transcriptionally regulated by the DNA-binding protein, Ume6, a transcriptional activator Ime1, and the GSK3 β homolog, Rim11 protein kinase. A model for meiotic induction proposed by the Mitchell lab and others in the 1990's/early 2000's posited that Ume6 recruits repressive factors to about 450 so called early meiotic genes (EMGs) in mitotically growing cells and that the transcriptional activator, Ime1, expressed when diploids are starved for key nutrients, replaces the repressive factors bound to Ume6 in a regulated pathway that is controlled by the Rim11 protein kinase and other signaling molecules. Although challenged by a series of papers published in the mid-2000's suggesting that Ume6 is degraded as EMGs are induced, this challenge to has more recently been refuted by published studies from several laboratories. Based on the totality of the data it now seems very likely that Ume6 is indeed the "landing pad" for either repressor proteins in mitotic cells or. Ime1 in starved diploids as proposed in the original model. In the early studies it was shown that Ume6 and Ime1 are phosphorylated by the protein kinase, Rim11 when diploid cells are starved. It has also been shown that Rim11 is controlled by PKA and the van Werven group has shown that inhibition of PKA and TORC are sufficient to induce EMG expression and entry into meiosis.

This paper does a thoughtful job of tying together TORC, PKA, Rim11, Ume6, and Ime1 while focusing on Rim11 and how it connects this signaling network. An impressive series of approaches that support the hypotheses being tested are described and these lead to a model in which PKA and TORC regulate Rim11 transcription and entry into nuclei respectively where Rim11 phosphorylates Ume6 in an Ime1-dependent manner as well as Ime1 itself to activate EMGs. Overall, the work puts a new focus on Rim11 as a key integrator of signals that guide this decision. On the plus side, the article paints the most complete picture of this regulatory transition yet. The work is carefully designed and rigorous. I was unable to find any major flaws or technical problems (although there are some issues pointed out that should be addressed). On the minus side, because meiotic induction in yeast has been studied for many years, many of the interactions in this model have previously been described in the published literature. While the authors should be congratulated on the rigorous and thoughtful analysis of the system, a criticism that could be levied is that the broader conclusions in the paper are somewhat confirmatory of what has already been worked out. Another criticism of the manuscript is that it feels as if there are a half dozen different projects that have been shoe-horned into a single paper. As such, the paper has a lot of transitions and it requires a lot of effort to get through. A less important criticism is that there are a number of awkward transitions and grammatical issues throughout the text of the paper - the text of this article should be tightened up throughout.

We went thoroughly over the manuscript and corrected the grammatical errors and improved the transitions between the different sections.

Listed below are the key findings of this study with associated comments.

1. The paper starts out by characterizing the phosphorylation of Ume6 by Rim11 (the phosphorylation of Ume6 by Rim11 target was initially reported by in Malathi K, et al. [1997] Mol Cell Biol 17:7230-6) and others.

2. The paper then moves on to convincingly show that Rim11 expression is upregulated by nutritional signals and that its localization to the nucleus is increased in meiotically induced cells.

3. An anchor away technology is used to inducibly eliminate Rim11 from nuclei and thereby show that nuclear localization of Rim11 is essential for activation of early meiotic gene (EMG) transcription and entry into meiosis. This important contribution is fully supported by the data presented.

4. I do not understand the experiment in Fig. 2F - why are 24h and 48h time points, which are long after the entire program has been completed in wild-type cells being tested here? The description of this experiment needs to be clarified.

The Rim11-FRB1-GFP+ RPL13A-FKBP12 + *fpr1*Δ showed a delay in undergoing meiosis compared to WT (see Figure EV2B). Hence, we decided to take a later timepoint to assess Ume6 phosphorylation status. We have added these new data and provided an explanation in main text.

5. The paper shows that increased expression of Rim11 and entry into the nucleus is not sufficient to cause EMG transcription/meiotic induction. This is not surprising since increased production of Ime1 is also required for EMG transcription and meiotic induction.

We agree.

6. It is next shown that Rim11 is inhibited by PKA (using an analog-sensitive form of Tpk1 in a *tpk2*Δ *tpk3*Δ background) and by TORC (using rapamycin). The PKA/Rim11 connection has previously been described. Data supporting previously published work suggesting that PKA regulates Rim11 by phosphorylating Rim11's amino terminus is presented. Most importantly, it is next shown that rapamycin promotes entry of Rim11 into nuclei.

7. The paper investigates the role of two genes, MDS3 and PMD1, previously shown to be involved in TORC signaling and also involved in regulating EMGs in the regulation of Rim11. These experiments suggest that Mds3 connects TORC to the localization of Rim11, thereby influencing EMG expression. These data are suggestive and more work is required to elucidate how this might be happening. These findings do not seem well integrated/connected with the rest of the paper.

We agree that the molecular mechanism by which Mds3 regulates Rim11 localization is not well characterized. However, the *mds3*Δ data serve the purpose of showing that the TORC1 signalling contributes to Rim11 localization.

8. Next, it is shown that Rim11 nuclear abundance is reduced and Ume6 is underphosphorylated in meiotically induced cells when the Rim11 kinase is mutationally inactivated or when IME1 is deleted. The Ime1-L321F mutant that has previously been shown to be defective in interacting with Rim11 is also shown to be defective in Ume6

phosphorylation. These findings are consistent with Rim11 activity and Ime1 contributing to Rim11 nuclear localization. However, because these mutations decrease meiotic induction cause/effect relationships are not proven. (Note that references describing Rim11-Y209F are missing and the reference for the Ime1-L321F mutant is also missing from this paragraph).

We have included the relevant references in the revised manuscript and described the mutations in detail in the main text.

9. Turbo ID of Ume6 is next used to identify a set of substrates in rich medium and SPO. The *ime1* Δ and *rim11* Δ mutants in SPO are also assayed. The authors state that the data were sorted for proteins known to interact with Ume6 or that are associated with transcriptional regulation. How the set of "proteins known to interact with Ume6/associated with transcriptional regulation" was not specified. The set of proteins identified were fairly minor in abundance (only one of the proteins ranked in the top 100 in terms of raw abundance) which creates concerns about how robust the Ume6 Turbo ID system is. Nevertheless, the comparisons of various strains and conditions do support the conclusions. These data support the prevailing model for Ime1 and Rim11 in EMG induction and further suggest that Rim11 and Ime1 are important in recruiting the SAGA complex. The explanation for why Ime1 and Rim11 were not detected in these experiments is not compelling and raise further concerns about the robustness of the system.

The TurboID (TID) tag was added to the C-terminus of Ume6. Rim11 and Ime1 associate with a defined region at the N-terminus of Ume6 (I. Rubin-Bejerano et al. 1996). The TID has an enzymatic reach of 10 – 15 nm to biotinylate its substrates (Kim et al. 2014). Ume6 is a relatively large protein of nearly 800 amino acids. The TID at the C-terminus of Ume6 was not in reach of Ime1 and Rim11, which was possibly because a distance of more than 500 amino acids spans a radius larger than 15 nm. In addition, the interactions between Ime1, Rim11 and Ume6 might be transient and dynamic in cells, and therefore challenging to capture robustly. As expected we did detect several subunits of the Sin3-Rpd3 complex in the TID experiment, which associate with Ume6 at a defined region more towards the C-terminus of Ume6 (Washburn and Esposito 2001). Taken together, we believe that the TID data support the conclusion that Ime1 and Rim11 are both required for Ume6 mediated transcriptional activation of EMGs. We have added a description in the text based on the description above.

"Rim11 and Ime1 were not detected by Ume6-TID. Plausible explanations are that Ime1 is an unstable protein and that Ime1 and Rim11 associates with the amino-terminus of Ume6 which falls outside the range of TID at the carboxy-terminus of Ume6 (Kim et al, 2014)."

10. Mechanistically significant work in the paper uses the Ime1-binding defective UME6T99N-allele fused to •GFP (GFP-binding nanobody) in cells expressing GFP-IME1 or GFP-B112 (a transcriptional activator) in the presence or absence of Rim11. These studies show that Rim11 is still essential for EMG induction/meiosis in the GFP-Ime1 system but not in the GFP-B112 system. This indicates that at least one role of Rim11 in inducing EMGs is via Ime1, consistent with a model in which the known phosphorylation of Ime1 by Rim11 converts it into an activator. These experiments connect Ume6 to Ime1 to Rim11 and nicely tie these pieces of the system together. The paper then describes experiments in which GFP-B112 and UME6T99N-•GFP were co-expressed in mitotic diploids growing in rich media and show that while EMGs and meiosis were not induced, that the further depletion of Sin3 (important for Ume6-dependent repression) allows EMGs to be induced. These experiments, showing that recruiting an activation domain to Ume6 in cells lacking Rpd3/Sin3 (the repressor proteins normally recruited to Ume6 in mitotic cells) can bypass the requirement of Rim11 in meiotic gene induction show a masterful application of genetic strategies to the system.

We agree that this is an important experiment to demonstrate how the system works.

11. Clustering analyses is applied to single cell time courses and these data are used to support a correlation between Rim11 in nuclei and the timing of Ime1/increased Ume6 in nuclei and completion of MI. Further follow-up analyses suggest that Rim11 can be rate limiting for MI in some cells while Ime1 can be rate limiting in others.

12. Finally, a mathematical model is constructed that incorporates the regulatory inputs (TORC and PKA) into three regulated steps (Rim11 production, import of Rim11 into nuclei, and Ime1 production). This model seems to explain the variation of single cells in the system.

References:

- Acuña-Rodríguez, Jessy Pamela, Jean Paul Mena-Vega, and Orlando Argüello-Miranda. 2022. "Live-Cell Fluorescence Spectral Imaging as a Data Science Challenge." *Biophysical Reviews* 14 (2): 579–97.
- Arguello-Miranda, O., Y. Liu, N. E. Wood, P. Kositangool, and A. Doncic. 2018. "Integration of Multiple Metabolic Signals Determines Cell Fate Prior to Commitment." *Molecular Cell* 71 (5): 733-744 e11.
- Bowdish, K. S., H. E. Yuan, and A. P. Mitchell. 1994. "Analysis of RIM11, a Yeast Protein Kinase That Phosphorylates the Meiotic Activator IME1." *Molecular and Cellular Biology* 14 (12): 7909–19.
- Harris, Anthony, and Elçin Ünal. 2023. "The Transcriptional Regulator Ume6 Is a Major Driver of Early Gene Expression during Gametogenesis." *Genetics*, July. <https://doi.org/10.1093/genetics/iyad123>.
- Kamat, Prashant V. 2019. "Absolute, Arbitrary, Relative, or Normalized Scale? How to Get the Scale Right." *ACS Energy Letters* 4 (8): 2005–6.
- Kim, Dae In, K. C. Birendra, Wenhong Zhu, Khaterreh Motamedchaboki, Valérie Doye, and Kyle J. Roux. 2014. "Probing Nuclear Pore Complex Architecture with Proximity-Dependent Biotinylation." *Proceedings of the National Academy of Sciences of the United States of America* 111 (24): E2453-61.
- Raithatha, Sheetal A., Shivani Vaza, M. Touhidul Islam, Brianna Greenwood, and David T. Stuart. 2021. "Ume6 Acts as a Stable Platform To Coordinate Repression and Activation of Early Meiosis-Specific Genes in *Saccharomyces Cerevisiae*." *Molecular and Cellular Biology* 41 (7): e0037820.
- Rubin-Bejerano, I., S. Mandel, K. Robzyk, and Y. Kassir. 1996. "Induction of Meiosis in *Saccharomyces Cerevisiae* Depends on Conversion of the Transcriptional Repressor Ume6 to a Positive Regulator by Its Regulated Association with the Transcriptional Activator Ime1." *Molecular and Cellular Biology* 16 (5): 2518–26.
- Rubin-Bejerano, Ifat, Shira Sagee, Osnat Friedman, Lilach Pnueli, and Yona Kassir. 2004. "The in Vivo Activity of Ime1, the Key Transcriptional Activator of Meiosis-Specific Genes in *Saccharomyces Cerevisiae*, Is Inhibited by the Cyclic AMP/Protein Kinase A Signal Pathway through the Glycogen Synthase Kinase 3-Beta Homolog Rim11." *Molecular and Cellular Biology* 24 (16): 6967–79.
- Washburn, B. K., and R. E. Esposito. 2001. "Identification of the Sin3-Binding Site in Ume6 Defines a Two-Step Process for Conversion of Ume6 from a Transcriptional Repressor to an Activator in Yeast." *Molecular and Cellular Biology* 21 (6): 2057–69.
- Weidberg, H., F. Moretto, G. Spedale, A. Amon, and F. J. van Werven. 2016. "Nutrient Control of Yeast Gametogenesis Is Mediated by TORC1, PKA and Energy Availability." *PLoS Genetics* 12 (6): e1006075.
- Xiao, Y., and A. P. Mitchell. 2000. "Shared Roles of Yeast Glycogen Synthase Kinase 3 Family Members in Nitrogen-Responsive Phosphorylation of Meiotic Regulator Ume6p." *Molecular and Cellular Biology* 20 (15): 5447–53.
- Zhan, X. L., Y. Hong, T. Zhu, A. P. Mitchell, R. J. Deschenes, and K. L. Guan. 2000. "Essential Functions of Protein Tyrosine Phosphatases PTP2 and PTP3 and RIM11 Tyrosine Phosphorylation in *Saccharomyces Cerevisiae* Meiosis and Sporulation." *Molecular Biology of the Cell* 11 (2): 663–76.

Dear Dr. van Werven,

Thank you for the submission of your revised manuscript to The EMBO Journal. We have now received the comments of the four referees that were asked to re-assess your study (included below). As you will see, the referees acknowledge that the manuscript has been significantly improved and most of their previous concerns have been addressed, but referees #2, #3 and #4 also point out that there are still several issues to be resolved before your manuscript can be published in The EMBO Journal.

I would thus like to invite you to submit another revised version of your manuscript, addressing all remaining concerns of referees #2, #3 and #4, along with a detailed point-by-point response to their comments. Please note that acceptance of your manuscript will depend on the completeness of your responses in this revised version.

From the editorial side, there are also a few changes and corrections that we need from you:

- Please enter all relevant funding information in our online manuscript handling system (eJP). It should match exactly the information provided in the Acknowledgements section of your manuscript. There are currently mismatches in the grant numbers (i.e. FC001203 in eJP, CC2043 in the manuscript).
- Please add a list of up to 5 keywords after the Abstract.
- Please change the heading of your conflict-of-interest statement to "Disclosure and competing interests statement".
- No author contributions statement should be included in the manuscript file. Instead, we use CRediT to specify the contributions of each author in the journal submission system. Please use the free text box to provide more detailed descriptions during submission. See also our guide to authors for more information:
<https://www.embopress.org/page/journal/14602075/authorguide#authorshipguidelines>.
- Please upload main and Expanded View Figures as individual files (see our guide for more information and instructions: <https://www.embopress.org/page/journal/14602075/authorguide#figureformat>). Their legends should remain in the main manuscript, at the bottom of the file.
- Please move the text describing the mathematical model (in your Dataset EV1) to the Materials and Methods of the main manuscript file.
- Please rename Tables EV1 and EV2 to Dataset EV1 and Dataset EV2, respectively. Their callouts should be updated accordingly throughout the manuscript. Please provide their legends in a separate tab in each Excel file.
- Tables EV3 and EV4 should then be renamed to Table EV1 and EV2, respectively; their corresponding callouts should be updated throughout the manuscript.
- Please note that the Appendix should be provided as a single PDF file. It should start with a brief Table of Contents including page numbers on its first page. The nomenclature of its figures should be Appendix Figure S1-S5 - please update accordingly all figures and their legends in the Appendix, as well as their callouts throughout the main manuscript file.
- We noticed that source data for Fig. 3D-F, 7G are missing.
- Please organize the source data files in one zipped folder per figure. For example, all source data files for Figure 1 panels need to be saved in a single folder, which then needs to be zipped and uploaded as "SD Figure 1.zip".
- Please note that EMBO press papers are accompanied online by:
 - A) a short (2 sentences) summary of the findings and their significance,
 - B) 2-5 short bullet points highlighting the key results, and
 - C) a synopsis image in .jpg or .png format that is exactly 550 pixels wide and 300-600 pixels high (the height is variable). You can either show a model or key data in the synopsis image. Please note that the text needs to be legible at the final size. Please upload this information along with your revised manuscript (the text for A and B should be provided in a separate Word file).
- Please note that a specific URL for the deposited dataset PXD049212 in the Data availability statement will be necessary before acceptance of the manuscript for publication in The EMBO Journal.
- Please note that information related to "n" is missing in the legends of Figures 3a-c; 4a-b, e; 5a-b; 6c; EV 1a-b.

- Please note that $n=2$ in Figure EV 1c. No statistics should be calculated and shown when $n=2$; instead, please show the individual data points.
- Please note that the error bars are not defined in the legends of Figures 4c-d; 6a, c; EV 1a-c.
- Please note that the box plot needs to be defined in terms of minima, maxima, centre, bounds of box and whiskers, and percentile in the legend of Figure EV 3b.
- Please note that the scale bar needs to be defined in the legends of Figures 1b; 2b; 3a-c; 4a, e; 5a-b.
- Please provide the legend of each movie file in a separate Word file zipped together with the corresponding movie file.

Please also note that as part of the EMBO publications' Transparent Editorial Process, The EMBO Journal publishes online a Peer Review File along with each accepted manuscript. This File will be published in conjunction with your paper and will include the referee reports, your point-by-point response and all pertinent correspondence relating to the manuscript. You can opt out of this by letting the editorial office know (contact@embojournal.org). If you do opt out, the Peer Review File link will point to the following statement: "No Peer Review File is available with this article, as the authors have chosen not to make the review process public in this case."

We look forward to seeing a final version of your manuscript as soon as possible. Please use this link to submit your revision: <https://emboj.msubmit.net/cgi-bin/main.plex>

Yours sincerely,

 Referee #1:

The authors have adequately addressed my comments from the previous review.

Referee #2:

- Fig. 5E and 5F: in the experiments presented so far, the expression of Ume6-V5 was significantly higher in WT compared to $\text{ime1}\Delta$ or $\text{rim11}\Delta$ strains after 4h in SPO medium (e.g. Fig. 1A or 5C), as might be expected if Ume6 stimulates its own expression in meiosis (page 22). The Ume6-TID was expressed in the same genetic background and differences in the Ume6 expression levels could potentially impact labelling efficiencies in the BioID assays. In Fig. 5E, the expression of Ume6-TID seems to be more or less equal between conditions. Was this representative for all experiments? If yes, how do the authors explain the different responses to Rim11 deletion for Ume6-V5 compared to Ume6-TID?

It is possible that the TID-tag makes Ume6 slightly less stable. However, Ume6 is fully functional as cells enter meiosis in the Ume6-TID strain (Appendix Figure S4D). Also, lower levels of Ume6 do not affect meiosis, hence we do not think that this is an issue (Figure EV3D).

This does not really answer the question. It was not my point to ask if altered Ume6 levels have an impact on the timing of meiosis or if Ume6-TID functions as the endogenous protein does.

1. It is not clear to me why decreased stability of Ume6-TID-tag would explain that there is no difference in Ume6-TID levels at 4h SPO between WT and $\Delta\text{ime1} \Delta\text{rim11}$ conditions? In Fig. 4C, Ume6-V5 displays a very clear difference at 4h SPO in the respective conditions.
2. Stability of Ume6-V5 and Ume6-TID could be easily compared by a CHX shut-off experiment.
3. The authors did not answer the question if the equal expression levels between WT and $\Delta\text{ime1} \Delta\text{rim11}$ conditions shown in Fig. 5E were representative for all replicate experiments or if the other replicates showed a behaviour similar to Fig. 5C. This is a really critical point, as labelling efficiency of potential targets will highly depend on the total amount of TID-tagged protein in each condition.

Referee #3:

EMBOJ-2023-115737

The authors have, for the most part, adequately addressed the points raised although there are still several issues to be resolved.

Major point

1. P.25 Rim11 nuclear accumulation can be rate-limiting for EMG activation

I did not quite understand the whole argument about what is rate-limiting. I understand that nuclear accumulation of Rim11 and Ume6 are concurrent, but so is Ime1. The authors say that Ime1 is rate-limiting in cluster 3, but it seems to me that Rim11's increase is pretty much concurrent with the increase of Ume6 there as well. In cluster 3, both Ime1 and Rim11 accumulate sharply from the beginning, which made me wonder why Ume6 accumulation is delayed there anyway. In any case, if there is a certain threshold in the amount of Ime1 necessary for induction of Ume6, Ime1 could as well be rate-limiting in every cluster. Then how could this experiment allow the authors to determine which factor is rate-limiting for EMG activation?

Minor points

2. P.6 Indeed Rim15 and Mck1 have been shown to be substrates of Ume6

Is this right? I think it is the other way around.

3. P.7 The Rim11-mNG strain exhibited wild type (WT) kinetics of meiosis..

P.19 The Ume6-TID exhibited a phosphorylation shift dependent on Rim11 and Ime1, and it did not impact meiotic entry...

To me, meiosis of the strain producing Rim11-mNG looks delayed, by 1 hr or so, as seen in Fig. EV1A. This trend is also evident in Fig. EV2 B and C. Similarly the UME6-TID strain is also delayed as seen in Fig. S4D. Then I am wondering why the authors do not admit these trends.

4. Figure 6E

What does 0 hr and 1 hr represent? Time after addition of copper, IAA, or both? Authors should specify that.

5. Figure 7D, left panel

What is "e"?

6. P.61, third line from the bottom

"in induced in WT.." does not make sense.

7. P.11 ...gene deletions in TPK2 and TPK3 led to an overall increase in Rim11-mNG signal YPD.

in YPD?

8. P.14 These data suggest that Mds3, but not Pmd1, acts as a repressor of IME2 expression, consequently influencing the onset of meiosis.

It seems to me that the fact that the mds3 pmd1 double mutant does not exhibit rapid induction of Ime2 is not supported by this suggestion.

9. P.17 However, unlike Ume6, Rim11 phosphorylates Ime1 in cells grown in pre-sporulation medium.

Here, Ime1 phosphorylation indeed relies on Rim11, but that does not necessarily mean that Rim11 phosphorylates Ime1.

10. P.20 Recent work showed that Ime1 can be made indispensable for entry into meiosis ..

It must be the other way around. Ime1 can be dispensable under certain conditions, not indispensable.

11. P.23 The four-color strain was able to undergo meiosis and form viable spores (92 {plus minus} % viability)

92 {plus minus} ?

12. P.23 Furthermore, Ime1 and Ume6 nuclear levels were up-regulated in cells entering meiosis but not in non-meiotic cells (Figure 7A, and Figure EV5A).

Figure EV5B?

13. Figure EV5E

No broken line in the left panel.

14. P.24 We conclude that Rim11 and Ime1 nuclear accumulation starts roughly at the same time, while Rim11 peaks well prior to Ime1 and Ume6.

I suppose it is "Ume6" and Ime1 instead of "Rim11" and Ime1?

Referee #4:

This revised manuscript describes a scientifically rigorous and comprehensive analysis of the mechanisms that tie PKA, TOR, Ime1, Ume6, and Rim11 together in a network that controls meiotic induction in yeast. My criticisms of the technical/scientific aspects of the original submission were minor and these criticisms have been addressed. Comments made about significance of the original submission were not addressed by the authors. However, as described in my initial review, it is clear that the paper integrates a lot of new data into the prevailing model for meiotic induction and this moves the field forward in a significant way. A problem with the original submission was the way it was presented, which made it difficult to read. I found this revision just as difficult to read as the original article and do not believe this criticism has been adequately addressed. The text of this manuscript needs to be tightened up throughout. There are multiple errors/grammatical problems on most pages of the Introduction, Results and Discussion, in the Figure legends and elsewhere (too numerous to mention). The paper would benefit from significant editing. While I appreciate the challenges of presenting so much data on multiple well-studied regulatory proteins in a single paper, I feel that additional efforts are required. It would be a shame if the outstanding science in this article received less readership than it deserves because of presentation issues. A few illustrative examples are below.

1- The section showing that "Rim11 nuclear accumulation is required for entry into meiosis" on p8 was particularly hard to get through. The point of all this is presumably to convince the reader that "nuclear accumulation of Rim11 is required for Ume6 phosphorylation and thus IME2 transcription" This should be communicated in a more linear, straightforward, and accessible manner.

2- The liberal inclusion of "Supplementary and Appendix" figures and videos made the job of getting through the article daunting. More effort to explain why the data has been included and help the reader along would help. For example, two thirds of the way through p 10 it is stated that "pCUP1-NLS-RIM11 induced cells also showed a comparable Ume6 migration pattern to WT, and the onset of meiosis was delayed in pCUP1-NLS-RIM11 cells (Figure 2H and Appendix Figure 1F)" This distracts the reader without driving the story forward. If the authors feel it is important to include these findings, then something along the lines of "driving overexpressed Rim11 into nuclei with pCUP1-NLS-RIM11 did not noticeably effect the migration of Ume6 and only modestly delayed the kinetics of meiosis". These findings show.... A minor point is that it should be emphasized that the RIM11 chromosomal locus is replaced with pCUP1-NLS-RIM11 in these experiments (this is not a plasmid or other genetic manipulation). These kinds of things can help the naive reader understand the considerable rigor and care that was used in the study.

3- Another problem related to the numerous pieces of data in figures proper, in the "expanded view (EV)" and in supplementary figures or movies is that the authors do not always explicitly state where a particular piece of data actually is. A simple example, on p 11 states "cells shifted to SPO plus rapamycin showed increased nuclear localization of Rim11, and a more rapid increase in Ume6 phosphorylation (Figure 3B and Appendix Figure S2A)." So, if the reader is interested in looking at one of these pieces of data they first need to sort out which conclusion is supported by which figure (or whether both actually show the same thing) and where to look next. These instances should be recast to something like "of Rim11 (Figure 3B), and a more rapid increase in Ume6 phosphorylation (Appendix Figure S2A)." This is a problem throughout the paper that makes it difficult for an attentive reader.

4- The fact that glucose and Tpk1 control the level of Rim11 is a significant piece of how this network works (as discussed around the top of p12) [but maybe the Mitchell lab previously showed this, and if so, this should be referenced]. If this is an original finding, the data to support this should not be relegated to a supplementary figure.

5- On the middle of p12, the authors describe transferring cells from starvation conditions to rich medium. It is unclear exactly what is being done here. Perhaps it is somewhere in the paper but this needs to be explained clearly at this point.

6- I still do not feel that the mds3 and pmd1 data substantially advance the story. Irrespective, if the authors choose to keep this section in the paper, the image in Fig. 4F should be straightened out and the labels are appropriately aligned.

7- The liberal use of adjectives like "critical" and "important" is distracting. For example, on p3 the phrase "undergo a critical cell-fate decision" should just be "undergo a cell-fate decision" and on p6 the phrase "Ume6, a key prime substrate of Rim11" should just be "Ume6, a substrate of Rim11". Similar changes throughout would help reduce unnecessary words.

8- The discussion could use some additional work.

9- Perhaps referring to the model in Fig. 7F in the discussion to guide the reader or even using the model as a way to direct the discussion might help to clarify how this system works and would also be helpful in summarizing the new findings of the paper.

From the editorial side, there are also a few changes and corrections that we need from you:

- Please enter all relevant funding information in our online manuscript handling system (eJP). It should match exactly the information provided in the Acknowledgements section of your manuscript. There are currently mismatches in the grant numbers (i.e. FC001203 in eJP, CC2043 in the manuscript).

We have corrected the grant numbers.

- Please add a list of up to 5 keywords after the Abstract.

We have added 5 keywords.

- Please change the heading of your conflict-of-interest statement to "Disclosure and competing interests statement".

Done.

- No author contributions statement should be included in the manuscript file. Instead, we use CRediT to specify the contributions of each author in the journal submission system. Please use the free text box to provide more detailed descriptions during submission. See also our guide to authors for more information: <https://www.embopress.org/page/journal/14602075/authorguide#authorshipguidelines>.

Done.

- Please upload main and Expanded View Figures as individual files (see our guide for more information and instructions: <https://www.embopress.org/page/journal/14602075/authorguide#figureformat>). Their legends should remain in the main manuscript, at the bottom of the file.

Done.

- Please move the text describing the mathematical model (in your Dataset EV1) to the Materials and Methods of the main manuscript file.

Done.

- Please rename Tables EV1 and EV2 to Dataset EV1 and Dataset EV2, respectively. Their callouts should be updated accordingly throughout the manuscript. Please provide their legends in a separate tab in each Excel file.

Done.

- Tables EV3 and EV4 should then be renamed to Table EV1 and EV2, respectively; their corresponding callouts should be updated throughout the manuscript.

Done.

- Please note that the Appendix should be provided as a single PDF file. It should start with a brief Table of Contents including page numbers on its first page. The nomenclature of its figures should be Appendix Figure S1-S5 - please update accordingly all figures and their legends in the Appendix, as well as their callouts throughout the main manuscript file.

Done.

- We noticed that source data for Fig. 3D-F, 7G are missing.

We have added the data for Figure 3D-F. Figure 7G is mathematical modelling. There is no underlying experimental data for this panel, however, we have included the modelling data.

- Please organize the source data files in one zipped folder per figure. For example, all source data files for Figure 1 panels need to be saved in a single folder, which then needs to be zipped and uploaded as "SD Figure 1.zip".

Done.

- Please note that EMBO press papers are accompanied online by:
A) a short (2 sentences) summary of the findings and their significance,
B) 2-5 short bullet points highlighting the key results, and
C) a synopsis image in .jpg or .png format that is exactly 550 pixels wide and 300-600 pixels high (the height is variable). You can either show a model or key data in the synopsis image. Please note that the text needs to be legible at the final size. Please upload this information along with your revised manuscript (the text for A and B should be provided in a separate Word file).

Done

- Please note that a specific URL for the deposited dataset PXD049212 in the Data availability statement will be necessary before acceptance of the manuscript for publication in The EMBO Journal.

Done

- Please note that information related to "n" is missing in the legends of Figures 3a-c; 4a-b, e; 5a-b; 6c; EV 1a-b.

Done.

- Please note that n=2 in Figure EV 1c. No statistics should be calculated and shown when n=2; instead, please show the individual data points.

Done.

- Please note that the error bars are not defined in the legends of Figures 4c-d; 6a, c; EV 1a-c.

Done.

- Please note that the box plot needs to be defined in terms of minima, maxima, centre, bounds of box and whiskers, and percentile in the legend of Figure EV 3b.

Done.

- Please note that the scale bar needs to be defined in the legends of Figures 1b; 2b; 3a-c; 4a, e; 5a-b.

Done.

- Please provide the legend of each movie file in a separate Word file zipped together with the corresponding movie file.

Done.

Please also note that as part of the EMBO publications' Transparent Editorial Process, The EMBO Journal publishes online a Peer Review File along with each accepted manuscript. This File will be published in conjunction with your paper and will include the referee reports, your point-by-point response and all pertinent correspondence relating to the manuscript. You can opt out of this by letting the editorial office know (contact@embojournal.org). If you do opt out, the Peer Review File link will point to the following statement: "No Peer Review File is available with this article, as the authors have chosen not to make the review process public in this case."

We look forward to seeing a final version of your manuscript as soon as possible. Please use this link to submit your revision: <https://emboj.msubmit.net/cgi-bin/main.plex>

Yours sincerely,

Referee #1:

The authors have adequately addressed my comments from the previous review.

Thank you.

Referee #2:

- Fig. 5E and 5F: in the experiments presented so far, the expression of Ume6-V5 was significantly higher in WT compared to *ime1*Δ or *rim11*Δ strains after 4h in SPO medium (e.g. Fig. 1A or 5C), as might be expected if Ume6 stimulates its own expression in meiosis (page 22). The Ume6-TID was expressed in the same genetic background and differences in the Ume6 expression levels could potentially impact labelling efficiencies in the BioID assays. In Fig. 5E, the expression of Ume6-TID seems to be more or less equal between conditions. Was this representative for all experiments? If yes, how do the authors explain the different responses to Rim11 deletion for Ume6-V5 compared to Ume6-TID?

It is possible that the TID-tag makes Ume6 slightly less stable. However, Ume6 is fully functional as cells enter meiosis in the Ume6-TID strain (Appendix Figure S4D). Also, lower levels of Ume6 do not affect meiosis, hence we do not think that this is an issue (Figure EV3D).

This does not really answer the question. It was not my point to ask if altered Ume6 levels have an impact on the timing of meiosis or if Ume6-TID functions as the endogenous protein does.

1. It is not clear to me why decreased stability of Ume6-TID-tag would explain that there is no difference in Ume6-TID levels at 4h SPO between WT and $\Delta ime1 \Delta rim11$ conditions? In Fig. 4C, Ume6-V5 displays a very clear difference at 4h SPO in the respective conditions.

2. Stability of Ume6-V5 and Ume6-TID could be easily compared by a CHX shut-off experiment.

3. The authors did not answer the question if the equal expression levels between WT and $\Delta ime1 \Delta rim11$ conditions shown in Fig. 5E were representative for all replicate experiments or if the other replicates showed a behaviour similar to Fig. 5C. This is a really critical point, as labelling efficiency of potential targets will highly depend on the total amount of TID-tagged protein in each condition.

We agree that there is no clear increase of Ume6-TID levels. In our view, there are several possibilities for why this is the case. For the experiment presented in Figure 5E, cells were grown in a larger culture, which may affect the synchrony of meiosis. Hence, these cells may have entered meiosis a bit slower. Moreover, the Ume6-TID strain itself has a delay in meiosis (as also pointed out by reviewer 3), which will impact the induction of Ume6-TID. We have now acknowledged there is a delay in the Ume6-TID strain.

We have quantified Ume6-TID levels in an additional experiment (see below, Figure 1). We noted that Ume6 levels in Ume6-TID increased to some extent, and that

phosphorylation is clearly present (which is most critical). We acknowledge that the increase in Ume6-TID levels is not as strong as other experiments in the manuscript.

We could spend a significant amount of time trying to dissect exactly why the Ume6-TID induction is not so strong. However, we feel that this may not be so trivial, and will potentially require a significant amount of work. As stated above a plausible explanation is that Ume6-TID has a small delay in meiosis, which was also pointed out by reviewer 3. Additionally, it could be that Ume6-TID + phosphorylation makes the protein less stable, which would be challenging to dissect. In our view, the observation that Ume6-TID levels are somewhat affected is not critical for conclusions of the TID experiment and other conclusions in the manuscript. However, if the reviewer considers the Ume6-TID invalid because of the observation the Ume6-TID levels were not induced efficiently, we will opt to remove the experiment from the current version of the manuscript.

Regarding the second point of the reviewer on whether the levels of Ume6 would impact the outcome of the proximity labelling experiment. We agree that Ume6 levels are ideally comparable, which was the case for the experiment we presented in the manuscript. However, the proximity labelling highlights relative differences between WT and the deletion mutants. The identified proteins and their abundance from pulled downs, depend on labelling efficiency, but also on IP efficiency, breakage of cell efficiency, elution efficiency, and likely several other parameters. Hence, in order to make comparisons, a normalization factor is required to correct for variations between the eluates obtained from WT and mutants. For the normalization we assume that overall levels of proteins in eluates do not differ between the conditions. As consequence we can only assess relative differences between the conditions. The approach is a standard way of comparing conditions with each other by mass spectrometry and is described in the methods section.

It is worth emphasizing that the data obtained from the TID experiment fit well with previously described observations. When acting as a repressor, Ume6 recruits the Sin3-Rpd3L complex. We identified several subunits of the Sin3-Rpd3L complex, of which some were more enriched in the *rim11* and *ime1* deletion mutants. Additionally, multiple subunits of the SAGA complex were enriched in the WT compared to *rim11* and *ime1* mutants. SAGA has been previously reported to associate with early gene promoters via Ume6 and activate early meiotic genes (Raithatha et al. 2021). Given the above, we believe that the TID experiment with Ume6-TID is valid.

Figure 1. Western blot of Ume6-TID in WT and *ime1* and *rim11* deletion mutants. Indicated are the relative levels to Hxk1 and the 0h time point.

Referee #3:

EMBOJ-2023-115737

The authors have, for the most part, adequately addressed the points raised although there are still several issues to be resolved.

Thank you.

Major point

1. P.25 Rim11 nuclear accumulation can be rate-limiting for EMG activation

I did not quite understand the whole argument about what is rate-limiting. I understand that nuclear accumulation of Rim11 and Ume6 are concurrent, but so is Ime1. The authors say that Ime1 is rate-limiting in cluster 3, but it seems to me that Rim11's increase is pretty much concurrent with the increase of Ume6 there as well. In cluster 3, both Ime1 and Rim11 accumulate sharply from the beginning, which made me wonder why Ume6 accumulation is delayed there anyway. In any case, if there is a certain threshold in the amount of Ime1 necessary for induction of Ume6, Ime1 could as well be rate-limiting in every cluster. Then how could this experiment allow the authors to determine which factor is rate-limiting for EMG activation?

We agree that we cannot discriminate between thresholds and concurrent patterns of increase in Rim11, Ime1, and Ume6. Hence, we have rewritten the section accordingly, and removed statements suggesting that Rim11 is rate limiting based on the single-cell patterns.

Minor points

2. P.6 Indeed Rim15 and Mck1 have been shown to be substrates of Ume6

Is this right? I think it is the other way around.

Indeed, it is the other way around. We corrected the sentence accordingly.

3. P.7 The Rim11-mNG strain exhibited wild type (WT) kinetics of meiosis..
P.19 The Ume6-TID exhibited a phosphorylation shift dependent on Rim11 and Ime1, and it did not impact meiotic entry...

To me, meiosis of the strain producing Rim11-mNG looks delayed, by 1 hr or so, as seen in Fig. EV1A. This trend is also evident in Fig. EV2 B and C. Similarly the UME6-TID strain is also delayed as seen in Fig. S4D. Then I am wondering why the authors do not admit these trends.

We agree. We have rewritten the sections and acknowledge that there is a delay in meiotic entry.

4. Figure 6E

What does 0 hr and 1 hr represent? Time after addition of copper, IAA, or both?
Authors should specify that.

We have added the information the figure and further clarified in the legend.

5. Figure 7D, left panel

What is "e"?

Thank you for spotting a formatting error. It was supposed to display Rim11-mKok.
Corrected accordingly.

6. P.61, third line from the bottom

"in induced in WT.." does not make sense.
"and induced"

7. P.11 ...gene deletions in TPK2 and TPK3 led to an overall increase in Rim11-mNG signal YPD.

in YPD?

Corrected accordingly.

8. P.14 These data suggest that Mds3, but not Pmd1, acts as a repressor of *IME2* expression, consequently influencing the onset of meiosis.

It seems to me that the fact that the *mds3 pmd1* double mutant does not exhibit rapid induction of *Ime2* is not supported by this suggestion.

The double mutant does not show the same extent of increase in *IME2* transcript as the *mds3* mutant. Likewise, the double mutant did not show a faster onset of meiosis compared to the *mds3* mutant. A plausible explanation is that double mutant has a negative effect on meiosis that is unrelated to Rim11 localization.

9. P.17 However, unlike Ume6, Rim11 phosphorylates *Ime1* in cells grown in pre-sporulation medium.

Here, *Ime1* phosphorylation indeed relies on Rim11, but that does not necessarily mean that Rim11 phosphorylates *Ime1*.

We agree. We have rephrased that sentence as suggested.

"However, unlike Ume6, *Ime1* is phosphorylated in a Rim11-dependent manner in cells grown in pre-sporulation medium."

10. P.20 Recent work showed that Ime1 can be made indispensable for entry into meiosis ..

It must be the other way around. Ime1 can be dispensable under certain conditions, not indispensable.

Corrected accordingly.

11. P.23 The four-color strain was able to undergo meiosis and form viable spores (92 {plus minus} % viability)

92 {plus minus} ?

Corrected accordingly. "(92% viability)".

12. P.23 Furthermore, Ime1 and Ume6 nuclear levels were up-regulated in cells entering meiosis but not in non-meiotic cells (Figure 7A, and Figure EV5A).

Figure EV5B?

Corrected accordingly.

13. Figure EV5E

No broken line in the left panel.

Corrected accordingly.

14. P.24 We conclude that Rim11 and Ime1 nuclear accumulation starts roughly at the same time, while Rim11 peaks well prior to Ime1 and Ume6.

I suppose it is "Ume6" and Ime1 instead of "Rim11" and Ime1?

We removed the first part of the sentence.

"We conclude that nuclear Rim11 levels peak well prior to Ime1 and Ume6."

Referee #4:

This revised manuscript describes a scientifically rigorous and comprehensive analysis of the mechanisms that tie PKA, TOR, Ime1, Ume6, and Rim11 together in a network that controls meiotic induction in yeast. My criticisms of the technical/scientific aspects of the original submission were minor and these criticisms have been addressed. Comments made about significance of the original submission were not addressed by the authors. However, as described in my initial review, it is clear that the paper integrates a lot of new data into the prevailing model for meiotic induction and this moves the field forward in a significant way. A problem with the original submission was the way it was presented, which made it difficult to read. I found this revision just as difficult to read as the original article and do not

believe this criticism has been adequately addressed. The text of this manuscript needs to be tightened up throughout. There are multiple errors/grammatical problems on most pages of the Introduction, Results and Discussion, in the Figure legends and elsewhere (too numerous to mention). The paper would benefit from significant editing. While I appreciate the challenges of presenting so much data on multiple well-studied regulatory proteins in a single paper, I feel that additional efforts are required. It would be a shame if the outstanding science in this article received less readership than it deserves because of presentation issues. A few illustrative examples are below.

1- The section showing that "Rim11 nuclear accumulation is required for entry into meiosis" on p8 was particularly hard to get through. The point of all this is presumably to convince the reader that "nuclear accumulation of Rim11 is required for Ume6 phosphorylation and thus IME2 transcription" This should be communicated in a more linear, straightforward, and accessible manner.

We agree. We think that the addition of controls and the clarification of the genotypes during revision made the section less accessible for readers. We have rewritten the section and simplified as much as possible.

2- The liberal inclusion of "Supplementary and Appendix" figures and videos made the job of getting through the article daunting. More effort to explain why the data has been included and help the reader along would help. For example, two thirds of the way through p 10 it is stated that "pCUP1-NLS-RIM11 induced cells also showed a comparable Ume6 migration pattern to WT, and the onset of meiosis was delayed in pCUP1-NLS-RIM11 cells (Figure 2H and Appendix Figure 1F)" This distracts the reader without driving the story forward. If the authors feel it is important to include these findings, then something along the lines of "driving overexpressed Rim11 into nuclei with pCUP1-NLS-RIM11 did not noticeably effect the migration of Ume6 and only modestly delayed the kinetics of meiosis". These findings show.... A minor point is that it should be emphasized that the RIM11 chromosomal locus is replaced with pCUP1-NLS-RIM11 in these experiments (this is not a plasmid or other genetic manipulation). These kinds of things can help the naive reader understand the considerable rigor and care that was used in the study.

We have re-written the section and removed the *CUP1-RIM11* data.

3- Another problem related to the numerous pieces of data in figures proper, in the "expanded view (EV)" and in supplementary figures or movies is that the authors do not always explicitly state where a particular piece of data actually is. A simple example, on p 11 states "cells shifted to SPO plus rapamycin showed increased nuclear localization of Rim11, and a more rapid increase in Ume6 phosphorylation (Figure 3B and Appendix Figure S2A)." So, if the reader is interested in looking at one of these pieces of data they first need to sort out which conclusion is supported by which figure (or whether both actually show the same thing) and where to look next. These instances should be recast to something like "of Rim11 (Figure 3B), and a more rapid increase in Ume6 phosphorylation (Appendix Figure S2A)." This is a problem throughout the paper that makes it difficult for an attentive reader.

We have tried to clarify the references to the data as much as possible.

4- The fact that glucose and Tpk1 control the level of Rim11 is a significant piece of how this network works (as discussed around the top of p12) [but maybe the Mitchell lab previously showed this, and if so, this should be referenced]. If this is an original finding, the data to support this should not be relegated to a supplementary figure.

There is one report that states that PKA phosphorylates Rim11 (which we cited), but the increase in Rim11 expression has not been clearly described before. We have now included the data in the main figure.

5- On the middle of p12, the authors describe transferring cells from starvation conditions to rich medium. It is unclear exactly what is being done here. Perhaps it is somewhere in the paper but this needs to be explained clearly at this point.

We have added a scheme to the figure panel and updated the legend. We note that the information was already presented in methods section.

6- I still do not feel that the *mds3* and *pmd1* data substantially advance the story. Irrespective, if the authors choose to keep this section in the paper, the image in Fig. 4F should be straightened out and the labels are appropriately aligned.

We have aligned the labels and straightened the image.

7- The liberal use of adjectives like "critical" and "important" is distracting. For example, on p3 the phrase "undergo a critical cell-fate decision" should just be "undergo a cell-fate decision" and on p6 the phrase "Ume6, a key prime substrate of Rim11" should just be "Ume6, a substrate of Rim11". Similar changes throughout would help reduce unnecessary words.

We went over the manuscript and removed unnecessary words.

8- The discussion could use some additional work.

9- Perhaps referring to the model in Fig. 7F in the discussion to guide the reader or even using the model as a way to direct the discussion might help to clarify how this system works and would also be helpful in summarizing the new findings of the paper.

We have updated the discussion.

References:

Raithatha, Sheetal A., Shivani Vaza, M. Touhidul Islam, Brianna Greenwood, and David T. Stuart. 2021. "Ume6 Acts as a Stable Platform To Coordinate Repression and Activation of Early Meiosis-Specific Genes in *Saccharomyces Cerevisiae*." *Molecular and Cellular Biology* 41 (7): e0037820.

Dear Dr. van Werven,

Thank you for the submission of your revised manuscript to The EMBO Journal. We have now received the comments of three referees that were asked to re-assess your study (included below). As you will see, the referees are satisfied with the revision, and they acknowledge that their previous concerns have been largely addressed. However, referees #3 and #4 also identify a few remaining minor issues that must be resolved before we can proceed with acceptance of the manuscript for publication in The EMBO Journal. I would thus like to invite you to submit another revised version of your manuscript addressing all remaining concerns and suggestions of the referees, along with a detailed point-by-point response to their comments.

We look forward to seeing a final version of your manuscript as soon as possible. Please use this link to submit your revision: <https://emboj.msubmit.net/cgi-bin/main.plex>

Yours sincerely,

Referee #2:

The authors have adequately addressed my concerns. Therefore, I recommend publication of the manuscript.

Referee #3:

EMBOJ-2023-115737R1

The authors have adequately addressed the points raised except minor point 8.

8. P.14 These data suggest that Mds3, but not Pmd1, acts as a repressor of IME2 expression, consequently influencing the onset of meiosis.

It seems to me that the fact that the mds3 pmd1 double mutant does not exhibit rapid induction of lme2 is not supported by this suggestion.

The double mutant does not show the same extent of increase in IME2 transcript as the mds3 mutant. Likewise, the double mutant did not show a faster onset of meiosis compared to the mds3 mutant. A plausible explanation is that double mutant has a negative effect on meiosis that is unrelated to Rim11 localization.

The point is, the statement "These data suggest that Mds3, but not Pmd1, acts as a repressor of IME2 expression", is not compatible with the observations. One possibility is to remove the pmd1 and pmd1 mds3 data from Fig.4C and D. They are not analyzed in Fig. 4E and 4F anyway.

I just noticed an additional minor point.

P.9 line 3- "..., we analysed its impact expression of the protein kinase lme2"
should read:

..., we analysed its impact on the expression of the protein kinase lme2

Referee #4:

As before, I do not have significant scientific or technical concerns with this paper. Indeed, the quality of the science is very high. My major concerns with the prior versions of this paper were related to clarity of presentation. The current version is a significant improvement over the previous versions and many of my concerns have been addressed. Nevertheless, there are still minor changes that would improve the paper. While most of these changes could be handled by the copy editor, in a few instances the suggested changes are related to organizational issues and might not be straightforward. For that reason, I suggest that the authors take one more shot at improving the presentation of the article.

The sections below should be modified for clarity:

1- p12 - the paragraph with the subheading "PKA mediated phosphorylation..." is difficult to follow. For example, it would be helpful to explicitly define what is meant by "which mis-regulates the onset of meiosis" so that the reader understands. For the sentence "We found that Rim11 protein expression..." the experiment is not defined (e.g. "during the first few hours after cells were transferred to sporulation medium" or something similar should be added). Overall this paragraph needs to be modified so that the naive reader can understand what has been published and what is new and therefore follow the logic.

2- p19 - the section "Rim11 and Ime1 can be dispensable for EMG induction and meiosis" is poorly organized and needs to be recast starting with the subheading. Perhaps something like "Mutations that tether Ime1 to Ume6 bypasses the requirement of Rim11 for EMG induction and meiosis"? Irrespective, leading into this section by describing a tenuous involvement in replication stress and lipid synthesis does not add much and is likely to confuse the reader. The rest of this section needs to be made more linear, cleaned up and made easier on the reader. A suggestion is to concisely summarize relevant findings from the Harris and Unal paper and then describe how this system was used to study the role of Rim11.

Below are a few minor issues that should be modified (there are other similarly minor issues that should be taken care of - these are just the ones I made notations of).

3- 7th line on p13 change "determined" to "measured".

4- Last line on p13 insert "in the *mds3Δ* mutant" for clarity (also there is a typo in this paragraph).

5- p16 last paragraph second sentence: rewrite to "*ime1Δ* cells did not exhibit a Rim11-dependent shift in Ume6 migration..."

6- p16 last sentence "Ime1 possibly provides a third signaling input that promotes..."

7- p18 line 4 and 6 remove "relatively" and line 20 remove "relative" (enriched means relative to something).

8- p24: the last sentence of first paragraph is a conclusionary statement that needs to be softened up somehow (use of "suggests/is consistent with" or something similar). The same is true for the use of the word "determine" in the first sentence of the next paragraph.

Referee #2:

The authors have adequately addressed my concerns. Therefore, I recommend publication of the manuscript.

Thank you.

Referee #3:

EMBOJ-2023-115737R1

The authors have adequately addressed the points raised except minor point 8.

8. P.14 These data suggest that Mds3, but not Pmd1, acts as a repressor of IME2 expression, consequently influencing the onset of meiosis.

It seems to me that the fact that the *mds3 pmd1* double mutant does not exhibit rapid induction of *Ime2* is not supported by this suggestion.

The double mutant does not show the same extent of increase in IME2 transcript as the *mds3* mutant. Likewise, the double mutant did not show a faster onset of meiosis compared to the *mds3* mutant. A plausible explanation is that double mutant has a negative effect on meiosis that is unrelated to Rim11 localization.

The point is, the statement "These data suggest that Mds3, but not Pmd1, acts as a repressor of IME2 expression", is not compatible with the observations. One possibility is to remove the *pmd1* and *pmd1 mds3* data from Fig.4C and D. They are not analyzed in Fig. 4E and 4F anyway.

We have removed the data of the *pmd1 mds3* double mutant. We think it is important to keep *pdm1* single mutant as Mds3 and Pmd1 are paralogs.

I just noticed an additional minor point.

P.9 line 3- "..., we analysed its impact expression of the protein kinase *Ime2*" should read:

..., we analysed its impact on the expression of the protein kinase *Ime2*

Corrected accordingly.

Referee #4:

As before, I do not have significant scientific or technical concerns with this paper. Indeed, the quality of the science is very high. My major concerns with the prior versions of this paper were related to clarity of presentation. The current version is a significant improvement over the previous versions and many of my concerns have been addressed. Nevertheless, there are still minor changes that would improve the paper. While most of these changes could be handled by the copy editor, in a few

instances the suggested changes are related to organizational issues and might not be straightforward. For that reason, I suggest that the authors take one more shot at improving the presentation of the article.

The sections below should be modified for clarity:

1- p12 - the paragraph with the subheading "PKA mediated phosphorylation..." is difficult to follow. For example, it would be helpful to explicitly define what is meant by "which mis-regulates the onset of meiosis" so that the reader understands. For the sentence "We found that Rim11 protein expression..." the experiment is not defined (e.g. "during the first few hours after cells were transferred to sporulation medium" or something similar should be added). Overall this paragraph needs to be modified so that the naive reader can understand what has been published and what is new and therefore follow the logic.

Thank you for the suggestion. We have attempted to re-write the section as instructed above.

2- p19 - the section "Rim11 and Ime1 can be dispensable for EMG induction and meiosis" is poorly organized and needs to be recast starting with the subheading. Perhaps something like "Mutations that tether Ime1 to Ume6 bypasses the requirement of Rim11 for EMG induction and meiosis"? Irrespective, leading into this section by describing a tenuous involvement in replication stress and lipid synthesis does not add much and is likely to confuse the reader. The rest of this section needs to be made more linear, cleaned up and made easier on the reader. A suggestion is to concisely summarize relevant findings from the Harris and Unal paper and then describe how this system was used to study the role of Rim11.

Thank you for the suggestion. We have attempted to re-write the section as instructed above.

Below are a few minor issues that should be modified (there are other similarly minor issues that should be taken care of - these are just the ones I made notations of).

3- 7th line on p13 change "determined" to "measured".

Corrected accordingly.

4- Last line on p13 insert "in the mds3 Δ mutant" for clarity (also there is a typo in this paragraph).

Corrected accordingly.

5- p16 last paragraph second sentence: rewrite to "ime1 Δ cells did not exhibit a Rim11-dependent shift in Ume6 migration..."

Corrected accordingly.

6- p16 last sentence "Ime1 possibly provides a third signaling input that promotes..."

Corrected accordingly.

7- p18 line 4 and 6 remove "relatively" and line 20 remove 'relative' (enriched means relative to something).

Corrected accordingly.

8- p24: the last sentence of first paragraph is a conclusionary statement that needs to be softened up somehow (use of "suggests/is consistent with" or something similar). The same is true for the use of the word "determine" in the first sentence of the next paragraph.

Corrected accordingly.

Dear Folkert,

Congratulations on an excellent manuscript, I am very pleased to inform you that it has been accepted for publication in The EMBO Journal. Thank you for your comprehensive responses to the referee concerns.

If you have any questions, please do not hesitate to contact the Editorial Office. Thank you for your contribution to The EMBO Journal, it has been a pleasure working with you.

Best regards,

Ioannis
